# Brain and psychological determinants of placebo pill response in chronic pain patients

Etienne Vachon-Presseau [1], Sara E. Berger [1,2], Taha B. Abdullah[1], Lejian Huang[1], Guillermo A. Cecchi [2], James W. Griffith[3], Thomas J. Schnitzer[4,5] & A. Vania Apkarian[1,5,6]

The placebo response is universally observed in clinical trials of pain treatments, yet the individual characteristics rendering a patient a 'placebo responder' remain unclear. Here, in chronic back pain patients, we demonstrate using MRI and fMRI that the response to placebo 'analgesic' pills depends on brain structure and function. Subcortical limbic volume asymmetry, sensorimotor cortical thickness, and functional coupling of prefrontal regions, anterior cingulate, and periaqueductal gray were predictive of response. These neural traits were present before exposure to the pill and most remained stable across treatment and washout periods. Further, psychological traits, including interoceptive awareness and openness, were also predictive of the magnitude of response. These results shed light on psychological, neuroanatomical, and neurophysiological principles determining placebo response in RCTs in chronic pain patients, and they suggest that the long-term beneficial effects of placebo, as observed in clinical settings, are partially predictable.

[1] Department of Physiology, Northwestern University Feinberg School of Medicine, 710N Lake Shore Drive, Room 1020, Chicago, IL 60611, USA. [2] Healthcare and Life Sciences Department, IBM Watson Research Center, 1101 Kitchawan Rd, Yorktown Heights, NY 10598, USA. [3] Department of Medical Social Sciences, Northwestern University Feinberg School of Medicine, 710N Lake Shore Drive, Room 1020, Chicago, IL 60611, USA. [4] Departments of Internal Medicine and Rheumatology, Northwestern University Feinberg School of Medicine, 710N Lake Shore Drive, Room 1020, Chicago, IL 60611, USA. [5] Department of Physical Medicine and Rehabilitation, Northwestern University Feinberg School of Medicine, 710N Lake Shore Drive, Room 1020, Chicago, IL 60611, USA. [6] Department of Anesthesia, Northwestern University Feinberg School of Medicine, 710N Lake Shore Drive, Room 1020, Chicago, IL 60611, USA. These authors contributed equally: Etienne Vachon-Presseau, Sara E. Berger. Correspondence and requests for materials should be addressed to A.V.A. (email: a-apkarian@northwestern.edu)

The placebo response refers to an improvement in symptoms caused by receiving an inert treatment. The phenomenon has been observed across different conditions, biological systems, and treatment types[1]. Placebo analgesia is especially relevant in the management of chronic pain, since most pharmacological treatments have long-term adverse effects or addictive properties[2], or show only modest improvements that are insufficient to achieve clinically meaningful amelioration of disability[3]. Placebo responses are observed universally in almost all randomized placebo-controlled clinical trials (RCT), including those testing chronic pain treatments[4–6]. Importantly, the effect size of placebo response is often equivalent to the active treatment studied[7] and often even greater than that seen in conventional therapy[6]. The duration of placebo responses has been shown to be comparable to those achieved with active treatments and the magnitude of placebo responses appear to be increasing in recent RCTs of neuropathic pain conducted in the US[7]. This implies that the placebo effect will remain a confounding nuisance in clinical testing and practice as long as its underlying properties remain unknown, and as long as approaches that predict this response or reliably harness it for medical benefit are not developed.

The neurobiological mechanisms underlying the placebo effect have been primarily studied for acute responses to conditioning-type manipulations or suggestion-based paradigms in healthy individuals, usually performed in the laboratory setting[8–11]. In healthy subjects, placebo response and its neural and psychological correlates lack consistency across different routes of administration[12–14]. Moreover, the translation of such findings to clinical settings is questionable not only because chronic pain patients exhibit distinct brain anatomy and neurophysiology[15–18] but also because such patients are repeatedly exposed to a myriad of medical rituals which may bias expectations toward treatment. Thus, it is likely that the principles of placebo pill analgesia in clinical settings may not be captured by experimental models of placebo.

A number of brain imaging studies have examined placebo response in the setting of a RCT. Some of these studies have demonstrated changes in brain functions as a consequence of placebo effects[19,20]. Others have shown that brain functions may actually predispose chronic pain patients to respond to a placebo treatment. RCTs comparing a lidocaine patch to placebo treatment in chronic low back pain and duloxetine to placebo treatment in osteoarthritis indicated that placebo response was predicted, respectively, by functional connectivity of the medial prefrontal with the insula[21] and the dorsolateral prefrontal cortex with the rest of the brain[22]. Unfortunately, these studies have the caveat of not relying on a no treatment arm to control for the natural history of the patients (e.g., spontaneous remission or regression to the mean), and thus do not account for inherent variability in measurements of brain activity when comparing baseline to post-treatment periods. One study has addressed these confounds by comparing active placebo to an inactive control group in major depressive disorder, showing that increased placebo-induced μ-opioid neurotransmission in limbic regions was predictive of greater improvement of symptoms in response to subsequent antidepressant treatment[23]. Similarly controlled studies in chronic pain patients are necessary to investigate placebo responses in clinical settings.

This prospective cohort study included four neuroimaging sessions, a large battery of questionnaires assessing personality traits and pain characteristics, and a proper no treatment arm that allowed us to disentangle placebo pill-related analgesia from non-specific effects. Given that the RCT placebo effect is embedded in predictable psychology and neurobiology and given that metabolic activity and functional connectivity change following placebo response, we designed a comprehensive RCT with two identical treatment periods (placebo or active treatment), each followed by a washout period. This allowed us to investigate the duration of response and potential differences in within-subject occurrence/re-occurrence of placebo response. The repeat brain imaging sessions were used: (1) to identify functional networks (constructed from spontaneous fluctuations in BOLD signal) predictive of placebo response prior to placebo exposure; and (2) to test stability of identified networks post-exposure to placebo, as such networks are known to be malleable and may change as a function of learned associations[24].

The primary aim of this study was to identify the psychological factors and the brain properties collected prior to placebo treatment that would determine placebo response. Given previous results, we hypothesized that responses to placebo pills are predetermined by functional connectivity between regions such as the medial and lateral prefrontal cortex, the anterior cingulate cortex, and the subcortical limbic structures, since they have been implicated in both experimental[25–27] and clinical placebo response[20–23]. We also hypothesized that other psychological and neurophysiological determinants, which would be more specific to the clinical setting and to chronic pain patients will also contribute to the prediction of placebo response. The second aim of this study was to examine the longitudinal effect of treatment exposures on brain properties. Based on prior studies[19,20], we hypothesized that some components of the functional networks predisposing patients to placebo response would change as a consequence of placebo response while others would remain stable.

Consistent with our predictions, we demonstrate-specific psychological factors, anatomical properties, and functional coupling of the lateral prefrontal cortex predetermine placebo pill responses. We show that components of the response-predictive functional network showed transient properties, dependent on the placebo response. Finally, a fully cross-validated algorithm applied on psychological factors and functional connectivity prior to exposure to placebo treatment successfully predicted response magnitude to placebo pill treatment.

## Results

**Specificity of placebo pill response**. This neuroimaging-based placebo RCT was conducted in 129 CBP patients assessed for eligibility, 63 of whom were analyzed (Supplementary Figure 1). Participants visited the lab on six occasions over 8 weeks and underwent identical scanning protocols on four of those visits (Fig. 1a). Throughout the duration of the trial, participants used a visual analogue scale, displayed on a smartphone app (Supplementary Figure 2), to rate back pain intensity two times per day in their natural environment. These ecological momentary assessments (EMAs) represented the primary pain measurement of this study and were used to determine placebo response. Secondary pain measurements were also collected but only in the lab on each visit. These included a numeric verbal recall of their average pain experienced over the last week (pain memory), a numeric rating scale at time of visit (NRS, commonly used to quantify pain in clinical trials[28]), the McGill Pain Questionnaire (MPQ) sensory and affective scales, and the PainDetect. The covariance matrix across the different pain measurements at baseline is presented in Supplementary Figure 3 and the demographics in Supplementary Table 1.

Our study was designed to test for both the effects of placebo pill ingestion and the effects of placebo response. We initially tested for the effect of placebo pill ingestion by comparing the analgesia between the 43 patients receiving placebo (PTx) with the 20 patients comprising the no treatment arm (NoTx); for this, we used the EMAs collected with the phone app. On average, PTx

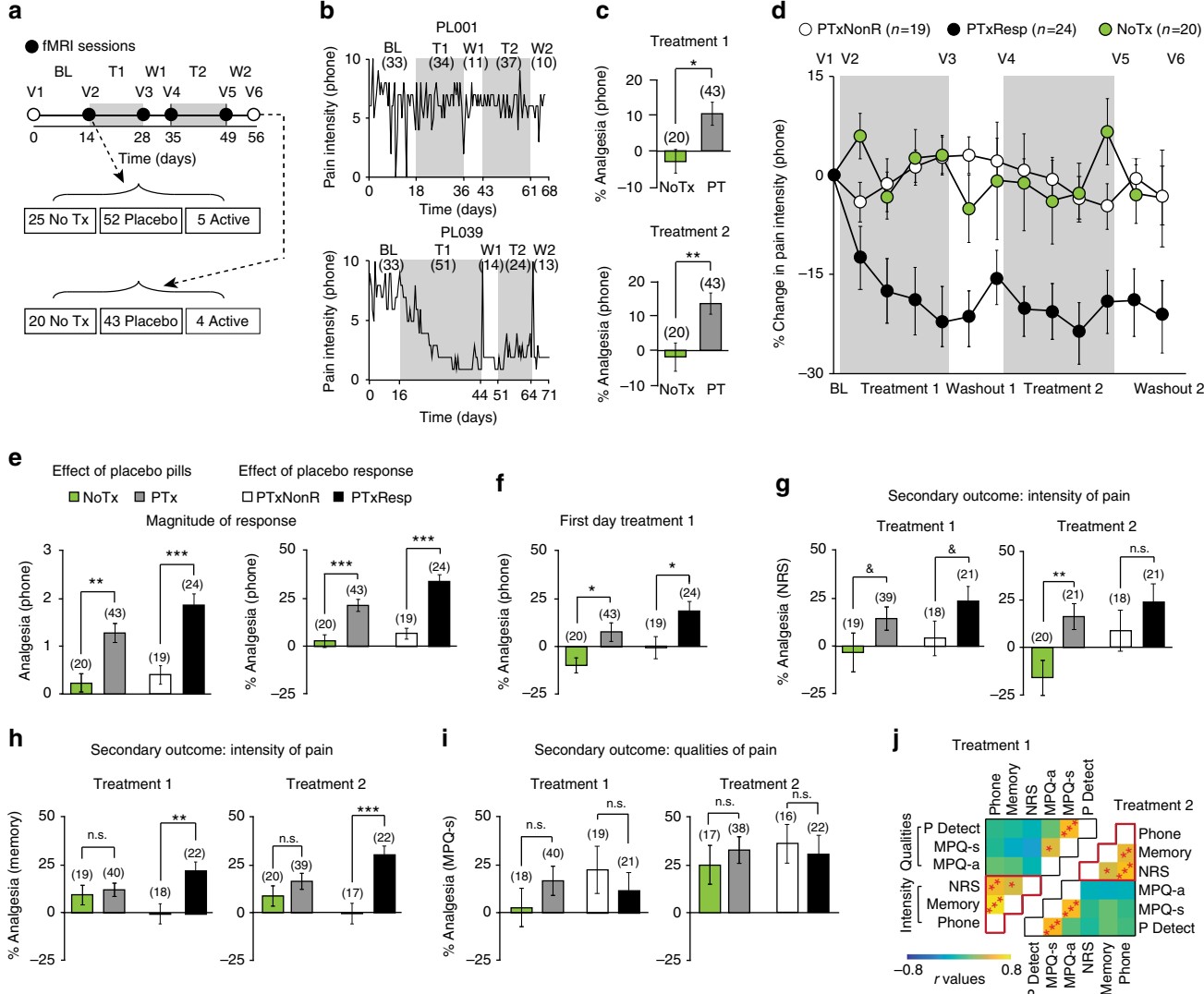

**Fig. 1** Placebo pill ingestion diminishes back pain intensity while trial participation non-specifically decreases qualitative pain outcomes. **a** Experimental design and time line: CBP patients entered a six-visit (V1-6) 8-week randomized controlled trial, including baseline (BL), treatment (T1, T2), and washout (W1, W2) periods. Participants entering and completing the study are indicated. **b** Example time series of smartphone app pain ratings in 2 patients (values in parentheses are numbers of ratings). **c** The patients receiving placebo treatment (PTx) showed lower pain levels during the last week of each treatment period, compared to the patients in the no treatment arm (NoTx). **d** Within-subject permutation tests between pain ratings entered during BL and T1 or T2 identified responders in PTx and NoTx groups. The group-averaged %change in phone app ratings of back pain intensity (2 bins/week) is displayed in **d**. **e** The magnitude of response represents the stronger response between the two treatment periods and is displayed in original score (left) and after conversion in %analgesia (right). **f** Placebo analgesia was present the first day after placebo pill ingestion. **g**, **h** Pain intensity decreased by about 20–30% for numerical rating scale (NRS) and memory during both treatment periods. **i** Qualitative outcome measures (McGill pain questionnaire, MPQ-sensory) decreased in time similarly in all groups, including the NoTx and PTxNonR. **j** Principal component analyses clustered pain outcomes into 2 factors, segregating intensity and quality. Group by time by Pain (intensity vs quality) interaction indicated that only the pain intensity was diminished by placebo pill ingestion. In **c**–**i** number of subjects are in parentheses, $^{\&}p < 0.12$; $^{*}p < 0.05$; $^{**}p < 0.01$; $^{***}p < 0.001$, n.s. not significant. Error bars indicate SEM. For each figure, a description of the analyses and $p$-values are reported in Supplementary Table 7

showed stronger diminution in pain intensity compared to NoTx (controlling for regression to the mean, spontaneous variations in symptoms, and/or placebo response to cues other than the pills) (Fig. 1c; Supplementary Table 7 summarizes all statistical outcomes). The effect sizes (E.S.) of the placebo pill ingestion were 0.62 for treatment 1 and 0.73 for treatment 2 (Supplementary Figure 4), which are comparable with E.S. reported in the placebo literature in chronic pain patients[29,30].

Next, the 63 patients were dichotomized into Resp and NonR based on a permutation test performed on the EMAs collected with the phone app (baseline vs. treatments 1 or 2) to determine within-subject improvement of symptoms (cutoff was set at

$p < 0.05$). The response rate indicated that placebo pill ingestion increased the frequency of CBP patients showing pain reduction —more patients were classified as responders in the placebo treatment group (PTxResp 24/43: response rate of 55%) compared to the no treatment group (NoTxResp 4/20: response rate of 20%; $\chi^2 = 7.09$, $p = 0.008$). This demonstrates that placebo pill ingestion increased the response rate when considering within-subject pain entries.

The time course of pain ratings across the RCT is displayed for PTxResp, PTxNonR, and NoTx groups in Fig. 1d (the original scale scores (absolute values) are presented in Supplementary Figure 5; time course for NoTxResp is presented in

**Table 1 Absolute levels of pain prior to treatment showed no group differences on all pain measures**

|         | Phone       | Memory      | NRS           | MPQa        | MPQs         | Pain detect  |
|---------|-------------|-------------|---------------|-------------|--------------|--------------|
| PTxNonR | 6.22 (1.12) | 6.59 (1.48) | 55.10 (23.14) | 3.21 (2.44) | 12.54 (5.34) | 9.97 (7.44)  |
| PTxResp | 6.10 (1.33) | 7.26 (1.29) | 60.75 (23.21) | 3.54 (2.87) | 11.25 (4.2)  | 9.58 (5.33)  |
| NoTx    | 5.68 (1.14) | 6.65 (1.66) | 48.85 (23.33) | 4.12 (2.80) | 14.90 (6.11) | 13.30 (6.15) |
| *p_vals* | 0.33       | 0.26        | 0.24          | 0.58        | 0.08         | 0.12         |

The phone entries were averaged across the baseline period and the secondary pain assessments were collected in the lab at visit 2. These values were used as baseline measurements to determine the %analgesia post treatment. There was no significant difference between the groups. Number in parenthesis represents the standard deviation

Supplementary Figure 6). Importantly, pain levels at baseline were equivalent between PTxResp, PTxNonR, and NoTx groups (Table 1) and external factors such as pain intensity during baseline, phone rating compliance, overall treatment compliance, treatment duration, rescue medication usage, and previous medication usage were not related to placebo response (Supplementary Table 2). On average, responders showed a diminution in back pain intensity that stabilized to a constant value of about 20% analgesia for both treatment and washout periods (T1, T2, and W1, W2; Fig. 1d). Because some patients responded more strongly to one treatment period over the other, we calculated the magnitude of response as the highest %analgesia between the two treatment periods. On average, the magnitude of response in the PTxResp was 1.7 units higher than in the NoTx arm (E.S.: 1.71), which corresponded to a 33% analgesia (Fig. 1e). Interestingly, phone app pain ratings at the start of treatment (2 ratings on day 1 of T1) already differentiated PTx from NoTx, and PTxResp from PTxNonR (Fig. 1f), indicating that observed analgesia is temporally coupled with the introduction of placebo pills.

Given the phone app pain stratification, the effects of placebo pill ingestion and/or the effects of placebo response were observed on two additional measures of back pain intensity used as secondary pain outcomes: NRS (Fig. 1g), and pain memory (Fig. 1h). Results show that NRS collected in the lab correlated with our primary pain outcome (Supplementary Figure 3) and captured the effects of placebo pill ingestion and placebo response. Memory of pain also correlated with our primary pain outcome and strongly dissociated PTxResp and PTxNonR, but also showed a non-specific improvement with exposure to the trial (PTx and NoTx arms equally improving). Despite some variability across these measures of pain intensity, the effects of placebo pill ingestion and the effects of placebo response were globally concordant across these outcomes. However, MPQ sensory and affective scales and PainDetect poorly correlated with our primary pain outcome (Supplementary Figure 3), did not differentiate between treatment cohorts, and groupings defined by the pain app showed improvement of symptoms in all groups (Fig. 1i). We conclude that the RCT placebo response is composed of two components: (1) a pill ingestion-related response specifically impacting perceived intensity of chronic pain (Supplementary Table 3), and 2) a non-specific response reflecting the effect of time or the mere exposure to healthcare (visits) that modulates qualitative pain measures (Fig. 1j). For the rest of the study, we concentrate on unraveling the mechanisms of the placebo-induced decrease in back pain intensity.

**Blinding of the analyses**. In this study, all brain imaging and questionnaires data were analyzed blindly. We employed cell scrambling to generate two random labeling of patients and all group comparisons were performed three times (two times for scrambled codes and one time for real labels). After completing all the analyses, the real labeling was revealed during a public lab meeting. The results are reported only when the real labeling of patients could be properly identified based on the statistical tests.

**Psychological factors predisposing to placebo pill response**. First, we sought to identify psychological parameters predisposing CBP patients to the placebo pill response from a battery of 15 questionnaires with 38 subscales (Supplementary Table 4) collected at visit 1. Figure 2a shows the covariance across all factors used to assess personality and psychological states. Univariate statistics were used to assess group differences and correlations with the magnitude of response. Here, the real labeling and the scrambled codes yielded no significant comparisons when correcting for multiple comparisons (Fig. 2b). However, a number of subscales from the Multidimensional Assessment of Interoceptive Awareness (MAIA) questionnaire were tightly coupled with the magnitude of response, as was the quality of "openness" from the Neo-5 personality dimensions (Fig. 2c). In particular, Emotional Awareness (MAIA/e) and Not Distracting (MAIA/nd) were strongly correlated with %analgesia (Fig. 2d, e).

We also monitored positive and negative expectations at visit 2 and visit 4, prior to placebo pill exposure/re-exposure. Our results showed that positive and negative expectations at both visits were not different between PTxResp and PTxNonR and changes in the levels of expectations following treatment 1 (representing the update of expectations) were not different between the groups (Supplementary Table 5). Thus, although a large body of literature demonstrates the influence of expectations on the placebo response[31], expectations were not a significant factor in the current study.

**Anatomical properties predisposing to placebo pill response**. We secondly sought for anatomical properties predisposing CBP patients to placebo pill response at visit 2 (prior to treatment 1). The volumes of the NAc, amygdala, and hippocampus were first examined because they represent risk factors for developing pathological emotional states[32,33], chronic pain[34], and placebo response in healthy individuals[35]. Comparing subcortical volumes between PTxResp and PTxNonR was not informative. Given the recent evidence that subcortical volume asymmetry can provide a brain signature for psychopathologies[36], we followed-up examining inter-hemispheric laterality of the combined volume of these three structures. The PTxResp showed rightward subcortical limbic volume asymmetry compared to PTxNonR, and this asymmetry was observed in all four visits/scans (Fig. 3a). Importantly, this result was validated using another brain segmentation software (Freesurfer, Supplementary Figure 7). The differences in anatomical properties of the cortex were assessed with gray matter density and cortical thickness (Supplementary Figure 8). Whole-brain cortical thickness measurements showed that PTxNonR had thicker cortex in the right superior frontal gyrus than PTxResp (Fig. 3b). These anatomical properties also mildly correlated with the magnitude of placebo response (Fig. 3c). The identification of brain morphological features, present before treatment and persisting throughout the study, provides evidence for placebo propensity stemming, in part, from stable brain biology. Performing this analysis using scrambled

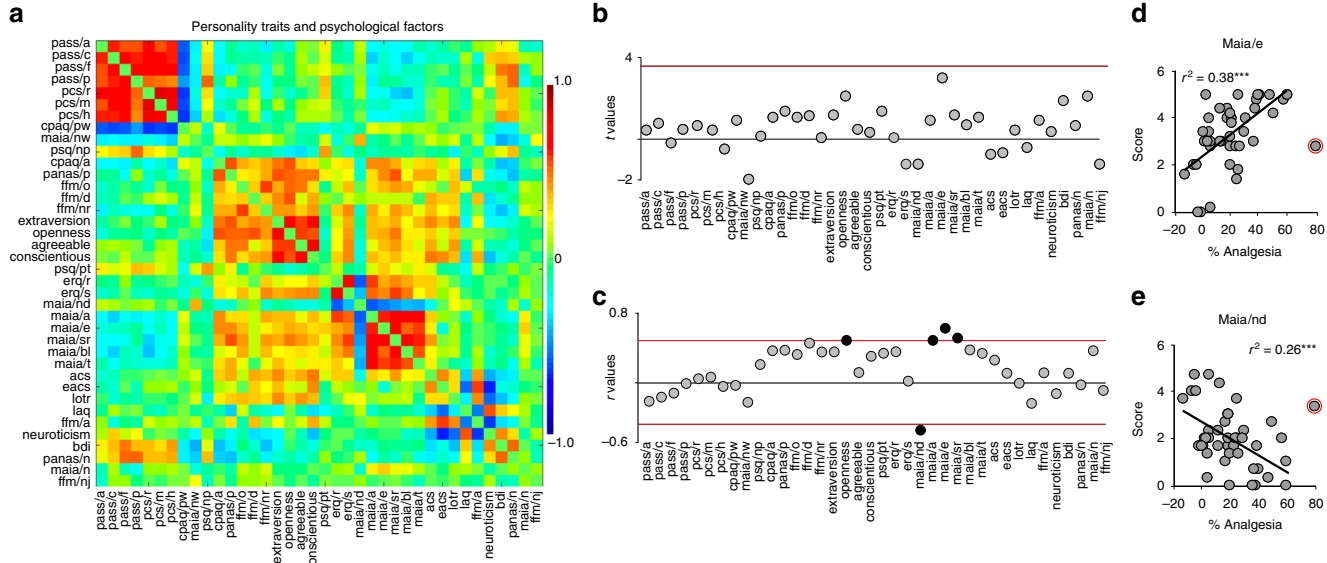

**Fig. 2** Personality traits and psychological factors were strongly linked to the magnitude of response. **a** The covariance matrix shows the relationship between each subscale in the questionnaire data described in detail in Supplementary Table 4. **b** Univariate statistics showed no group differences surviving Bonferroni correction for 37 comparisons. **c** However, openness (from the big 5 personality dimensions) and 4 out of 8 subscales from the Multidimensional Assessment of Interoceptive Awareness (MAIA)- Not Distracting (maia/nd), Attention Regulation (maia/a), Emotional Awareness (maia/e), and Self-Regulation (maia/sr)- correlated with the magnitude of response after correcting for multiple comparisons ($p < 0.0013$; shown in black). Patients with increased emotional awareness experience greater placebo analgesia (Maia/e displayed in **d**), whereas patients who tend not to ignore or distract themselves from discomfort experience less placebo analgesia (Mais/nd displayed in **e**). The red circle marks an outlier on the magnitude of response that was excluded. Other abbreviations used: Pain Anxiety Symptoms Scale (PASS) and its 4 subscales—Avoidance (pass/a), Cognitive (pass/c), Fear (pass/f), and Physiological (pass/p); Pain Catastrophizing Scale (PCS) and its 3 subscales—Rumination (pcs/r), Magnification (pcs/m), and Helplessness (pcs/h); Chronic Pain Acceptance Questionnaire (CPAQ) and its 2 subscores—Activity Engagement (cpaq/a) and Pain Willingness (cpaq/pw); the remaining 4 MAIA subscales—Noticing (maia/n); Not Worrying (maia/nw); Body Listening (maia/bl); and Trusting (maia/t); Pain Sensitivity Questionnaire (PSQ) and its 2 subscales—Painful (psq/p) and Non-Painful (psq/np); Positive and Negative Affect Scale (PANAS) with Positive (panas/p) and Negative (panas/n) subscales; Five Facets of Mindfulness (FFM) with its 5 subscales—Observe (ffm/o), Describe (ffm/d), Act with Awareness (ffm/a), Non-Judge (ffm/nj), and Non-React (ffm/nr); Emotional Regulation Questionnaire with its Reappraisal (erq/r) and Suppression (erq/s) subscales; Attentional Control Scale (acs); Emotional Attentional Control Scale (eacs); Life Orientation Test—Revised version (lotr); Loss Aversion Questionnaire (laq); and Beck Depression Index (bdi)

codes for labeling patients generated no significant group differences.

**Functional properties predisposing to placebo pill response.** We thirdly tested whether brain functions provided useful information to determine placebo pill response in our RCT. Here we examined brain networks constructed from rsfMRI to directly identify functional connectivity that predisposes patients' response prior to placebo pill treatment, at visit 2 (Supplementary Figure 9). Building on previous findings, we derived placebo-related networks of interest from results in OA patients exposed to placebo treatment in an RCT (Fig. 4a)[22]. We performed a modularity analysis segregating the functional networks into 6 communities, and restricted all analyses to the default mode network (DMN), sensorimotor (SM), and frontoparietal (FP) communities, due to their overlap with placebo-related networks observed in OA. Subcortical limbic regions were added along with the PAG because of their involvement for placebo response and in pain chronification (Supplementary Figure 9). No other exploratory analyses were performed outside of this initially planned strategy.

The connections differentiating the two PTx response groups were all connected to nodes located in the VLPFC (Broadman area 47) or the DLPFC (Broadman area 46). Precisely, PTxResp displayed stronger connections for the link VLPFC-PreCG, and weaker connections for links VLPFC-rACC and VLPFC/DLPFC-PAG (Fig. 4b, c). As expected, all three networks differentiated

PTxResp from PTxNonR at visit 2 (Fig. 4d–f). Moreover, the VLPFC-PreCG (Fig. 4d) and the VLPFC-rACC connectivity (Fig. 4e) showed neither a main effect of time nor an interaction effect of group × time, indicating that these connections represented time-invariant mechanisms that differentiate PTxResp from PTxNonR across all visits. On the other hand, the initial differences between groups of VLPFC/DLPFC-PAG connections dissipated by visit 4 (Fig. 4f). These results therefore demonstrate the existence of a lateral prefrontal-functional network, whose components either stably or transiently determine the likelihood of placebo pill response. Importantly, each component of this VLPFC/DLPFC-functional network also tracked the magnitude of placebo response (Fig. 4g). Here again, the scrambled codes yielded no significant group differences.

We then examined the variability of each of these anatomical and functional brain measurements in patients of the NoTx arm. We observed no changes across the visits, indicating stability of the measure without placebo effects (Supplementary Figure 10). We further tested if the mere exposure to placebo pills, regardless of the response, impacted these brain measurements by comparing the PTx group with the NoTx group. These analyses revealed absence of pill exposure effects on anatomical and functional brain measurements (Supplementary Figure 10). The result suggests that the changes observed in VLPFC/DLPFC-PAG connectivity were primarily driven by the actual placebo response rather than by mere pill exposure or inherent variability in the measurements.

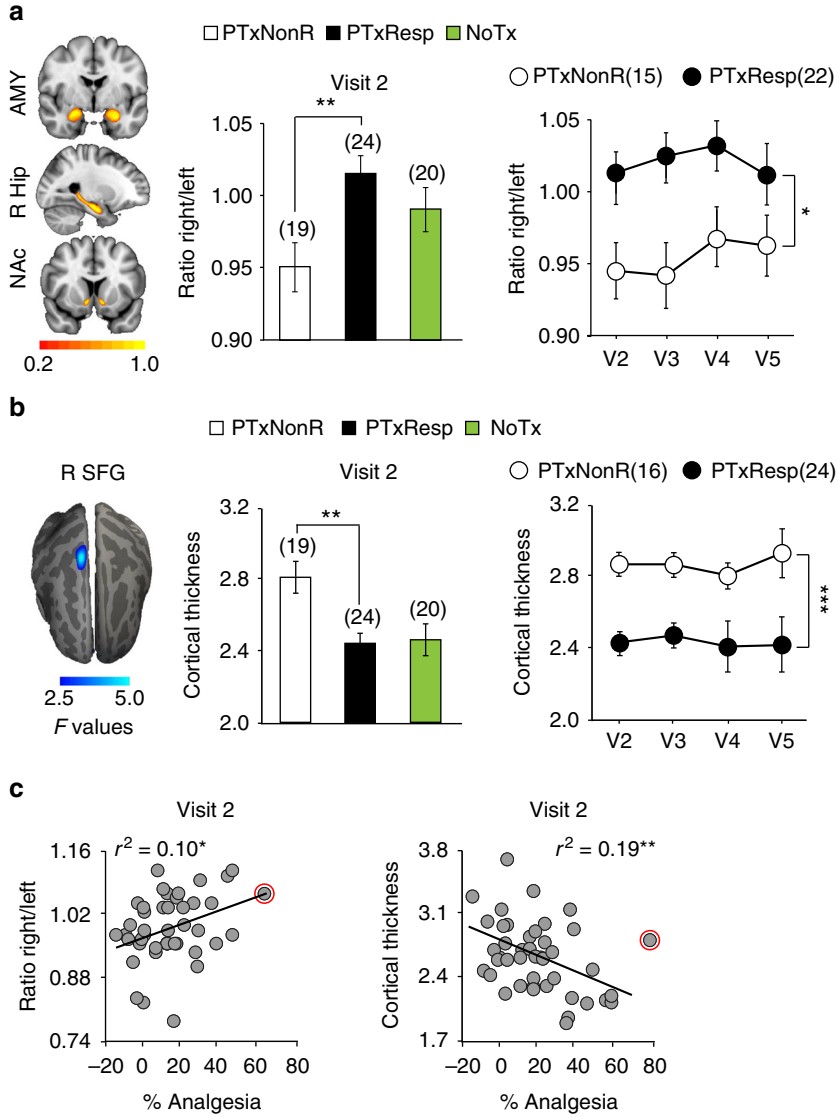

**Fig. 3** Placebo pill response is predetermined by subcortical limbic volume asymmetry and sensorimotor cortex thickness. **a** Heat maps display overlap of automated segmentation for nucleus accumbens (NAc), right hippocampus (R Hip), and amygdala (AMY) across all patients. The placebo pill responders displayed a rightward asymmetry in overall volume of these three subcortical limbic structures after controlling for peripheral gray matter volume, age, and sex at visit 2. The placebo group-dependent asymmetry was observed at all 4 visits/scans. **b** Whole-brain neocortical vertex-wise contrast indicated thicker sensorimotor, right superior frontal gyrus (R SFG), in PTxNonR. This effect was consistent across all 4 visits/scans. **c** These anatomical properties correlated with the magnitude of the response. All post hoc comparisons were Bonferroni corrected for three comparisons: *$p < 0.05$; **$p < 0.01$; ***$p < 0.001$. Error bars indicate SEM. The red circle marks an outlier on the magnitude of response and was excluded from the %analgesia correlation analyses

**Classifying placebo pill response using machine learning**. We used machine learning to determine whether placebo response could be predicted from brain imaging and questionnaires data collected prior to placebo treatment. We implemented a nested leave-one-out cross-validation (LOOCV) procedure where placebo outcome of each patient was predicted using an independent training sample. Within each $n$-1 patients training sample set, the model parameters were tuned using tenfolds cross-validations. The optimized model showing the least error was then applied to the left-out patient, repeated for every patient.

We initially used data from the questionnaires to classify the patients into response groups (binary variable approach). Within each training sample set, the scores of the normalized 38 subscales were used to build the support vector machine (SVM) classifier. SVM classification achieved an accuracy of 0.72 in classifying placebo pill response [95% CI, 0.56–0.85]. Sensitivity of this

approach was 0.73 (95% CI, 0.52–0.88), and specificity was 0.71 (95% CI, 0.44–0.90) (Fig. 5a–d).

We next trained a classifier using brain imaging data entering either brain anatomy or rsfMRI as predictors, but they failed at classifying PTxResp and PTxNonR above chance level (accuracies were <0.67, which corresponded to a $z$-score of 1.96). Moreover, adding features from rsfMRI or brain anatomy to the questionnaire data did not improve the classifier's accuracy (accuracy of 0.65).

**Predicting the magnitude of response using machine learning**. We next used Least Absolute Shrinkage and Selection Operator (LASSO) regression to predict the magnitude of placebo response (continuous variable approach), using a nested LOOCV procedure. The model was trained in $n$-1 patients in an inner loop using tenfolds cross-validations for tuning the LASSO

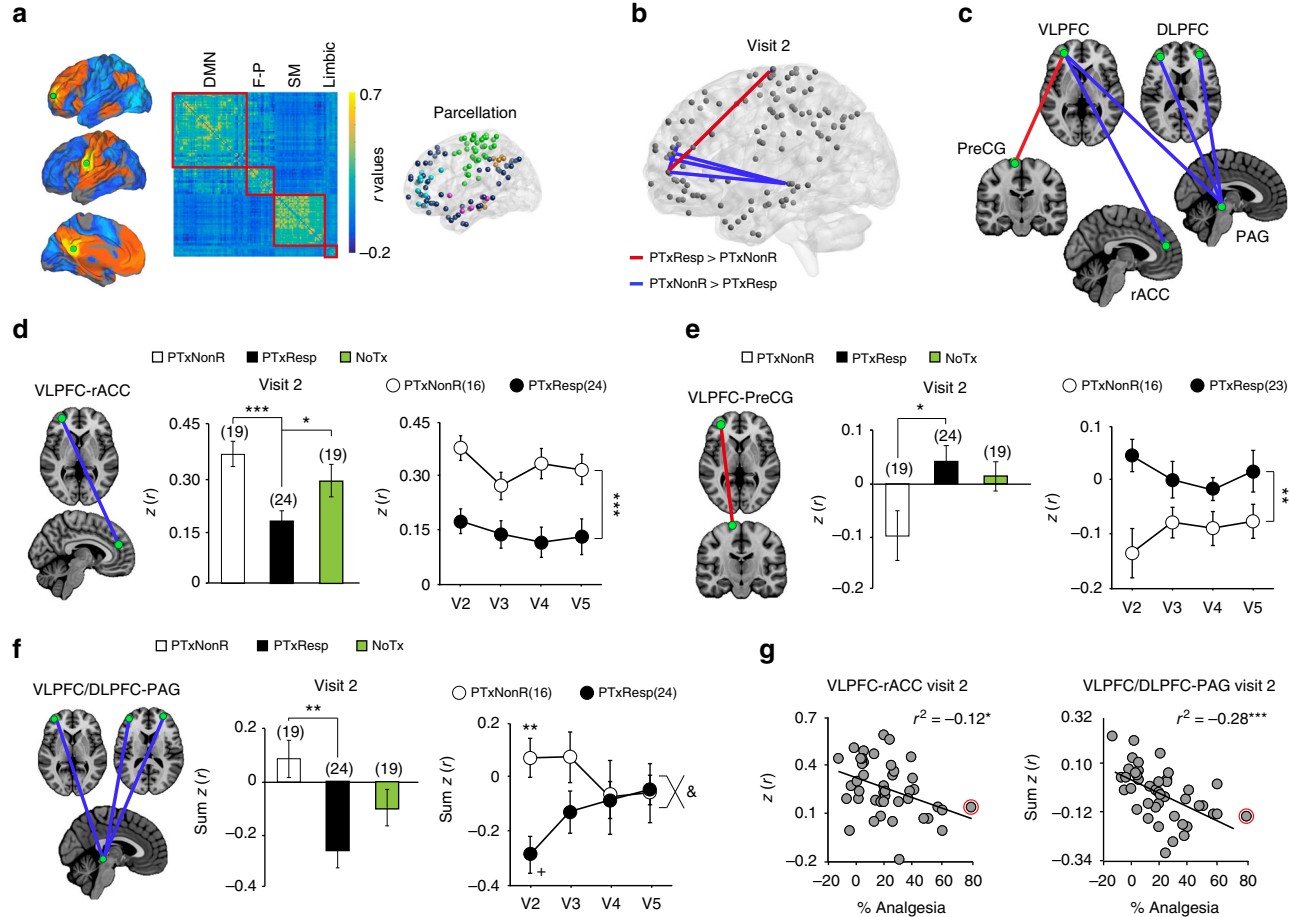

**Fig. 4** Placebo pills response identifies a lateral prefrontal functional network with invariant and transient components. **a** Previous results indicate that functional connectivity in four regions: middle frontal gyrus (MFG), rostral-anterior and posterior cingulate cortex (rACC, PCC), and sensorimotor (SM) cortices predict placebo response in OA patients[22]. Co-activation maps derived from these seeds (green) (1000 healthy subjects from http://neurosynth. org) were used to restrict brain networks of interest. The connectivity matrices of our data were restricted to communities overlapping with these networks of interest: default mode network (DMN), frontoparietal (F-P), sensorimotor (SM), and subcortical limbic. **b, c** A permutation test was performed on the weighted 7381 resting state connections between all possible pairs of nodes within the modules of interest (FDR-corrected $p < 0.05$). Stronger connectivity link Ventrolateral prefrontal cortex (VLPFC)-Precentral gyrus (PreCG) and weaker connectivity links VLPFC-rACC, and VLPFC/DLPFC-periaqueductal gray (PAG) identified placebo responders prior to treatment (visit 2). **d–f** VLPFC-PreCG and VLPFC-rACC connections differentiated placebo responders and remained constant in time while VLPFC/DLPFC-PAG connectivity differentiated placebo pill response transiently (interaction time with group trending $p = 0.09$). **g** Each of these brain parameters correlated with the magnitude of the response. All post hoc comparisons were Bonferroni corrected for three comparisons: $^{\&}p = 0.09$; $^{*}p < 0.05$; $^{**}p < 0.01$; $^{***}p < 0.001$; + within comparison V2 vs. V5: $p < 0.05$. Error bars indicate SEM. The red circle marks an outlier on the magnitude of response and was excluded from the %analgesia correlation analyses

parameters, and then tested in the unseen held out patient, repeated for every patient.

As with the previous approach, we initially predicted the magnitude of response using just the combination of questionnaire data (Fig. 5e, f). The scales contributing to the prediction of response magnitude were: (1) the Emotional Awareness, Noticing, and Not-Distract subscales of the MAIA Questionnaire (MAIA/e; MAIA/nd); (2) the Non-painful Situations subscale of the Pain Sensitivity Questionnaire (PSQ/np); (3) the Describing subscale of the Five Facets of Mindfulness Questionnaire (FFM/d); and (4) Openness from the NEO-FFI (Fig. 5g). The model was no longer able to predict the magnitude of response after removing these psychological parameters. This not only indicates their importance for response, but also reveals that neither the traditional psychological measures reported in healthy controls under placebo conditioning (e.g., neuroticism, extraversion, and optimism) nor any of the chronic pain-related personality traits usually linked to severity of symptoms (e.g., anxiety, catastrophizing, and fear of pain) contributed to the prediction.

Next, we sought to determine if rsfMRI collected prior to placebo pill ingestion could predict the magnitude of response. Within each $n$-1 patients training sample set, we performed feature selection to identify links correlating with the magnitude of response (robust regression, $p < 0.001$) prior to the LASSO regression. These connections were used to train a predictive model using tenfolds cross-validations for tuning the LASSO parameters, which was tested in the left-out patient (Fig. 5h, i). Because the number of features and weights differed between each loop, a "consensus" was generated by averaging the weights across the $n = 43$ loops to create a final single set of weights (Fig. 5j). The resultant network consisted of a combination of 19 weighted connections predicting the magnitude of placebo analgesia. The most frequently selected features included links connecting nodes located in the orbitofrontal cortex (OFC), the PreCG, the DLPFC, the PAG, the amygdala (AMY), the precuneus, and the supramarginal gyrus (SMG) (Fig. 5j). Because the functional connections were restricted between 122 preselected ROIs, we can't exclude the possibility that increasing

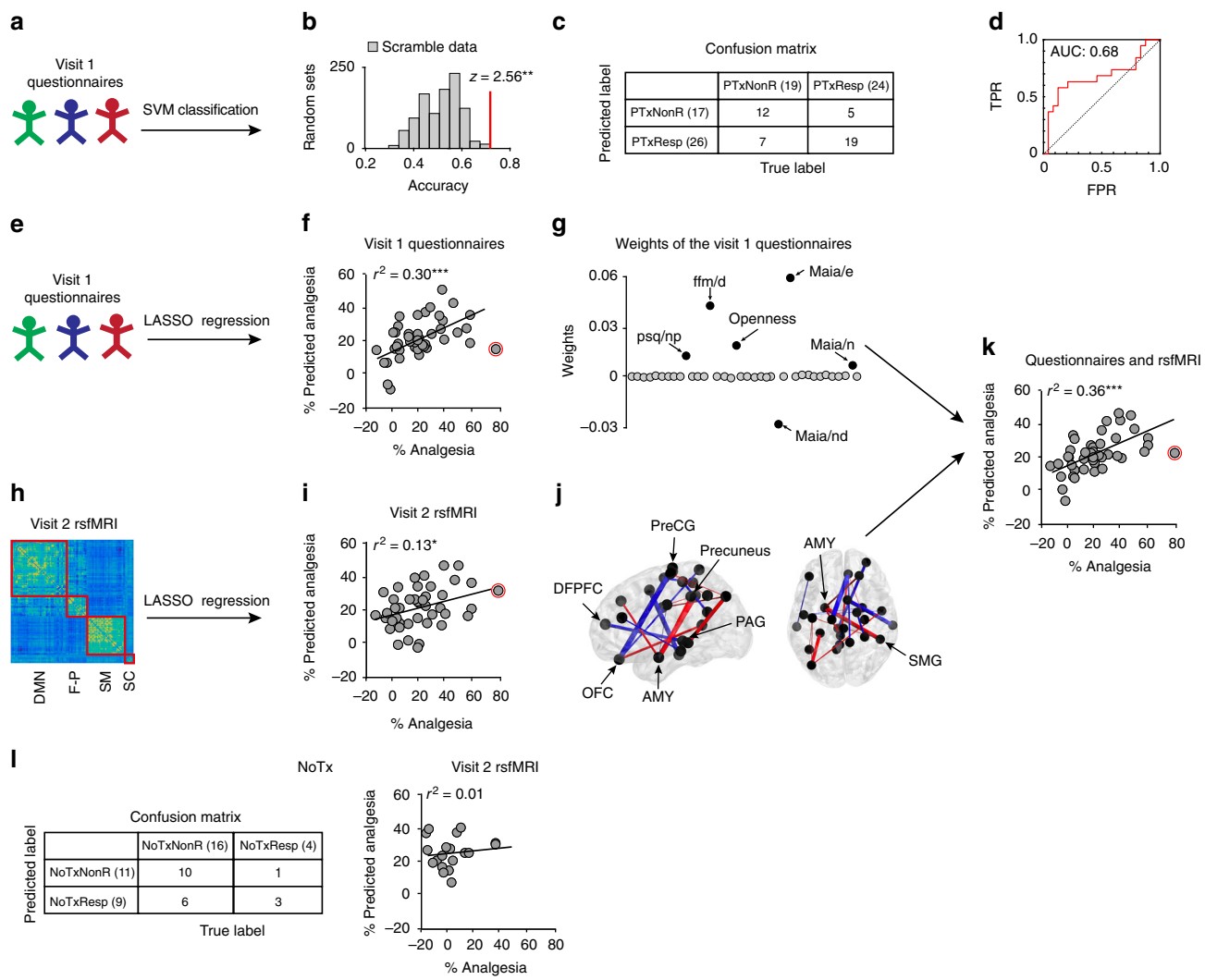

**Fig. 5** Machine learning classifies placebo response (PTxResp/PTxNonR) and predicts the magnitude of response. **a** A support vector machine (SVM) classifier was applied to the questionnaire data in a nested leave-one-out cross-validation (LOOCV) procedure. **b** The observed accuracy of this classification analysis is displayed against the null distribution generated using scrambled codes from 1000 randomized labels. **c**, **d** Classification performance was assessed based on the confusion matrix. The ROC curve shows specificity and sensitivity of the model. **e**, **f** LASSO regression was applied to the questionnaire data and predicted the magnitude of response in the left-out patients. **g** Averaged weights across the $n = 43$ loops of the cross-validation procedure show that six psychological factors predicted the magnitude of response. **h**, **i** Features selected from rsfMRI within each training sample predicted the magnitude of response in the left-out patients. **j** A single set of consensus weights was generated by averaging the weight of each feature across the $n = 43$ loops; this consensus set is projected back onto the brain for display. Nodes of the most frequently selected features (>0.84 of loops) are labeled on the figure **k**. Linear regression analysis indicated that the predicted magnitude of response from rsfMRI at V2 and from personality at V1 were independent contributors that jointly explained 36% of the variance in actual response to placebo treatment. **l** The classifier (**a**) and the regression model (**k**) failed to predict placebo response in patients of the NoTx group. The red circle marks the predicted magnitude of response for the outlier, which was excluded for assessing the error of the model. *$p < 0.05$; **$p < 0.01$; ***$p < 0.001$; AMY amygdala; DLPFC dorsolateral prefrontal cortex; OFC orbitofrontal cortex; PAG periaqueductal gray; PreCG precentral gyrus; SMG supramarginal gyrus

the number of connections between other ROIs, or using other metrics of brain function, may have provided a better prediction of placebo response.

Although this set of edges was predictive of magnitude of response prior to the first placebo treatment, the prediction did not generalize to rsfMRI data collected at other visits post treatment. This is likely due to the small number of connections included in our model, which are either changing in time as a consequence of learning and adjustment of expectations with the introduction of a placebo treatment or due to inherent variability of the measurements themselves.

Applying LASSO regression on features from brain anatomy generated a predictive model, which did not correlate with the actual magnitude of response. The model was therefore not considered significant and is not reported.

**Comparing models from brain imaging and psychological factors**. We tested whether predictive models from psychological factors and brain functions were independent or if they instead predicted redundant information. The predicted analgesia from personality factors was not correlated with the predicted analgesia from rsfMRI (Pearson correlation $r = 0.23$; $p = 0.15$). Furthermore, a linear regression entering the value of the predicted magnitude of response from rsfMRI and the value of the predicted analgesia from the questionnaires data revealed that both

models explained independent variance of the actual response, suggesting that they are complementary to one another (Fig. 5k). Thus, although the questionnaires were strong predictors of the magnitude of response, they should not be considered as proxy for the brain imaging data, and vice versa.

**The predictive models did not generalize to the NoTx arm**. We finally tested whether our models were specific for placebo pill analgesia or if they were predicting unspecific improvement of symptoms. We first used univariate statistics to compare NoTx-Resp with NoTxNonResp on the brain properties dissociating PTxResp from PTxNonR (Figs. 3 and 4). There were no differences on any of these parameters in the NoTx group suggesting that our results were specific for placebo response (Supplementary Figure 11). Second, we tested whether our multivariate classifier based on psychological factors (Fig. 5a) was capable of stratifying the patients randomized in the NoTx arm. The classifier accuracy was considered non-significant (0.65) and the difference between predicted response rate in the NoTxResp was not higher than in the NoTxNonR ($\chi^2 = 1.82$ $p = 0.18$; Fig. 5i). Finally, we applied the regression model predicting the magnitude of response (Fig. 5k) to patients in the NoTx group, and here again, the model was inaccurate in the NoTx (Fig. 5i). In this case, the coefficient of correlation between the predicted and the actual analgesia was stronger in the PTx group (Fig. 5i) than in the NoTx group (Fig. 5i; $z$-test = 2.00, $p = 0.046$), indicating that the model was more predictive in individuals ingesting placebo pills.

## Discussion

This is the first brain imaging RCT specifically designed to study chronic pain patients receiving placebo pills compared to a no treatment arm. Daily pain ratings from a smart phone revealed that patients receiving placebo pills showed stronger pain reduction and a higher response rate compared to patients in the no treatment arm, indicating that placebo pills successfully induced analgesia that could not be explained by the natural history of the patient or the mere exposure to the study. Our results show a multiplicity of biological systems, partially overlapping with complex inter-relationships, underlying placebo pill response. The identified systems seem to encompass brain properties known to be involved in chronic pain maintenance or in the transition to chronic pain (e.g., SM cortex thickness/volume[37]), mechanisms described for placebo response in healthy subjects experiencing acute pain[38] (e.g., involvement of OFC, DLPFC, rACC, and PAG[39,40]), as well as novel systems associated with placebo pill response in chronic pain (e.g., subcortical volume asymmetry, emotional awareness, and functional coupling of the amygdala, SMG, precuneus, and PreCG). Machine learning applied to questionnaire and brain imaging data could, only in part, classify placebo response and predict the individuals' magnitude of placebo response. Given the moderate to large effect size of RCT placebo effect (also observed here), our results imply that gaining a better understanding of placebo pill response has important clinical utility. Our results demonstrate the psychological, functional, and anatomical determinants of the placebo response and suggest that once patients begin a placebo treatment, their individual pain relief may be predicted in the context of a RCT.

One of the main behavioral findings was that the treatment outcomes exhibited a high level of dimensionality that has not been specifically investigated or accounted for in other clinical or basic science studies regarding modulation/perception of pain. The no treatment arm documenting the natural history of the patient allowed us to show that placebo treatment impacted a particular dimension of chronic pain (pain intensity) without

changing the trajectory of pain quality (which improved in time regardless of placebo pill ingestion and/or placebo response). Even various measures of pain intensities—daily ratings, memory, and NRS—provided slightly different information about the extent of analgesia and pain fluctuations. This highlights the importance of examining a multiplicity of pain-related outcomes as the analgesic properties of any given treatment may not be constrained to a single dimension of the pain experience, and the measures used to capture analgesia may be differentially influenced by different factors. Most current RCTs assessing new treatments are designed with a single primary outcome representing a "gold standard", thus likely missing out on the complexity of treatment effects that chronic pain may exhibit. Our behavioral results stress the importance of moving away from a single, cross-sectional pain measurement, and demonstrate that distinct dimensions respond differentially to placebo pill ingestion, impacting mainly perceived magnitude but not its qualities. It is however possible that placebo might only impact the intensity of pain while a successful active pharmacological treatment would improve both intensity and qualities. This remains an open but important area of inquiry.

Our study design included two washout periods in order to determine stability and within-subject re-occurrence of response. The use of EMAs allowed us to determine that placebo analgesia started on the first day of treatment, but the return to baseline levels of pain started only several days after washout. Thus, the washout periods were proven too short to test the re-occurrence of response. Instead, our data showed a carryover effect of the placebo response after discontinuation of treatment.

Associating a psychological profile with placebo pill response in CBP departs from the literature regarding placebo in healthy subjects. None of the often-cited personality traits in placebo literature[35,41–43]—optimism, anxiety, extraversion, neuroticism—successfully differentiated placebo responders from non-responders in our chronic pain patients. In our patients, placebo pill response was driven primarily by a combination of a greater openness to experience, increased emotional awareness, decreased distraction about pain and discomfort, augmented capabilities in describing inner experiences, and higher sensitivity to non-painful situations. Our results reveal that placebo response can be predicted from an ability to recognize subtle cues in the body regarding emotional and physical well-being, to remain attentive to these cues and emotions by not ignoring or suppressing them, and to choose to accept these states as opposed to becoming worried or burdened by them. These factors of personality were able to differentiate PTxResp and PTxNonR as well as predict the magnitude of placebo response in new patients. These results are critical, as questionnaires are easy to administer and may be sufficient to predict placebo pill response in chronic pain.

In healthy individuals, the placebo response recruits endogenous pain pathways acting upon the opioid system to regulate descending inhibition from the rACC[9] through the PAG[25], a mechanism that can be reversed by naloxone[44]. Besides these anti-nociceptive circuits, the placebo effect is also dependent on subcortical circuitry involved in reward/aversion prediction error, as well as higher-order frontal mechanisms (including the DLPFC and the OFC) involved in context generation, expectation of treatment outcomes, and emotional appraisal of events[39]. As such, levels of activation in the DLPFC and the OFC are believed to represent the strongest predictors of experimental placebo response in healthy controls[45]. The present results indicate that these systems are also part of the placebo pill response in CBP patients, although direct correspondences between functional networks and regional activity remain uncertain. The coupling of the DLPFC and rACC with anti-nociceptive circuitry is also

consistent with our previous observation that these regions were predictive of placebo response in OA patients[22]. Therefore, there are close correspondences in the mechanisms underlying placebo pill response across different types of pain (chronic back pain observed in this study, chronic knee pain[22], and acute experimental pain) and in different settings (RCT vs. laboratory).

For the placebo response-related network differentiating PTxResp from PTxNonR, the DLPFC-PreCG, and DLPFC-rACC links persisted in time and across visits. However, the initial VLPFC/DLPFC-PAG connections differentiating PTxResp and PTxNonR dissipated, likely due to this network being more dependent on learned expectations and thus requiring novel reinforcing experiences to become reactivated. As the VLPFC/ DLPFC-PAG includes commonly described brain regions associated with placebo response in healthy subjects, its dissipation with exposure is consistent with evidence pointing to the failure of placebo re-exposure in healthy subjects in replicating the original placebo response[12]. Brain networks predicting the magnitude of response in a fully cross-validated procedure further reveal the contribution of limbic circuitry, TPJ, and prefrontal connectivity (DLPFC-PAG and OFC-PreCG) for placebo pill response. Because the model was not stable in the post-treatment visits, its capacity for predicting magnitude of response in a new set of chronic pain patients remains to be determined in further studies. The procedure was nevertheless informative regarding the neurophysiological contributors to the placebo response. Our original experimental design included two washout periods (W1, W2) specifically designed to monitor perturbations in functional features, like VLPFC/DLPFC-PAG, and to observe whether following washout, they would continue to track placebo pill responders. Unfortunately, the pain trajectories showed sustained effects of placebo analgesia during washout periods that were proven too short to be informative regarding within subject replication of placebo effects.

Several pitfalls have been raised when trying to predict complex behaviors like the placebo response[46]. Here, many of these potential confounds were accounted for by incorporating novel methodological strategies such as: including a no-treatment arm documenting the natural history of the patients, using smart phone technology accounting for natural fluctuations of pain outside of the clinical setting, collecting multiple pain outcome measures, performing analyses blindly using one real code and two scrambled codes to minimize bias, and utilizing machine learning methods to estimate predictability in a fully cross-validated procedure.

Some have argued that clinical trials do not provide an appropriate context to study the psychobiology of placebo because they are contaminated by uncontrollable confounds and that instead placebo should be studied in a controlled environment, such as a laboratory[47]. Despite the complexity of the phenomenon, our results challenge this assumption as placebo response could be partially predicted in chronic pain patients. Precisely, machine learning applied to psychological factors or functional connectivity showed that magnitude of response was predictable. The predictions from both models were not correlated to one another and they were independent predictors of analgesia. Importantly, the joint prediction from both models was more accurate in the PTx arm compared to the NoTx, suggesting a certain level of specificity unique to placebo analgesia. Together, our results contribute to the placebo literature by demonstrating the existence of psychological and neurobiological principles determining the placebo response in RCTs.

## Methods
**Participants (study population).** The total duration of the study lasted ~15 months. The first patient was seen on 11/06/2014, and the last patient was seen on 02/04/2016. During that time, 129 participants with chronic low back pain (CBP) were initially recruited from the general population and clinical referrals via hospital databases and advertising in the community. Patients were assessed for general eligibility via self-report using a screening intake form that the lab has used for years in other studies; this screening interview covered co-morbid health and psychological conditions, MRI safety, concomitant medication dosages and indications, pain levels/location/duration, current and previous illicit drug/alcohol use, litigation status, and overall willingness to be in a research study. To meet inclusion criteria, individuals had to be 18 years or older with a history of lower back pain for at least 6 months. This pain should have been neuropathic (radiculopathy confirmed by physical examination was required), with no evidence of additional co-morbid chronic pain, neurological, or psychiatric conditions. Individuals had to agree to stop any concomitant pain medications and had to be able to use a smartphone or computer to monitor pain twice a day. Additionally, the enrolled patients had to report a pain level of at least 5/10 during the screening interview, and their averaged pain level from the smartphone app needed to be higher than 4/10 during the baseline rating period (explained below) before they were randomized into a treatment group. Finally, for safety precautions, clinical measurements taken at visit 1 were required to be within the pre-specified healthy range (as determined by the standards utilized by Northwestern University Feinberg School of Medicine Laboratory Services Department) and all participants passed the MRI safety screening requirements at each scanning visit. Informed consent was obtained from all participants on their first visit.

Supplementary Figure 1 consort diagram illustrates the flow of patients through the clinical trial. From the initial 129 chronic back pain (CBP) patients recruited in the study, 4 individuals were assessed for eligibility but met exclusion criteria before consenting. Of the enrolled 125 patients, 43 failed to meet the inclusion criteria at visit 1 or during the 2-week baseline period between visits 1 and 2. The remaining 82 patients were randomized into one of three groups: no treatment (n = 25); active treatment (n = 5); or placebo treatment (n = 57). Of the no treatment group, n = 5 were either discontinued from the study or lost to follow-up; of the placebo treatment group, n = 11 were either discontinued or lost to follow-up, with an additional 2 participants being excluded from final analysis due to having average baseline pain rating values below 4/10. The inclusion of an active treatment group was used to ensure that the double blind for placebo treatment was maintained for the duration of the study and that no deception took place during the informed consent process (i.e., we could truthfully tell patients that they *may receive* a placebo or they may receive an active treatment). Therefore, the 5 participants randomized in the active treatment group were not analyzed. The final sample size included 20 CBP patients randomized to the no treatment group and 43 CBP patients randomized to the placebo treatment group; demographics for these individuals can be found in Supplementary Table 1. Participants were compensated $50 for each visit completed, and they were reimbursed up to $20 for travel and parking expenses if applicable.

The number of patients recruited was based both on the power analysis and on our previous experience with attrition rates in studies with similar patient populations; the final sample size based on the following effect size estimates were approved by the sponsor (NCCIH) prior to starting the study. We estimated our statistical power using the Cohen's *d* effect sizes for differences in pain with a 2-week placebo treatment; this was based on preliminary results from a different study that was ongoing at the time this RCT was being planned. For responders, we anticipated a mean decrease of 30 units on a 0–100 scale, with an estimated standard deviation of 15; this results in an effect size estimate of 2.0. In non-responders, the mean decrease in pain was anticipated to be negligible and we did not expect to have enough power to detect this. Power analyses performed in G*Power, version 3.1.3, indicated that we would have ample power—even with a conservative estimated effect size of $d = 1.0$, power would be 80% for a sample size of $n = 17$ per group, which would also permit detection of interaction effects. In addition, it ensured adequate sample sizes even assuming some attrition in each group. For brain imaging contrasts, we thought that 20 per group should be adequate given preliminary fMRI results and earlier studies; for T1 results, our earlier studies indicated that 20/group for within-subject contrasts would have been adequate but possibly just at the limit for whole-brain contrasts to detect between-group differences. Therefore, we ended up aiming for a sample size of $n = 20$ per group to achieve effect sizes of about 1.0 (i.e., $n = 20$ placebo responders, 20 placebo non-responders, and 20 no treatment). Since, we did not classify patients as "responders" or "non-responders" until after the study, we had no way of knowing exact stratification of groups during the study, which resulted in slightly uneven group sizes.

**Study design and procedures.** This study was conducted in the setting of a clinical randomized controlled trial specifically designed for assessing the placebo response (registered at https://www.clinicaltrials.gov/ct2/show/NCT02013427). The study consisted of 6 visits spread over ~8 weeks (Fig. 1a), including a baseline monitoring/screening period and two treatment periods, each followed by a washout period. The design was setup to track placebo response in time and to test the likelihood of response to multiple administrations of placebo treatment in order to optimize accuracy in the identification placebo response. The overall protocol included four scanning sessions collected before and after each treatment period.

**Randomization**. The randomization scheme was performed using 2 kinds of blocks, each with 8 patients; the first block assigned 5 patients to placebo and 3 to no treatment, and the second block assigned 5 patients to placebo, 2 to no treatment, and 1 to active treatment. Each patient ID was randomly attached to a randomization code. The initial randomization included codes for the first 80 patients. It was followed by a second randomization of 50 additional codes about 6 months later. For those assigned to either of the treatment groups, the allocation was performed in a double-blinded fashion: a biostatistician performed the randomization; drugs were ordered and re-encapsulated by the Northwestern research pharmacy and bottled by designated lab members; a member of the Northwestern University Clinical and Translational Sciences (NUCATS) institute matched the appropriate treatment drug with patients' randomization code; critically, only this NUCATS member had access to the document linking patient IDs to randomization IDs, and this linking information was only made available if a serious adverse event (SAE) occurred (which did not happen). After these procedures, study coordinators picked up the blinded agent from NUCATS for storage and dispensing; all drugs were stored at room temperature in a locked cabinet within the lab and monitored daily for temperature changes, bottle counts, and expiration dates. The double blind for treatment groups was maintained by the identical encapsulation of the study agent—blue pills were either Naproxen (500 mg) or placebo (lactose) and bi-colored pills were either Esomeprazole (20 mg) or placebo.

Each person assigned to treatment received a mixture of blue and bi-colored pills. This way, neither the participants nor the researchers knew which treatment the participant had received. For those assigned to the no-treatment group, no blind was maintained, as both study staff and participants knew that they were not receiving the study agent. Once ~50% of all participants had been entered into the study, a preliminary analysis of the electronic pain rating data was completed in order to confirm that there were participants who were experiencing a diminution in pain (no action was taken).

**Description of visits**. Visit 1: Participants were screened for eligibility and consented on visit 1. Following informed consent, a blood sample was drawn (for a comprehensive chemistry panel, a complete blood count, and a pregnancy test if applicable), vital signs were taken (blood pressure, heart rate, respiration rate, height, and weight), and a medical professional completed a physical examination and took a comprehensive pain history. Participants were then asked to complete a battery of 29 questionnaires regarding basic demographics, pain, mood, and personality (Supplementary Table 4). These self-reported measures were collected online via REDCap (Research Electronic Data Capture version 6.5.16, © Vanderbilt University) through a survey link sent to the participant's email address (or a backup study email if they did not have an email account). Once submitted, questionnaire answers were finalized in the database and were rendered un-editable by both participants and study staff. To best avoid questionnaire fatigue due to the number of questionnaires administered, participants were allowed to take breaks and walk around the testing room, although they were required to complete all questionnaires at the designated visit. Any remaining information, including clinical data collected at the visit, were entered manually into the database by study staff. The relevant information was verified via double-data entry by different staff members at a later time. At the end of visit 1, participants were asked to stop all medication they were taking for controlling their pain. Rescue medication in the form of acetaminophen tablets (500 mg each) was provided as a controlled replacement to be used at any time in the study if their pain became too intense. At this time, participants were also trained on how to use our electronic pain rating application on either the phone or the computer (explained below; Supplementary Figure 2); if participants did not have access to either, they were provided with a smartphone and data plan for the duration of the study. The baseline rating period started at the end of this visit and lasted until they came back for their second visit approximately two weeks later.

Visit 2: If patients' pain ratings and blood lab results met inclusion criteria, they returned for visit 2 where they completed a 35-min brain imaging session that collected a T1-weighted image, 2 resting state scans, and 2 diffusion tensor imaging (DTI) scans (details are presented below). Following the imaging protocol, the patients completed another battery of questionnaires, a subset of which were repeated from the first visit to track longitudinal changes in pain. They were asked whether they had experienced any changes in health status since the last visit. Additionally, they were asked to verbally recall their average pain levels over the previous 2 weeks, and over the preceding week. This self-reported recalled pain was referred to as "pain memory" and was used as an alternative outcome measure of pain levels.

At the end of this visit, participants were randomized into one of three groups: no-treatment, placebo treatment (lactose) or active treatment (the standard of care, which was a combination of Naproxen, 500 mg bid, and Esomeprazole, 20 mg bid). Participants in the treatment groups were instructed to take a blue pill with a bi-colored pill in the morning and again at night with plenty of water, and they were asked to record this in their electronic rating app. Note that study staff never informed participants about the odds for receiving active versus placebo treatment —this is important, as the goal was to have participant's own baseline expectations influence whether or not they responded to the placebo treatment. Both treatment and no treatment groups continued to receive rescue medication to use if needed, and all participants were asked to continue rating their pain twice a day until visit 3. The duration of this first treatment period was ~2 weeks long.

Visit 3: Patients returned at visit 3 and were queried about their memory of their pain, any changes in health since the last visit, and rescue medication usage. If on treatment, patients were asked to report any side effects experienced and bring back any unused medication so that study staff could calculate their treatment compliance. Participants underwent another scanning session that was identical to the one completed at visit 2 and completed another set of questionnaires with some repeated from the previous visit. At the end of visit 3, individuals assigned to the treatment group were told that the study agent would be temporarily discontinued until their next visit so that the effects of the agent could "washout" of their system. Again, all participants were given rescue medication to use if needed and were asked to continue using their app twice a day until the next visit. This first washout period was ~1 week long.

Visit 4: Patients returned at visit 4, where all measurements and procedures from visit 2 were repeated identically, including the scanning session and questionnaires. Again, they were queried about their pain memory, rescue medication usage, and changes in health. The study agent was reintroduced to those individuals allocated to one of the treatment groups according to the same regimen described above (treatment assignment was kept the same within subjects, as this was not a cross-over study design). During the consent process and treatment administration, patients were informed that they were receiving the same treatment as the one administered during the first treatment period so that expectations were not explicitly manipulated in anyway. All participants were given rescue medication and asked to rate their pain and mood twice a day, as with previous visits. Like the first treatment period, the second treatment period was also ~2 weeks in length.

Visit 5: Following this period, participants returned for visit 5, where all measurements and procedures from visit 3 were repeated identically. Patients underwent the same scanning procedures as on visits 2–4. Finally, patients filled out a series of questionnaires about their pain, some of which were repeated from the last visits. As before, those participants allocated to a study agent had their treatment discontinued for a second washout period, which was also ~1-week long. Participants continued to use their electronic app twice daily and were given rescue medication if needed.

Visit 6: Patients returned for the last visit during which they were again queried about their pain memory, changes to health, and rescue medication usage. During this visit, the patients completed a semi-structured, open-ended exit interview with a designated staff member. They were asked more detailed questions about their pain and medical history, quality of life, overall mood, and time in the study. Participants finished with a final battery of questionnaires and were asked to return study smartphones, if applicable. There were no scanning procedures on this visit. Any ratings submitted for the duration of the study were totaled, and in addition to their visit compensation, participants received their compensation for the electronic app at this time.

**Monitoring pain intensity with phone app**. Each patient's pain was monitored electronically using an application designed specifically for the study (Fig. 1; Supplementary Figure 2). This app was used to track patients' pain over time and to query them on their medication usage; it could be accessed using either a smartphone or a website link on a computer. The app had a VAS scale with sliding bars: it asked participants to rate their current pain level from 0 (no pain) to 10 (worst imaginable). The app also included fields to indicate the participant's assigned ID number, query if participants had taken any rescue medication at that time, and ask if they had taken the study medication. There was a comments section that they could use to describe their pain, mood, or medication usage if they chose. Participants were instructed to use the app twice a day, once in the morning and once at night. To encourage compliance, participants were compensated $0.25 for each rating they submitted, up to $0.50/day. This additional payment was given to them on the last visit of the trial. Submitted ratings were immediately sent to a secure server and both date- and time-stamped. Rating compliance was assessed by a separate program, which monitored whether the list of currently enrolled patients had provided the necessary ratings during the previous day. In the case that a patient omitted a rating, staff were alerted via an email. If patients missed more than 2 consecutive ratings (~24 h-worth), a member of the study team contacted them to remind them to use the app. Two patients were discontinued from the study because they did not comply with the daily rating requirements despite repeated contact from the study team.

To verify that pain levels remained within the inclusion criteria specified above, all participants' ratings were closely monitored for the first 2 weeks of the study as part of a run-in/baseline pain period. Individuals not meeting this level were deemed ineligible and did not continue in the study ($n = 16$ screen failures). It was later noticed that 3 additional participants had met this exclusion criteria but accidentally continued in the study. One person was assigned to no-treatment and was discontinued as a protocol deviation before study completion; the other two individuals finished the study in the placebo treatment group but were not included in the analysis.

**Preprocessing of phone app ratings**. App rating data from all participants were pre-processed as follows. Although participants were asked to rate twice a day (and only compensated for this amount), many participants exceeded this number of app ratings in 24 h due to over-compliance, reassessment of their pain, and/or

cellular service problems. If pain ratings were entered within 30 min of each other, only the last rating was kept and taken as indicative of the participant's final assessment of their pain levels at that time. Any additional ratings outside of this 30 min window were not considered duplicates and were kept as valid entries. Beside this cleaning process, no other changes were made to the ratings. In the instances where participants missed ratings, no attempts were made to interpolate or re-sample the data so that the temporal aspects of the ratings were left intact. The overall compliance of the phone ratings is reported for each group in Supplementary Table 2.

**Defining placebo response**. This smartphone technology permitted us to track fluctuation in pain levels throughout the study. Figure 1b displays the time series generated using the pain ratings entered by participants PL001 and PL039. In this study, we assessed two different components of the placebo response: the response or absence of response as well as the magnitude of the response. To best make use of the daily rating data, we initially developed a new classification scheme of responders versus non-responders that accounts for the within-subject variability of pain levels. Each patient was classified based on a permutation test between the pain ratings acquired during his baseline rating period (visit 1 to visit 2) and the pain ratings acquired during his treatment periods (either baseline versus treatment 1, or baseline versus treatment 2). The null hypothesis was generated by randomly resampling 10,000 times the distribution of pain ratings, which provides a large set of possible $t$-values obtained from the rearrangement of the pain ratings. The overall $t$-value obtained between baseline and treatment was used to determine if the null hypothesis could be rejected ($p < 0.05$) for each of the treatment periods. In the cases where the null hypothesis could not be rejected for either of the treatment periods, the patient would be stratified as a "Non-Responder". Alternatively, the patient would be stratified as a "Responder" if there was a significant diminution in the pain ratings. The main advantages of using a permutation test is that it takes into consideration the variability across pain ratings during the baseline and treatment periods and it represents a statistically defined cutoff point for response (unlike cutoff points arbitrarily defined by a percentage change in pain). Correcting for autocorrelation in the time series of pain ratings did not change stratification (Supplementary Figure 12).

Because group stratification may have dampened individual response to placebo treatment, we secondly studied the magnitude of response by subtracting the averaged pain ratings entered during the baseline period with the averaged pain ratings entered during the last week of each treatment period separately. The magnitude of analgesia was defined as the highest difference between baseline and the 2 treatment periods. This provides a different facet of placebo response: the placebo response identifies significant improvement of symptoms (a small but constant improvement of symptoms may have stratified a patient as a placebo responder) while the % analgesia rather represents a continuous measure determining the importance of the response.

Effect sizes were calculated as the difference between the analgesia in the PTx group and analgesia in the NoTx group, divided by the standard deviation of all the data.

**Secondary pain outcomes**. The primary pain outcome measured with the phone app was compared with five additional secondary outcome measures of pain level. The numerical rating scale (NRS) and the memory of pain were reported as the two other primary pain outcomes relying on numerical scales. Their correspondence with the phone app is presented in Fig. 1i. The NRS represents the traditional standard pain measurement usually used in clinical trials assessing pain levels of participants for both placebo-controlled trials (compared against an active medication) and placebo-only trials (where the placebo effect is being manipulated)[28]. The memory of pain represents one of the standard pain assessments used by physicians in clinical practice and has been shown to correlate well with daily pain diaries in previous studies[48]. Other pain outcomes were collected using the McGill pain questionnaire (MPQ), affective and sensory scales, and the pain detect, which have been widely used in both randomized clinical trials and research labs, although their utilization in placebo-only trials remains minimal[49]. The neuropathic pain scale (NPS) was initially administered but not included as a main pain outcome since we aim to dissociate measurements of intensity from qualities of pain, while the NPS represents a combination of intensity and qualities of pain.

**Blinding of the analysis**. Given the recent issues regarding a lack of reproducibility in scientific findings[50], and the importance of transparency in data analysis, we followed recommendations by MacCoun and Pearlmutter[51] and employed cell scrambling to further blind our data and minimize bias. See Supplementary Table 6. For all endpoints, a lab member not involved in analyses was selected to organize data files and spreadsheets for processing and statistical analyses of the data. This person first renamed all the data files in order to ensure that analysts were blinded to each participant's unique ID, and to minimize bias from previous interactions with patients during data collection. Next, all analyses were performed with 3 randomized codes (which we refer to here as "classifiers") for each condition, with only one of them being the proper classification of placebo treatment responders, non-responders, and no treatment responders and non-responders. We refer to this as "triple blinding" because analyzers were blind to participant ID,

participant treatment, and correct participant group classification. The selected lab member did this blinding prior to any analyses, with the exception of the pain ratings from the app, which were used to stratify patients from the outset. As a result, each analysis was done three different times in an unbiased manner. Importantly, the three lab members who contributed to the analyses were not informed that they were provided different classifiers to make sure they could not collaborate to figure out which one was the real code. The results were presented in a public lab meeting where the lab member un-blinded the analyzers to the data to confirm which results were true. Although, we refer to these 3 classifiers throughout the paper, we only present the outcomes and data from the correctly classified group in each instance. Results from the 2 false classifiers are presented where applicable in Supplementary material for the purpose of comparison. This procedure aims to decrease uncontrolled bias during data analyses and to enhance the reproducibility of results.

**Brain imaging protocol and data analysis**. Brain imaging data were acquired with a Siemens Magnetom Prisma 3 Tesla. The entire procedure was completed in about 35 min, but an extra 25 min was allocated to install the patients in a comfortable position to keep their back pain at a minimum, and to re-acquire images if the data were contaminated by head motion.

High-resolution T1-weighted brain images were collected using integrated parallel imaging techniques (PAT; GRAPPA) representing receiver coil-based data acceleration methods. The acquisition parameters were: isometric voxel size = 1 × 1 × 1 mm, TR = 2300 ms, TE = 2.40 ms, flip angle = 9°, acceleration factor of 2, base resolution 256, slices = 176, and field of view (FoV) = 256 mm. The encoding directions were from anterior to posterior, and the time of acquisition was 5 min 21 s.

Blood oxygen level-dependent (BOLD) contrast-sensitive T2*-weighted multiband accelerated echo-planar-images were acquired for resting-state fMRI scans. Multiband slice acceleration imaging acquires multiple slices simultaneously, which permits denser temporal sampling of fluctuations and improves the detection sensitivity to signal fluctuation. The acquisition parameters were: TR = 555 ms, TE = 22.00 ms, flip angle = 47°, base resolution = 104, 64 slices with a multiband acceleration factor of 8 (8 × 8 simultaneously acquired slices) with interleaved ordering. High-spatial resolution was obtained using isomorphic voxels of 2 × 2 × 2 mm, and signal-to-noise ratio was optimized by setting the field of view (FoV) to 208 mm. Phase encoding direction was from posterior to anterior. The time of acquisition lasted 10 min 24 s, during which 1110 volumes were collected. Patients were instructed to keep their eyes open and to remain as still as possible during acquisition. The procedure was repeated two times.

**Preprocessing of functional images**. The pre-processing was performed using FMRIB Software Library (FSL) and in-house software. The first 120 volumes of each functional data set were removed in order to allow for magnetic field stabilization. The decision to remove this number of volumes was taken arbitrarily (it was not motivated upon examination of data) and we explored no other option. This left a total of 990 volumes for functional connectivity analyses. The effect of intermediate to large motion was initially removed using fsl_motion_outliers. Time series of BOLD signal were filtered with a Butterworth band-pass filter (0.008 Hz < $f$ < 0.1 Hz) and a non-linear spatial filter (using SUSAN tool from FSL; FWHM = 5 mm). Following this, we regressed the six parameters obtained by rigid body correction of head motion, global signal averaged overall voxels of the brain, white matter signal averaged overall voxels of eroded white matter region, and ventricular signal averaged overall voxels of eroded ventricle region. These nine vectors were filtered with the Butterworth band-pass filter before being regressed from the time series. Finally, noise reduction was completed with Multivariate Exploratory Linear Optimized Decomposition into Independent Components (MELODIC tool in FSL) that identified components in the time series that were most likely not representing neuronal activity. Components representing motion artefact were identified if a ratio between activated edge (one voxel) and all activated regions on a spatial component was >0.45, or if ratio between activated white matter and ventricle and whole-brain white matter and ventricles was >0.35. Moreover, noisy components were identified if the ratio between high frequency (0.05–0.1) and low frequency (0.008–0.05) was >1. This ICA regression process was kept very conservative so that only components obviously related to motion or noise were removed.

The functional image registration was optimized according to a two-step procedure. All volumes of the functional images were averaged within each patient to generate a contrast image representative of the 990 volumes. This image was then linearly registered to the MNI template and averaged across patients to generate a common template specific to our CBP patients. Finally, all pre-processed functional images were non-linearly registered to this common template using FNIRT tool from FSL. The registered brains were visually inspected to ensure optimal registration.

On average, relative head motion was relatively low (mean frame displacement (FD) = 0.11; std 0.07 for the first rsfMRI run, and mean FD = 0.11; std 0.09 for the second run. Importantly, there were no group differences between the mean relative frame displacement for either the first run ($F_{(2,60)} = 0.24$; $p = 0.79$) or the second run ($F_{(2,59)} = 0.66$; $p = 0.52$).

**Parcellation scheme**. The brain was divided into 264 spherical ROIs (5-mm radius) located at coordinates showing reliable activity across a set of tasks and from the center of gravity of cortical patches constructed from resting state functional connectivity[52] (Supplementary Figure 9a). Because subcortical limbic structures are believed to play a role in placebo response, 5-mm radius ROIs were manually added in bilateral amygdala, anterior hippocampus, posterior hippocampus, and NAc (Supplementary Figure 9b). Linear Pearson correlations were performed on time courses extracted and averaged within each brain parcel. Given a collection of 272 parcels, time courses are extracted to calculate a $272 \times 272$ correlation matrix. These matrices allowed for the construction of weighted brain networks, where nodes represent brain regions and links represent weighted connections from Pearson correlations between any given set of these regions.

One patient from the no treatment arm was excluded from all rsfMRI analyses because of aberrant values in the correlation matrix (values were above 20 std from the mean). This subject was a priori rejected during initial quality check and was never included in any of the analyses.

**Community detection analyses**. We used the Louvain algorithm integrated in the Brain Connectivity Toolbox (BCT; https://sites.google.com/a/brain-connectivity-toolbox.net/bct/)[53] to determine consistent community structures across a large number of network partitions[54]. For each subject, the individual community structure was initially constructed from 100 repetitions of the same network. The group community was then constructed from $100 \times 63$ patients, generating a total of 6300 networks. The final community structure was created by thresholding the averaged within-module connectivity likelihood matrix at 0.5, meaning that if the likelihood for two nodes belonging to the same module was above 50%, they were considered in the same module. This permitted us to identify six separate communities, including the four communities of interest (Fig. 2a).

**Identifying communities of interest**. We used localizers from an independent data set consisting of osteoarthritis patients where placebo response was predicted from resting state fMRI functional connectivity[22]. We used resting state functional connectivity to identify four regions predicting patients in the placebo arm that responded to treatment: the right mid-frontal gyrus connectivity ($x = 28$, $y = 52$, $z = 9$), the anterior cingulate cortex ($x = -3$, $y = 40$, $z = 2$), the posterior cingulate cortex ($x = -1$, $y = -45$, $z = 15$), and the right somatosensory cortex ($x = 60$, $y = -7$, $z = 21$). We next entered these coordinates as seeds in the Neurosynth analytic tool (http://neurosynth.org[55]; seed based functional networks generated in 1000 healthy subjects[56]) and extracted three networks sharing strong connectivity with these seeds: the DMN, the frontoparietal network, and the sensorimotor network. We identified communities corresponding to these networks based on spatial overlap, by multiplying the networks of interest with the nodes pertaining to each community. A total of 113 nodes were affiliated with these communities (Fig. 2a). The 151 nodes affiliated with the visual and saliency communities and those nodes without affiliation to any community were excluded from the analyses. The limbic nodes and a node located in the PAG from the Power parcellation scheme (which was not affiliated with any community) were added for a total of 122 nodes of interest. This approach was part of our initial analysis design because it has many advantages, including increasing statistical power by limiting the number of comparisons, preventing over-fitting of the data, and identifying hypothesis-driven functional networks with the potential of generalizing obtained results across different chronic pain conditions.

**Network statistics**. Network statistics were performed to identify brain networks predisposing to placebo response (performed on the $122 \times 122$ connectivity matrix). Group differences were examined using a permutation test (5000 permutations) on the connections of the weighted network ($122 \times 121$ nodes), controlling for false discovery rate (FDR $p < 0.05$) using the Network-Based Statistics toolbox implemented in Matlab[57] and freely available on the Neuroimaging Informatics Tools and Resources Clearinghouse (NITRC). The toolbox is specially designed for mass-univariate testing of connections in graphs and for accounting the dependence issue. Briefly, the first step consists of independently test every connection in the network and identify the connections with a test statistic value exceeding a threshold. Then, the toolbox identifies the topological clusters among the set of suprathreshold connections. The fisher-z transformed correlation coefficients $z(r)$ of the significant connections were extracted at each visit and entered in a repeated measured ANOVA testing for an interaction of time with placebo treatment response.

**Voxel-based morphometry**. Gray matter density was examined using voxel-based morphometry from FSLVBM. All T1-weighted images were first brain extracted and then segmented into gray matter, white matter, or cerebrospinal fluid. A common gray matter template was generated for CBP by registering and averaging all gray matter images. The gray matter image of each participant was then registered to the common template using non-linear transformation. A voxel-wise permutation test was used to test the significance of group differences between placebo responders and non-responders to a distribution generated from 5000 permutations of the data for each voxel of the template, using a sigma filter of

3 mm for smoothing. The initial analysis established significance level using the Threshold-Free Cluster Enhancement (TFCE) method (FWE $p < 0.05$).

**Cortical thickness**. Cortical thickness was examined using Freesurfer software library (http://surfer.nmr.mgh.harvard.edu/). In brief, the structural processing includes skull stripping, intensity normalization, Taliarch registration, segmentation of the subcortex, reconstruction of the cortical surface, and tessellation of the gray/white matter boundary and pial surface. Following reconstruction of the cortical surface, brains were inflated, averaged across participants to produce a study-specific brain, and then smoothed using a 10 mm full-width at half maximum Gaussian kernel. A direct measure of cortical thickness was calculated using the shortest distance (mm) between the pial surface and gray-white matter boundary at each point or vertex. Cortical thickness analysis for each hemisphere was conducted using FreeSurfer's Query, Design, Estimate, Contrast (QDEC) graphical interface. The initial vertex-wise comparison was performed between placebo responders and non-responders for each hemisphere. Correction for multiple comparisons was performed using random-field-theory-based significant clusters at $p < 0.05$. Values of cortical thickness were extracted in the significant cluster surviving multiple comparison and compared between PTxNonR, PTxResp, NoTxNonR, NoTxResp groups using a one-way ANOVA. The values of cortical thickness in the significant cluster were then extracted at each visit and entered in a repeated measured ANOVA.

**Subcortical volumes**. Volumetric analyses of T1-weighted images were performed through automated processes using both FSL (version 5.0.8) and FreeSurfer (version 6) software. We investigated volume differences in 3 subcortical nuclei selected a priori; the NAc, the amygdala, and the hippocampus. After using FSL's brain extraction tool (BET) to remove the skull from all images, FSL's integrated registration and segmentation tool (FIRST) was utilized to segment these specific subcortical regions and extract their volume measurements[58]. Unilateral volume measurements for each region were initially compared between responders and non-responders. Given the recent evidence from the ENIGMA consortium showing that subcortical volume asymmetry can provide a brain signature for psychopathologies[36], we also investigated the possibility that asymmetry differences may provide a biomarker for placebo propensity in our data. All subcortical regions' volumes were summed for the right and the left hemisphere separately; for each patient, the ratio between the two (right/left) was created, where a result $= 1$ would be indicative of perfect subcortical symmetry, whereas numbers >1 or <1 would indicate asymmetry biased toward the right or left hemispheres, respectively. Volumes and subcortical asymmetry were compared between PTxNonR, PTxResp, NoTx using a one way ANCOVA controlling for peripheral peripheral gray matter volume, age and sex. The effect was tested across all visits using repeated measure ANCOVA controlling for peripheral peripheral gray matter volume, age, and sex.

**Analysis of questionnaire data**. Over the course of the 6 visits, participants filled out 29 unique questionnaires. These specific self-report measures were chosen for one of 4 reasons: (1) to gather basic information about participants, including demographics and pain/medical history, (2) to track any changes in the quality and/or intensity of pain characteristics as measures of treatment efficacy, (3) to monitor any changes in emotional affect which may have influenced someone's time in the study or their treatment response, and (4) to capture trait-based qualities, general habits and beliefs, or state-related expectations of individuals that may predispose them to respond to placebo. Questionnaires used to track pain and mood changes overtime were repeated across all study visits. Questionnaires that targeted expectations towards treatment and satisfaction after treatment were conducted twice—either before treatment sessions (visits 2 and 4) or after treatment periods (visits 3 and 5), respectively. In contrast, measures that aimed to identify more stable traits of participants were completed at visit 1, which allowed us to use them as possible predictors of response. Finally, a subset of questionnaires regarding beliefs toward alternative medicines and suggestibility were administered at the final visit after the exit interview. A full list of all questionnaires used, along with descriptions and references, can be found in Supplementary Table 4. The data analyzed here, with the exception of the pain questionnaires collected at every visit to determine treatment outcome, come from those questionnaires collected at visit 1 only, as we were interested in looking at predictors of placebo response.

Data from these self-report measures were downloaded directly from REDCap as a CSV file and scored in Excel according to their references. Because all questionnaires were converted to an electronic format in order to be used in REDCap, an option to "skip" a question was provided if the participant did not feel comfortable answering a certain item. If >20% of the data from a given questionnaire (or questionnaire subscale, if applicable) was missing, the person's data for the questionnaire was not scored; for all other missing data, the mean was used to fill in missing items (if the questionnaire had sub-scoring, the mean was calculated from the remaining items in the sub-dimension as opposed to the entire questionnaire); this approach is one of the most commonly used methods in data analysis[59]. It was utilized in order to conserve statistical power, given our relatively small sample size. Of all the self-report data analyzed, <3% was totally missing and thus unable to be filled in as described above.

For pain measurements converted in %analgesia, outliers were defined as values exceeding 3 standard deviation from the mean and were excluded from the analyses. For each analysis, the number of participant is displayed in each figure.

**Machine learning analyses**. The predictive value of brain imaging and questionnaires data were tested using models of machine learning. We implemented a nested leave-one-out-cross-validation (LOOCV) procedure where models were trained in an inner loop ($n = 42$) and applied to a left-out participant. The purpose of the inner loop was to optimized the parameters of the model through cross-validation. Once the optimal model showing the least amount of error was identified, it was applied to the left-out participant to either classify the patient as a PTxResp or PTxNonR or predict the magnitude of his response. This procedure was applied to build predictive model from rsfMRI, brain anatomy and personality independently.

**Features selection**. rsfMRI: The model was built based on the weighted 7381 connections from the matrices used for network analyses. Feature selection was initially performed within the inner loop using several data reduction strategies: principal component analyses (PCA; 42 components), unsupervised machine learning (CorEx (https://github.com/gregversteeg/CorEx); 40 variables), or averaged connectivity within and between communities. Features selection was also performed using univariate t-test on each connection to identify group differences (PTxResp Vs PTxNonR within the raining set; $p < 0.001$) or links correlating with the magnitude of response (robust regression with %anagesia within the raining set; $p < 0.001$; results shown in Fig. 5j).

Anatomy: Three different measures were used to predict the magnitude of response using brain anatomy: (1) the averaged cortical thickness in the 74 labels per hemispheres from the Destrieux Atlas[60], (2) volumes of the 16 subcortical structures segmented with FSL FIRST, (3) subcortical volume asymmetry (ratio Right/Left) of these 16 subcortical structures. Because the features were derived from different measurements, the data were normalized.

Questionnaires: All self-report measures collected at V1 were entered in the model. Because the questionnaires were on different scales, the data were normalized.

**SVM for classifying PTxResp and PTxNonR**. Support vector machine (SVM) was used to discriminate between PTxResp and PTxNonR using fitcsvm function implemented in Matlab. We implemented a LOOCV procedure where SVM models were trained in an inner loop ($n = 42$) and applied to a left-out participant. The box constraint and the radial basis function (rbf) kernel were optimized through a tenfolds cross validation strategy within the inner loop. Once the optimal SVM model was identified, it was applied to classify the left-out patient as a PTxResp or PTxNonR.

**LASSO for predicting magnitude of response**. We used Least Absolute Shrinkage and Selection Operator (LASSO) regressions to train a model predicting the magnitude of response. Here again, we used a nested LOOCV procedure where models were trained in an inner loop ($n = 42$) and applied to a left-out participant. The inner loop determined feature selection and lambda regularization parameters using tenfolds cross-validation. The generalization error was estimated by testing the model to the left-out patient. The procedure was performed using the lasso function, implemented in Matlab.

**Statistical analyses**. A description of each statistical test and its exact values are reported in Supplementary Table 7. All test performed in this study were two-sided.

**Data availability**. Data from our previous studies are already available on http://www.openpain.org/. The data are part of a longitudinal study that will generate more than one manuscript. The data will eventually be made available on open pain once these manuscripts are completed. Since then, data are available upon reasonable request.

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

## Acknowledgements

We are thankful to all Apkarian lab members that contributed to this study with their time and resources. We would also like to thank all patients that participated in this study. This study was funded by National Center for Complementary and Integrative Health AT007987. E.V.-P. was funded through Canadian Institutes of Health Research (CIHR) and Fonds de Recherche Santé Québec (FRQS).

## Author contributions

E.V.-P. co-lead the RCT, collected the data, analyzed the data, and wrote the manuscript; S.E.B. co-lead the RCT, collected the data, analyzed the data, and wrote the manuscript; T.B.A. collected the data and analyzed the data; L.H. analyzed the data, G.A.C. analyzed the data, J.W.G. analyzed the data, T.J.S. designed the RCT, A.V.A. designed the RCT, analyzed the data, wrote the manuscript.

## Additional information

**Competing interests:** The authors declare no competing interests.

