## [Peer Review File · Nature Communications]

Editorial Note: Parts of this peer review file have been redacted as indicated to remove third-party material where no permission to publish could be obtained.

Reviewers' comments:

Reviewer #1 (Remarks to the Author):

The manuscript by Vachon-Preseau and colleagues describes multiple psychological and neural determinants of the placebo response in a group of back pain patients. The paper is technically highly demanding and was no doubt demanding during data collection. I admire the work that has gone into this manuscript. Unfortunately, as the submission stands, I don't think it will have the impact it probably deserves.

The paper is immensely dense. I have no direct suggestions for how to relieve the reader of the density, possibly my further critical comments will be helpful, but it is currently difficult to see how the paper can be published in a journal for a general audience. The manuscript includes five figures, but those five figures deliver 33 panels.

Beyond the density of the paper, I have two major criticisms.

1. The authors suggest that they will be prospectively looking at emotion awareness and openness from questionnaires as predictors of placebo response, but then they rightly note in the discussion that the placebo literature largely indicates optimism, anxiety, extraversion, neuroticism (and expectancy, which they miss) as predictors of placebo response. Those latter items were not predictive for the current study, which may or may not matter, but it does call into question the idea that certain variables were pursued prospectively.

Related to that, the authors also suggest that they will be pursuing certain brain regions prospectively (DLPFC, PAG, rACC, PreCG). But then the methods describe a (surprisingly complex) approach to generating ROIs in different regions. Again the prospective nature of the analysis is unclear.

2. The observed placebo response is quite limited. Placebo effects were restricted to pain intensity ratings and did not extend to qualitative measures of pain. The pain intensity ratings themselves were also small, just a 20% drop in pain intensity ratings. Those changes would not be considered clinically significant and are not likely to have made much of an impact on the patients in the study; they may not have even been noticed. Consequently, the authors are building an immense amount of statistical modeling onto a quite flimsy foundation.

Minor criticisms:

1. Responder and non-responder are not fixed categories. A patient may not respond to one placebo (say blue pills) but will respond to a different one (say red pills). Alternately, the same patient can start responding or stop responding to the same placebo at a later date. These changes imply that fairly transient factors might predict response rather than relatively fixed features of the brain or psychological traits.

2. Limbic covers a lot of brain regions, and there is dispute over what should and should not be included in the "limbic system" – a more precise term would be better.

3. There were no direct references supplied for the paragraph describing the prospective measures.

4. The responders to no treatment appear to me to indicate a baseline false positive rate, or natural remission rate, rather than an alternate or additional measure of placebo responses.

5. Absolute baseline levels of pain at entry are not given, making it impossible to assess whether there might have been regression to the mean or other factors driving the placebo response.

Reviewer #2 (Remarks to the Author):

This study takes on the important question of predicting placebo response magnitude in chronic low back pain. It is also one of a very few studies to establish causal benefits of placebo treatments in chronic pain with neuroimaging, by examining placebo treatment vs. no treatment in a randomized controlled trial. This is very important, as there are virtually no other examples in the neuroimaging literature (though Pecina et al.'s 2015 study of depression are an exception; see below). Study is impressive in that there is a reasonably large cohort ($n = 63$) scanned on four separate occasions, with ecological momentary assessment (EMA) of pain via smartphone over a number of days. The cross-validated prediction analyses are helpful, and offer some hope that placebo responses might be predicted by a combination of questionnaire and brain data in advance of treatment.

Overall, this is a very strong paper and a really good effort by the research team, with a number of strong features, and translational implications.

I have various comments about the strengths and some questions about the procedures and findings, below.

Placebo responses on pain ratings were significant in the whole group and reasonably large (about 10% reduction), and the EMA data across > 2 months adds value — this is a fairly unique demonstration of sustained placebo analgesia. Placebo responders tended to respond immediately. (The authors might discuss why placebo washouts seem to have failed in clinical trials, however.)

The authors might also comment on why the McGill and PainDetect did not show much placebo effect; they might be expected to as they are sensitive to pain intensity as well.

The idea of blinding the analyst and performing three separate analyses is innovative and adds rigor. If "The results are reported only when the real labeling of patients could be properly identified," what happened to the results where the analyst was incorrect about which is the correct subject labeling? Were there such results and analyses that were discarded?

Another strength is that the search for predictors of placebo response was constrained somewhat by the authors' previous findings.

The study jumps right into differentiating placebo responders from nonresponders. One of the strengths of the analysis is the ability to look at placebo treatment versus no treatment groups. Where there significant differences between placebo and no treatment groups? These differences would also estimate the causal effects of placebo treatment on brain networks.

One might also expect placebo-predictive brain features to predict the magnitude of the placebo response in the placebo group but not predict the magnitude of pain reduction in the no treatment group.

It is also interesting that in the plots, no treatment often falls right between placebo responders and nonresponders. It seems unlikely that the findings would actually reliably discriminate who received a placebo from who did not. This is curious, as the measures seem to discriminate responders from nonresponders who received placebo. Wouldn't one expect to be able to tell responders from those who didn't get a placebo at all?

The seed labeled DLPFC really looks like it is in the anterior ventrolateral prefrontal cortex. How far is it from region identified as placebo predictive in the Tetrault paper?

Effect sizes when picking the most predictive questionnaire from among a larger set tested are going to be upwardly biased, and we can't make so much of the effect size (r^2 is probably less than 0.36); Fig 4.

The cross-validated prediction analyses are helpful, and offer some hope that placebo responses might be predicted by a combination of questionnaire and brain data in advance of treatment.

Did the classifiers and cross-validated regression models also classify no-treatment responders vs. non-responders, however? Only if there is a strong effect in the placebo group but not the no-treatment group is this likely to be prediction of the placebo effect. Otherwise, it may predict those whose symptoms are high at visit 2 in ways not captured by ratings, and who will show decline due to regression to the mean or other factors.

It is interesting that the weights in the cross validated support vector machines analysis do not qualitatively match up very well with the results presented earlier in the paper. This could be misleading, however, and an effect of thresholding. Is there a systematic relationship between the univariate effect strength and the multivariate pattern weights?

What variables were optimized over in nested cross-validation, and what were the most frequent optimal choices?

It is curious that the brain data could predict the magnitude of the placebo response but could not classify placebo responders versus nonresponders, which should be fairly correlated with the placebo response magnitude. Any thoughts about why?

Regarding this: "the prediction did not generalize to rsfMRI data collected at other visits post treatment. This is likely due to the small number of connections included in our model..." Was this tested for questionnaire predictors as well? If the problem is unique to the brain data, as the authors suggest, then the questionnaire data should predict future visits. I don't think this is reported.

Discussion: "novel systems we uniquely associate with placebo pill response"...unique in what sense? Not that these connections have no other functions in the brain, surely?

The discussion could be toned down some, and the introduction as well, to moderate some of the very strong statements and claims. Also, very little of the relevant literature is cited in the introduction.

Additional comments

Fig 1d: It does not make sense to statistically test responders vs. non-responders - the groups were selected based on a difference, so this is biased.

"the placebo effect is observed universally" ... there is an emerging consensus in the placebo literature that placebo effect should be defined as the estimated causal effect of placebo, which requires comparison to a no-treatment group. Placebo responses, on the other hand, are overall improvements on placebo treatment. It may be more accurate to say that placebo responses are observed universally.

"Yet, current scientific dogma assumes that RCT-related placebo responses are due primarily to uncontrollable confounds..." This is true in many areas of medicine, but there has been a robust literature on predictors and correlates of placebo responses, in the brain imaging literature and older behavioral literature as well. This might be mentioned at this point. The consistency of which variables are good predictors across studies, however, has not been high.

Another imaging RCT of placebo effects:

Peciña, Marta, Tiffany Love, Christian S. Stohler, David Goldman, and Jon-Kar Zubieta. 2015. "Effects of the Mu Opioid Receptor Polymorphism (OPRM1 A118G) on Pain Regulation, Placebo Effects and Associated Personality Trait Measures." *Neuropsychopharmacology: Official Publication of the American College of Neuropsychopharmacology* 40 (4). nature.com: 957–65.

The within-subject permutation test is likely not valid unless the serial autocorrelation in the data is corrected for, which is nontrivial. (And what was the cutoff?) However, this is a minor point.

There are a number of typos in the legend of figure two.

"All post hoc comparisons were Bonferroni corrected" how many comparisons, and what was the correction factor? This doesn't mean much without more information.

"A permutation test was performed on the weighted 7,381 resting state connections..." It is not clear how it was done, or whether it is correct, due to the potential dependence issue...Nichols and Hayasaka have recommendations. More detail would be helpful.

"The volumes of the NAc, amygdala, and hippocampus were first examined"... the authors might cite Schweinhardt and Bushnell's study here, which was possibly the largest study to date predicting placebo responses from brain, and showed a relationship with NAc volume.

"Interhemispheric laterality of the combined volume of these three structures, however, indicated..."

Why combine them? And was this done a priori?

"scramble codes" is not grammatically correct. "scrambled codes" = better

A hundred randomized labels is not very many permutations. At least a thousand would give a better estimate of the distribution. But this is a very minor point.

There are also some other typos related to standard English usage: e.g., "10 folds cross validations"

Reviewer #3 (Remarks to the Author):

Vachon-Preseu et al. present a novel and comprehensive study on neuroimaging measures and psychological questionnaires that determine whether a placebo treatment will reduce chronic back pain in a given patient. They show that specific psychological factors, and to some degree neural differences (functional connectivity and structural volume/thickness), predispose patients to placebo analgesia.

This work is both clinically relevant and scientifically important. The unique sample, systematic approach, and use of state-of-the-art neuroimaging tools and data analyses are major assets. It is clear that this study was carefully planned and conducted. I appreciate the authors' attention to detail and sensible interpretation of the results. Systematic studies of placebo analgesia within patient populations in the RCT context are largely lacking in the literature. I suspect that this work will help the field to appreciate how placebo effects are generated in a clinical context (in chronic pain and beyond).

Most of my comments below are minor, although I do have major comments about the functional connectivity analyses that I would like to see addressed.

Major comments:

- A major conclusion is that psychological questionnaires are more informative than neuroimaging metrics in predicting placebo analgesia. However, this conclusion may be dependent on the specific features selected for neuroimaging analysis. I appreciate that the authors emphasize a focus on pre-planned analyses grounded in the literature, but to some degree the choices made could be biased (e.g. the rs-fMRI regions of interest rely on specific results from a study of a different patient population). In rsfMRI, an exclusive focus on the DMN, sensorimotor and frontoparietal networks (and limbic/brainstem areas) misses the salience network, which includes critical regions involved in pain such as the anterior insula and mid-cingulate cortex. Moreover, functional connectivity is just one metric that can be derived from rsfMRI. Others are based on local amplitude fluctuations or, importantly, dynamic connectivity metrics that are showing increasing relevance to pain in the literature. Of course there is a risk for overfitting, but it remains possible that the inclusion of additional regions and metrics would allow a better prediction of placebo analgesia. I suggest that the authors address this possibility with additional analyses and/or critical discussion.

- I could not find a description of individual differences in head motion for rs-fMRI analyses/results. The impact of relative frame-wise displacement (FD) on functional connectivity is known to persist even after performing many of the preprocessing steps done by the authors, and thus I recommend that the authors report on this metric and a comparison of its values in the patient subgroups analyzed (see guidelines from Power et al 2015 Neuroimage).

Minor comments:

- In the main text, I think some description is needed of what rsfMRI and functional connectivity are and why one would look to this approach for predicting placebo analgesia.

- In the main text, I think some justification is needed about why the authors decided to study hemispheric laterality of brain volume (as described in methods)

- I don't see a description of Figure 3c in the main text

- In organization of the Results, I wonder why the authors chose to present the questionnaire data (from visit 1) after neuroimaging (data from all visits), which does not necessarily reflect chronological order or the subsequent order in which the machine learning results are presented.

- Which specific cingulate subregion is the region described as "rACC" (e.g. according to the Vogt scheme)? Is this the pregenual ACC or anterior MCC?

- Some methodological details are missing in describing the rsfMRI analysis that would limit the ability of a researcher to reproduce this work.

- I could not find information about the type of MRI scanner used and Tesla strength

- Why were 120 volumes deleted from each fMRI run (more excessive than usual)?

- Please explain what "fsl_motion_outliers" does specifically

- Please explain how white matter and ventricle volumes were "eroded"

- Please explain how “components obviously related to motion or noise” were identified after ICA. This is often non-trivial, and thus automated approaches (e.g. ICA-AROMA) are increasingly used.
- Please cite Yeo et al (2011 J Neurophysiol) which describes the data that Neurosynth uses to identify connectivity networks
- Please check for typos (e.g. lines 290, 426, 563, 786, 812, 823) and replace “data was” with “data were”

Reviewer #4 (Remarks to the Author):

There is continued interest in both the mechanisms and clinical implications of the placebo effect that reaches far beyond the pain field. While this is not the first clinical trial testing the efficacy of placebo treatment in patients with chronic pain, this work is original and novel especially due to a thus far unique and impressive combination of state-of-the art methodologies designed to test psychological and neurobiological mechanisms differentiating placebo responders and non-responders and identifying predictors of the placebo response.

The manuscript is highly complex and in parts difficult to follow. Some parts of the results are difficult to grasp and more data need to be shown prior to data reduction/transformation to allow a proper assessment of the pain outcomes. Several issues remain unclear, as specified in more detail below:

The introduction is not well-balanced and does not adequately acknowledge relevant studies in the field (e.g., IBS placebo trials; German acupuncture placebo trials; brain imaging studies in patients with diverse chronic pain conditions). Aims and hypotheses are not adequately specified and explained. Some crucial and novel aspects of the study design, especially inclusion of two treatment periods, each followed by a washout period, are not motivated in the introduction and are only briefly explained in the methods. The crucial role of patients' expectations, arising from the entire psychosocial treatment context, is neglected both conceptually and literally. Several statements in the introduction are not unequivocal and/or not supported by appropriate references. Examples: “... the placebo effect is observed universally in almost all randomized placebo-controlled clinical trials (RCT), particularly in pain treatment trials” – seems to imply that the placebo effect in pain RTCs is higher than in RTCs on other medical conditions – the reference provided here does not state this; “Effect size of placebo response is usually equivalent or superior to active treatment” – disagree with that broad statement, especially since it is positioned in a general context of RCTs and yet only one reference (for neuropathic pain) is provided; why will the placebo effect in RCTs stop to be a “confounding nuisance” once its underlying properties are understood?; disagree with the statement that it is “current scientific dogma” that RCT-related placebo responses are “due primarily to uncontrollable confounds”; it is not true that the neurobiological mechanisms have almost exclusively been studied for acute responses to conditioning-type manipulations in healthy individuals, and the reference provided does not provide support for this statement. There exists an abundance of studies in patients with diverse medical conditions, including several studies conducted in patients with chronic pain. Also, many existing studies laboratory studies have used manipulations built on suggestions rather than conditioning to induce the placebo response.

Methods/results:

Inclusion/exclusion and recruitment strategy/patient information: Given the crucial role of expectations as mediator of placebo effects, what was the information provided to patients during recruitment and informed consent? How were other co-morbid chronic pain conditions, neurological and psychiatric conditions excluded? Was there a psychiatric interview? Screening

questionnaires for comorbidity? Inclusion criterion of at least 5/10 for pain during screening: was this a verbal assessment and why is the number of those who failed this criterion not indicated on the flow chart (Fig. S1)? What time frame was this based on (e.g., at this moment, today, this week?). Which clinical measurements during visit 1 were required to be within the pre-specified healthy range and what was that range?

Placebo intervention: Given that a number of different placebo interventions have been tested in the field, what was the rationale for choosing this type of placebo pill combination? Is it not likely that many patients already had experience with this particular combination of drugs? Could this have introduced a recruitment bias, e.g., by discouraging patients who had unsuccessfully used this drug in the past?

Only 5 patients were given active treatment to maintain double-blind for placebo. Given the ratio of 10 (placebo) to 1 (active treatment) – is this truly adequate to maintain double-blind? The methods state that patients were never informed about the odds of receiving placebo or active treatment. It is difficult to imagine that patients did not inquire about this. What were they told if they did ask? Is there an ethical aspect to consider given the odds of receiving active treatment and deception? In this context, it is stated that this was done with the goal to have patients' "baseline expectations influence whether or not they responded to placebo treatment". Were baseline expectations assessed? Were prior treatment experiences with this particular treatment regimen or other pain treatment assessed? After the end of the trial, were patients asked about perceived group assignment?

Along the same lines: How were treatment interruptions communicated to the patients and what effects could this have had on treatment expectations, especially for subsequent treatment phases? One would expect strong "carry-over" effects. Arguably, this treatment interruption causes a nocebo-type situation, i.e., withdrawal of active treatment, with likely effects on subsequent treatment results depending on the subjective experience during the first treatment and first washout phase. Although Fig. 1d appears to indicate that this was not the case, since these data are only displayed after stratification into responders and non-responders, it is difficult to properly assess this.

What exactly happened in the no treatment group? What information was provided for participants? Were they required to remain off pain medications for the entire duration of the study? Have the authors considered nocebo-type effects in the no treatment group (e.g., if recruitment strategy / adverted for a study providing pain relief, to be confronted with the decision to be in a no-treatment group, this would lead to frustration, disappointment, anxiety)?

What was the number of patients recruited and then randomized based on? Please provide power analyses. After stratification into responders and non-responders, group size is not only uneven but also much too small in some groups, especially in the group of no-treatment responders (N=4 in most analyses), and yet, many comparisons were carried out using ANOVA with 4 groups.

Pain measures: It is a strength that the authors utilized several pain measures. However, I question the validity of the "memory pain score". In addition, presentation of all data as % change from baseline does not allow the reader to properly inspect and understand the results. While the need for data reduction is obvious, percent change arguably does not allow conclusions about clinical relevance of changes and "distorts" actual changes. At minimum, in a first step, the authors should provide more details on all pain measures across measurement points without group stratification into responders and non-responders and show how real changes (in mm or original scale scores) compare to % change used in subsequent analyses. It is crucial to clarify which of the pain measures actually responded to placebo intervention compared to no treatment. As is, Fig. 1c is the only such evidence that is provided within the manuscript and that is not sufficient. Based on pain data provided e.g. in Fig. S3, it appears that placebo treatment in this study was possible not superior to no treatment (?). After stratifying into responders and non-

responders, group differences in pain ratings (e.g. as indicated in Fig. S3) merely reflect the stratification.

Table S1 should provide basic demographics for groups as shown in Fig. S1 rather than for responders and non-responders. The table legend states "There were no differences between groups in age, gender, duration of pain reported, or years of education" – is this the result of some sort of statistical analysis or based on visual inspection? Given the small N, it appears difficult to properly analyze these data statistically and visual inspection at least does not fully support that all groups were equal, especially for proportion of females and duration of pain.

Since the main goal was to predict placebo response: what was the amount of variance that could be explained with psychological (questionnaire) versus biological (imaging) variables?

Discussion:

As in the introduction, the results are not properly discussed in the context of existing placebo brain imaging studies and placebo trials. How does the finding that pain intensity (assessed with daily ratings) responds to placebo treatment and validated other pain measures do not respond compare to the literature? Should existing "gold standards" for assessing pain in pain trials be changed? Conclusions about correspondences in the mechanisms across different types of pain and settings are speculative and not referenced properly. The possibility that prediction of placebo effects is better in patients with chronic pain in the context of a clinical trial is pure speculation and in fact counter-intuitive especially due to different individual treatment histories. The authors did not assess expectations – the mediator of placebo effects – which is not acknowledged in the discussion section. The conclusion that placebo effects have their own psychological and neurobiological determinants is almost trivial at this stage given the abundance of research on the topic.

Reviewer #1 (Remarks to the Author):

The manuscript by Vachon-Preseu and colleagues describes multiple psychological and neural
determinants of the placebo response in a group of back pain patients. The paper is technically
highly demanding and was no doubt demanding during data collection. I admire the work that
has gone into this manuscript.

Unfortunately, as the submission stands, I don't think it will have the impact it probably
deserves. The paper is immensely dense. I have no direct suggestions for how to relieve the
reader of the density, possibly my further critical comments will be helpful, but it is currently
difficult to see how the paper can be published in a journal for a general audience. The
manuscript includes five figures, but those five figures deliver 33 panels.

**Response:** *We appreciate the positive assessment and take into consideration the issue of the*
*density of the paper. As the reviewer notes this is a large undertaking and we want to report the*
*important results transparently. Thus, we chose to present as much of the data as possible. Given*
*the importance of the topic and its implications, we feel that it is crucial that the results are*
*reported thoroughly.*

*Still the point regarding density of the paper is well taken. In this revised version, the narrative*
*was reorganized and simplified, and we took extra care to improve the clarity of the language*
*used, including statements that summarize reported observations. Moreover, figure 1 has been*
*simplified.*

Beyond the density of the paper, I have two major criticisms.

1. The authors suggest that they will be prospectively looking at emotion awareness and
openness from questionnaires as predictors of placebo response, but then they rightly note in the
discussion that the placebo literature largely indicates optimism, anxiety, extraversion,
neuroticism (and expectancy, which they miss) as predictors of placebo response. Those latter
items were not predictive for the current study, which may or may not matter, but it does call into
question the idea that certain variables were pursued prospectively.

**Response:** *The reviewer's statement is not quite correct. The study is prospective but the*
*personality questionnaires were only administered prior to start of trial and used for*
*determining placebo response. Brain properties and pain were repeatedly assessed throughout*
*the trial. We apologize if the description of the design was not clear. We have further clarified*
*this point in the introduction.*

Related to that, the authors also suggest that they will be pursuing certain brain regions
prospectively (DLPFC, PAG, rACC, PreCG). But then the methods describe a (surprisingly
complex) approach to generating ROIs in different regions. Again the prospective nature of the
analysis is unclear.

**Response:** *We believe that there may have been a certain level of confusion regarding the design*
*of the study and the analyses (we have attempted to clarify the language, especially in the*

*introduction). We observed patients receiving placebo treatment in the setting of a randomized*
*controlled trial (RCT) and identified the placebo responders. Then we interrogated the brain*
*imaging data and psychological factors collected at baseline to identify the determinants of*
*placebo response. The method may seem complex because the brain features (connectivity*
*between brain regions) were not selected a priori, but were rather identified based on statistical*
*modeling (i.e., machine learning models identified features capable of dissociating responders*
*from non-responders, or predicting the magnitude of response). Note that as all our analyses*
*were conducted according to a pre-determined plan, properly corrected for multiple*
*comparisons (fig. 2-4) and/or cross-validated in a nested leave-one-out procedure (fig. 5).*
*Similar prospective studies have been used to predict recovery one year after a stroke using grey*
*matter and functional connectivity at baseline (Ramsey et al., 2017 Nat. Hum. Behav.), or*
*children who develop autism at 2 years old from grey matter or brain connectivity collected at 6*
*months (Emerson et al., 2017 Sci Transl Med.; Hazlett et al., 2017 Nature).*

*Regarding assessing expectancy, please see response to reviewer 4.4 below.*

2. The observed placebo response is quite limited. Placebo effects were restricted to pain
intensity ratings and did not extend to qualitative measures of pain. The pain intensity ratings
themselves were also small, just a 20% drop in pain intensity ratings. Those changes would not
be considered clinically significant and are not likely to have made much of an impact on the
patients in the study; they may not have even been noticed. Consequently, the authors are
building an immense amount of statistical modeling onto a quite flimsy foundation.

**Response:** *The reviewer is correct to note that an important new concept in this paper is the idea*
*that only pain intensity but not its qualitative properties responded to placebo. This distinction is*
*a first of its kind and was only possible by having a no-treatment group, where qualitative*
*properties changed similarly to the placebo treated group.*

*We believe that placebo responders (PTxResp) showed a strong response to placebo pills for*
*several reasons. First, the placebo response was immediate and sustained in time over dozens of*
*data entries. Second, the 20% analgesia pointed out by the reviewer represents the response*
*averaged over both treatment periods across PTxResp (by itself a clinically significant change).*
*However, some patients responded more strongly to one treatment period over the other.*
*Looking at the highest %analgesia between the two treatment periods (referred to as magnitude*
*of response) indicated an average of 33% analgesia (median: 28%; see **Figure 7** page 7 below).*
*Note that the literature usually considers a 15% analgesia as a ‘minimal clinically important*
*difference’ (Salaffi F, 2004, EJP) and a 30% analgesia as ‘clinically important pain diminution’*
*(Farrar 2001, PAIN). Third, we computed effect sizes (Cohen’s d) in PTxResp with respect to the*
*NoTx group controlling for the natural history of the patient. We observed that the effect sizes*
*were 1.16 (95% CI: 0.82 to 1.50) for treatment 1, 1.13 (95% CI: 0.79 to 1.15) for treatment 2,*
*and 1.71 (95% CI: 1.42 to 2.0) for the magnitude of response. Note that effect sizes > 0.8 are*
*usually considered strong. We conclude that PTxResp met both criteria of statistical and clinical*
*significant analgesia. We have added the effect size results to the paper.*

**Minor criticisms:**

1. Responder and non-responder are not fixed categories. A patient may not respond to one
placebo (say blue pills) but will respond to a different one (say red pills). Alternately, the same
patient can start responding or stop responding to the same placebo at a later date. These changes
imply that fairly transient factors might predict response rather than relatively fixed features of
the brain or psychological traits.

**Response:** *We partially agree with this statement. Our study shows that personality and brain*
*properties identify placebo responders. The reviewer is correct that previous literature shows*
*placebo response depends upon contextual cues. However, it is also likely that placebo response*
*depends on the **interaction** between both external contexts and individual differences. Here, we*
*show that our predictive models of personality and brain connectivity accounted for about 36%*
*of the variance in the magnitude of response. Some of these predictors were transient and may*
*(at least partly) reflect changes in patients' expectations or changing contexts. Furthermore,*
*most of the variance remained unexplained and probably depended on other factors including*
*the context in which the treatment was administered and other external influences that cannot be*
*easily measured or controlled for.*

2. Limbic covers a lot of brain regions, and there is dispute over what should and should not be
included in the “limbic system” – a more precise term would be better.

**Response:** *We changed limbic for subcortical limbic structures, and in other places simply*
*removed the descriptor.*

3. There were no direct references supplied for the paragraph describing the prospective
measures.

**Response:** *We have clarified the language especially regarding the study design; again, there*
*were no a priori selected measures.*

4. The responders to no treatment appear to me to indicate a baseline false positive rate, or
natural remission rate, rather than an alternate or additional measure of placebo responses.

**Response:** *We largely agree. However, alternative or additional explanations, such as placebo*
*response to other elements of the RCT (MRI sessions or exposure to a medical environment, for*
*instance, or an increase in attention/personal communication via interactions with study staff*
*and daily app ratings) cannot be ruled out.*

5. Absolute baseline levels of pain at entry are not given, making it impossible to assess whether
there might have been regression to the mean or other factors driving the placebo response.

**Response:** *This is an excellent point and it should have been presented in the original*
*submission. We have now included absolute baseline levels of pain. None of the 6 pain*
*assessments were different between the groups.*

	Phone	Memory	NRS	MPQa	MPQs	Pain Detect
--	-------	--------	-----	------	------	-------------

PTxNonR	6.22 (1.12)	6.59 (1.48)	55.10 (23.14)	3.21 (2.44)	12.54 (5.34)	9.97 (7.44)
PTxResp	6.10 (1.33)	7.26 (1.29)	60.75 (23.21)	3.54 (2.87)	11.25 (4.2)	9.58 (5.33)
NoTx	5.68 (1.14)	6.65 (1.66)	48.85 (23.33)	4.12 (2.80)	14.90 (6.11)	13.30 (6.15)
p vals	0.33	0.26	0.24	0.58	0.08	0.12

**Table 1: Absolute levels of pain at baseline showed no differences between PTxResp and**
**PTxNonR on all pain measures.**

*These findings have been included in a table in the revised version of this manuscript.*

Reviewer #2 (Remarks to the Author):

This study takes on the important question of predicting placebo response magnitude in chronic
low back pain. It is also one of a very few studies to establish causal benefits of placebo
treatments in chronic pain with neuroimaging, by examining placebo treatment vs. no treatment
in a randomized controlled trial. This is very important, as there are virtually no other examples
in the neuroimaging literature (though Pecina et al.'s 2015 study of depression are an exception;
see below). Study is impressive in that there is a reasonably large cohort (n = 63) scanned on
four separate occasions, with ecological momentary assessment (EMA) of pain via smartphone
over a number of days. The cross-validated prediction analyses are helpful, and offer some hope
that placebo responses might be predicted by a combination of questionnaire and brain data in
advance of treatment.

Overall, this is a very strong paper and a really good effort by the research team, with a number
of strong features, and translational implications.

**Response:** *We appreciate the positive assessment.*

I have various comments about the strengths and some questions about the procedures and
findings, below.

1. Placebo responses on pain ratings were significant in the whole group and reasonably large
(about 10% reduction), and the EMA data across > 2 months adds value — this is a fairly unique
demonstration of sustained placebo analgesia. Placebo responders tended to respond
immediately. (The authors might discuss why placebo washouts seem to have failed in clinical
trials, however.)

**Response:** *We believe that the washout periods may simply have been too short. Although the*
*present study does not directly address issues related to failure of placebo washout, it does*
*provide hints on the topic. While our data shows that pill placebo response is based on brain and*
*personality properties, it also shows that some of these properties change with exposure to*
*placebo. Thus, our study suggests that testing subjects via exposure to placebo can change the*
*probabilities for ongoing and/or future. Still this idea needs a systematic study. The washout*
*issues are now summarized in the discussion section.*

*p.18 line 404: “Our study design included two washout periods in order to determine stability*
*and within subject co-occurrence of response. The use of EMAs allowed us to determine that*
*placebo analgesia started on the first day of treatment but return to baseline levels of pain*
*started only several days after washout. Thus, although the washout periods were proven too*
*short to test the co-occurrence of response, our data showed a carryover effect of the placebo*
*response after discontinuation of treatment.”*

2. The authors might also comment on why the McGill and PainDetect did not show much
placebo effect; they might be expected to as they are sensitive to pain intensity as well.

**Response:** *The MPQ uses adjectives to describe the qualities of the pain and the pain detect uses*
*a combination of words and pictograms to assess the nature of chronic pain (spontaneous and*
*fluctuating or continuous). The information provided by these questionnaires assess different*
*dimensions of chronic pain. As shown on the figure below, the correlations between the different*
*pain assessments at baseline indicated that measures of intensity (phone, memory, NRS) and of*
*qualities (MPQ and Pain Detect) weakly correlated with each other (Figure 1 below).*
*Furthermore, our data show that qualities of pain did in fact improve throughout the RCT but*
*this improvement was also observed in the no-treatment group. We conclude that they represent*
*different dimensions of pain that behaved differently in the RCT. This stresses the importance of*
*documenting the natural history of patients with a no treatment arm and the need to assess pain*
*repeatedly in the natural setting, using more than a single intensity measurement. In our case, it*
*is possible that placebo only impacted the perceived intensity of pain while a successful active*
*pharmacological treatment might improve both intensity and qualities. This remains an open*
*question, now discussed in greater detail in the discussion.*

*p.17 line 385: “One of the main behavioral findings was that the treatment outcomes exhibited a*
*high level of dimensionality that has not been specifically investigated or accounted for in other*
*clinical or basic science studies regarding modulation/perception of pain. The no treatment arm*
*documenting the natural history of the patient allowed us to show that placebo treatment*
*impacted a particular dimension of chronic pain (pain intensity) without changing the trajectory*
*of pain quality (which improved in time regardless of placebo treatment). Even the various*
*measures of pain intensity - daily ratings, memory, and NRS - provided slightly different levels of*
*information about the extent of analgesia and pain fluctuations. This highlights the importance*
*of examining a multiplicity of pain-related outcomes as the analgesic properties of any given*
*treatment may not be constrained to a single dimension of the pain experience. Most current*
*RCTs assessing new treatments are designed with a single primary outcome representing a*
*“gold standard”, thus likely missing on the complexity of treatment effects that chronic pain may*
*exhibit. Our behavioral results stress the importance of moving away from a single, cross-*
*sectional, pain measurement, and demonstrate that distinct dimensions respond differentially to*
*placebo pill ingestion, impacting mainly perceived magnitude but not its qualities. It is however*
*possible that placebos only impact the intensity of pain while successful pharmacological*
*treatments would improve both intensity and qualities. This remains an open but important area*
*of inquiry”*

 **Figure 1: Covariance matrix across pain assessments at baseline.** The intensity of pain was
 defined by Phone app, Memory of pain (a verbal report about the average pain experienced over
 the last 7 days), and a numerical rating scale (NRS collected in lab). The quality of pain referred
 to pain measurements from the MPQ and the Pain Detect. This has been added in the
 supplementary material (Fig. S3). * $p < 0.05$; ** $p < 0.01$, *** $p < 0.001$.

 3. The idea of blinding the analyst and performing three separate analyses is innovative and adds
 rigor. If “The results are reported only when the real labeling of patients could be properly
 identified,” what happened to the results where the analyst was incorrect about which is the
 correct subject labeling? Were there such results and analyses that were discarded?

 **Response:** We experienced no situation where an analysis showed significant effect for an
 incorrect label. There were, however, few cases where none of the 3 labels showed significant
 results. This happened for subcortical volumes (analyzed prior to the asymmetry measurement,
 and reported as uninformative in the main text) and the questionnaire data, which generated no
 significant results surviving correction for multiple comparisons, for all 3 scrambled labels. The
 univariate statistics are nevertheless reported in fig. 2b because questionnaire data were
 capable of predicting placebo response and magnitude of response in the multivariate analyses
 (fig. 5).

 4. Another strength is that the search for predictors of placebo response was constrained
 somewhat by the authors’ previous findings.

 **Response:** We appreciate the reviewer’s enthusiasm about our methods.

 5. The study jumps right into differentiating placebo responders from nonresponders. One of the
 strengths of the analysis is the ability to look at placebo treatment versus no treatment groups.
 Where there significant differences between placebo and no treatment groups? These differences
 would also estimate the causal effects of placebo treatment on brain networks.

**Response:** This is a good point. The effect of placebo on the brain is partially dealt with here, as
some of the networks predicting placebo response showed transiency in PTx groups. However,
this manuscript aimed at identifying predictors of placebo pill response rather than the
consequences of placebo pill response on the brain. A systematic study of brain changes between
responders and non-responders with respect to the NoTx arm is postponed to a subsequent
analysis, which is already planned.

However, following the reviewer's point, we used a repeated-measures ANOVA to test the
stability of functional connections (VLPFC-PreCG; VLPFC-rACC; VLPFC/DLPFC-PAG) and
anatomical properties (subcortical asymmetry and cortical thickness) in the NoTx arm.
Furthermore, we used a repeated-measures ANOVA, entering group assignment (PTx and NoTx)
as the between-subject variable, to determine if the mere exposure to placebo pills had an impact
on these brain properties. The results showed no effect of time or Gr or Gr*Time interactions.
These results are in **fig. S10** and reported in the main manuscript as follows:

p.12 line 266: "We then examined the variability of each of these anatomical and functional
brain measurements in patients of the NoTx arm. We observed no changes across the visits
indicating stability of the measure without placebo effects (fig. S10). We further tested if the
mere exposure to placebo pills, regardless of the response, impacted these brain measurements
by comparing the PTx group with the NoTx group. These analyses revealed absence of pill
exposure effects on anatomical and functional brain measurements (fig. S10). The result suggests
that the changes observed in VLPFC/DLPFC-PAG connectivity were primarily driven by the
actual placebo response, rather than mere pill exposure or inherent variability in the
measurements."

**Figure 2: Exposure to placebo pills has no causal effects on the brain properties determining**
 **placebo response.** The stability of the functional (a-c) and anatomical (d,e) properties was
 examined in the NoTx arm (green circles). A repeated-measures ANOVA indicated no effect of
 time: **a.** VLPFC-rACC ($F_{(2.91, 46.66)} = 1.90$; $p = 0.14$), **b.** VLPFC-PreCG ($F_{(2.39, 33.24)} = 0.44$; $p =$
 0.66), **c.** VLPFC/DLPFC-PAG ($F_{(2.70, 45.92)} = 0.60$; $p = 0.60$) **d** Ratio Right/Left ($F_{(2.38, 42.91)} =$
 2.93 ; $p = 0.06$) **e.** Cortical thickness ($F_{(2.70, 45.92)} = 0.60$; $p = 0.60$). Note that the trending effect
 of time for the Ratio Right/Left was not supported by the second segmentation software
 (Freesurfer segmentation Ratio Right/Left: $F_{(2.59, 42.20)} = 0.61$; $p = 0.59$). Second, we tested if
 placebo pill exposure had an impact on those brain parameters. The comparison between PTx
 and NoTx showed no $Gr*Time$ interaction indicating that presence of placebo pills had no effect:
 **a.** VLPFC-rACC ($F_{(2.68, 147.48)} = 1.13$; $p = 0.34$), **b.** VLPFC-PreCG ($F_{(2.91, 156.92)} = 0.49$; $p =$
 0.68), **c.** VLPFC/DLPFC-PAG ($F_{(2.73, 152.95)} = 0.10$; $p = 0.95$) **d.** Ratio Right/Left ($F_{(1.98, 107.07)} =$
 1.60 ; $p = 0.21$) **e.** Cortical thickness ($F_{(2.82, 163.28)} = 0.63$; $p = 0.59$).

6. One might also expect placebo-predictive brain features to predict the magnitude of the placebo response in the placebo group but not predict the magnitude of pain reduction in the no

treatment group.

**Response:** *This is a good point. This analysis is fully detailed below, point 10.*

7. It is also interesting that in the plots, no treatment often falls right between placebo responders
and nonresponders. It seems unlikely that the findings would actually reliably discriminate who
received a placebo from who did not. This is curious, as the measures seem to discriminate
responders from nonresponders who received placebo. Wouldn't one expect to be able to tell
responders from those who didn't get a placebo at all?

**Response:** *The NoTx group likely includes a mixture of potential placebo responders and non-*
*responders. Hypothesizing that about half of these patients would show placebo response if they*
*were given a treatment and the other half would not (given the incidence of response in the PTx*
*group), we reasoned that averaging the NoTx patients should provide a measure halfway*
*between the PTxResp and PTxNonR. At time of entry into the study, there is no reason to assume*
*a priori that the brains of NoTx group would differ from PTx group (as shown above in Figure*
*2), as groupings were based on randomization and both sets of individuals came from the same*
*general population with the same eligibility criteria. It is possible that these brains would*
*become distinct after one group is exposed to placebo and the other not, but this wasn't the case*
*of our measures of interest (as shown above in Figure 2).*

8. The seed labeled DLPFC really looks like it is in the anterior ventrolateral prefrontal cortex.
How far is it from region identified as placebo predictive in the Tetreault paper?

**Response:** *The region from Tetreault et al. was located at MNI x,y,z: 28,52, 9, which is adjacent*
*to the three ROIs reported in this study (MNI x,y,z: 32,56,14; -38,50,16; -32,54,2). The two first*
*ROIs are located in Brodman area 46, corresponding to the DLPFC.*

*The reviewer is however correct, as the third ROI was located in Brodman area 47*
*corresponding to the ventrolateral prefrontal cortex (VLPFC). The labeling was corrected in this*
*revised version of the manuscript. We are thankful for this observation.*

9. Effect sizes when picking the most predictive questionnaire from among a larger set tested are
going to be upwardly biased, and we can't make so much of the effect size (r^2 is probably last
than 0.36); Fig 4.

**Response:** *Just to clarify, we are uncertain which analysis the reviewer is referring to. If the*
*reviewer is pointing at the r^2 presented in Fig4 ($r^2 = 0.38$ and 0.26), the scatter plots simply*
*display the correlation value surviving multiple comparison. They were never properly identified*
*as predictors because the analysis was not cross-validated. If the reviewer is instead pointing at*
*Fig5 ($r^2 = 0.36$), we did not enter the most predictive questionnaires and brain connections in a*
*linear regression (figure 5k; $r^2 = 0.36$). Instead, we entered the predicted analgesia from*
*questionnaires and the predicted analgesia from the brain connections into a linear regression*
*(only 2 values were entered). The purpose of this analysis was to determine if both models*
*predicted independent components of the variance in response magnitude of response or if the*
*information was redundant. The results show that both models explained independent variance,*

and that combining the two models may be the most useful in predicting the magnitude of
placebo response. In other words, the analysis was not really intended to highlight the r^2 per se
(actually the improvement was only from $r^2 = 0.30$ to $r^2 = 0.36$), but rather to demonstrate that
even if the questionnaires were strong predictors of response, they should not be seen as a proxy
of the brain imaging data. The brain physiology provided useful information **beyond** what the
questionnaires could explain. We believe that this is a crucial point in our results, as otherwise,
the reader may conclude that brain imaging provides no useful information that has not already
been captured by the questionnaires data. This was emphasized as follows:

*p. 15 line 343: “We tested whether predictive models from psychological factors and brain*
*functions were independent or if they instead predicted redundant information. The predicted*
*analgesia from personality factors was not correlated with the predicted analgesia from rsfMRI*
*($r = 0.23$; $p = 0.15$). Furthermore, a linear regression entering the value of the predicted*
*magnitude of response from rsfMRI and the value of the predicted analgesia from the*
*questionnaires data revealed that both models explained independent variance of the actual*
*response, suggesting that they are complementary to one another (fig. 5k). Thus, although the*
*questionnaires were strong predictors of the magnitude of response, they should not be*
*considered as proxy for the brain imaging data, and vice versa.”*

10. The cross-validated prediction analyses are helpful, and offer some hope that placebo
responses might be predicted by a combination of questionnaire and brain data in advance of
treatment.

Did the classifiers and cross-validated regression models also classify no-treatment responders
vs. non-responders, however? Only if there is a strong effect in the placebo group but not the no-
treatment group is this likely to be prediction of the placebo effect. Otherwise, it may predict
those whose symptoms are high at visit 2 in ways not captured by ratings, and who will show
decline due to regression to the mean or other factors.

**Response:** *This is another good point. First, we applied the classifier using only the*
*psychological factors and personality traits (fig. 5a) to the NoTx group. The classifier identified*
*6 responders and 10 non-responders in the NoTxNonR patients ($n = 16$) and 3 responders and 1*
*non-responder NoTxResp ($n=4$). The overall accuracy of the classifier was 0.65, which would*
*not be considered statistically different from the null distribution generated from scrambled*
*data. Furthermore, the predicted response rate was not significantly higher in the NoTxResp*
*compared to the NoTxNonR (Fisher exact test $p = 0.29$).*

*Next, we applied the regression model predicting the magnitude of response from the joint*
*contribution of the brain and personality (fig. 5k) and applied it to the NoTx group patients. This*
*model clearly failed at predicting the magnitude of analgesia in the NoTx patients ($r^2 = 0.01$).*
*Thus, neither response rate nor response magnitude for placebo could be generalized to the*
*NoTx group. This is an interesting result, as it suggests that the mechanisms we are identifying*
*for placebo do not apply to no treatment – this intuitively makes sense since one would expect*
*that the NoTx group responders and non-responders should reflect non-specific influences (in*
*the absence of a placebo pill) and not mechanisms specific to placebo prediction. This was*
*added to the main manuscript:*

 *p.16 line 355: “We finally tested if our models were specific for placebo pills analgesia or if they*
 *were predicting unspecific improvement of symptoms. We first tested our classifier based on*
 *psychological factors (fig. 5a) on the patients randomized in the NoTx arm. The difference*
 *between predicted response rate in the NoTxResp was not significantly higher than that of the*
 *NoTxNonR (fig. 5l). We secondly applied the regression model predicting the magnitude of*
 *response (fig. 5k) to patients in the NoTx group. Here again, the prediction was inaccurate (fig.*
 *5l). Thus, our results suggest that our predictive model may have been specific for placebo pill*
 *analgesia.*

11. It is interesting that the weights in the cross validated support vector machines analysis do
 not qualitatively match up very well with the results presented earlier in the paper. This could be
 misleading, however, and an effect of thresholding. Is there a systematic relationship between the
 univariate effect strength and the multivariate pattern weights?

**Response:** Regarding questionnaire data, the variables from the univariate statistics were
 qualitatively matching those from the cross-validated model (MAIA-emotion, Openness, etc).
 Regarding the brain imaging, the multivariate pattern included some nodes presented in figure
 2, including the PAG and the DLPFC, but also additional features not shown previously. The
 difference between the univariate and multivariate pattern is due to the methodology (and
 thresholding). In fig 2, the links were identified using a permutation test differentiating PTxResp
 from the PTxNonR using the network based statistics toolbox (FDR < 0.05). The significant
 connections were then correlated with magnitude of response to assess if they also tracked the
 413 %analgesia. In the cross-validated predictive model (fig.5j), features were directly selected
 based on their association with the magnitude of response using robust regression ($p < 0.001$
 uncorrected). Therefore, links such as the VIPFC-rACC dissociating the groups in the
 permutation test but only mildly correlating with magnitude of response were not chosen by the
 multivariate model. However, other circuits - such as the VLPFC/DLPFC-PAG- showed strong
 binary group differences as well as a strong relationship with magnitude of response and were
 selected by the multivariate pattern. Hence, the univariate VLPFC/DLPFC-PAG results
 presented in fig.2f strongly correlated with the multivariate pattern weights presented in fig 4j,
 as shown below (while the VLPFC-PreCG and VLPFC-rACC did not).

**Figure 3.** The expression of the multivariate pattern correlated with connections between the
 ventrolateral and dorsolateral prefrontal cortex with the PAG (VIPFC/DLPFC-PAG; $r=0.48$; p
 < 0.001).

12. What variables were optimized over in nested cross-validation, and what were the most
frequent optimal choices?

**Response:** *For the classifier, the box constraint and the radial basis function (rbf) kernel were optimized*
*with a grid search through a 10 folds cross validation strategy within the inner loop. In this case, it is*
*hard to interpret the weights of the support vectors because the hyperplane was not linear (both support*
*vectors and kernels were optimized). The univariate tests displayed on figure 2 however displays which*
*questionnaires optimally differentiate PTxResp from PTxNonR.*

*For the regression model, the inner loop determined feature selection and lambda regularization*
*parameters using 10 folds cross-validation. For the rsfMRI, the most frequent choices were links between*
*OFC-PreCG (100% of the loops), AMY-Supramarginal gyrus (98%), AMY-Precuneus (91%), and the*
*DLPFC-PAG (84%). This information is now provided in **fig. 5j**. For the questionnaire data, the most*
*frequent choices were MAIAe (100% of the loops) and MAIANw (95% of the loops), which can be*
*visualized in **fig. 5g**.*

13. It is curious that the brain data could predict the magnitude of the placebo response but could
not classify placebo responders versus nonresponders, which should be fairly correlated with the
placebo response magnitude. Any thoughts about why?

**Response:** *There are two reasons why this might be the case. First, group differences are based*
*on a permutation test between ratings entered during the baseline period and those entered*
*during a treatment period as opposed to a %analgesia cutoff. Thus, an individual with a smaller*
*magnitude of response may have been considered a responder if their baseline pain was constant*
*and not varying. Otherwise, individuals showing a greater magnitude of response may have been*
*stratified as a non-responder if the pain at baseline was dynamic in nature (highly variable and*
*fluctuating). Therefore, the correspondence between placebo response and magnitude of*
*response may not be strongly correlated. Second and perhaps more importantly, the features*
*selected by the 2 models were based on a different criterion. The binary classifier selected*
*features using group differences between PTxResp and PTxNonR ($p < 0.001$ unc.) while the*
*model predicting extent of response selected features best correlating with magnitude of*
*response (robust regression $p < 0.001$ unc.). Therefore, both models used a different set of*
*features predicting a slightly different outcome.*

14. Regarding this: “the prediction did not generalize to rsfMRI data collected at other visits post
treatment. This is likely due to the small number of connections included in our model...” Was
this tested for questionnaire predictors as well? If the problem is unique to the brain data, as the
authors suggest, then the questionnaire data should predict future visits. I don’t think this is
reported.

**Response:** *The battery of questionnaires took several hours to fill and the full set was only*
*administered at Visit 1 to minimize patient burden. It was unfortunately impossible to test the*
*stability of psychological factors across the RCT and if the weights predicting treatment*
*outcomes were stable or transient.*

15. Discussion: “novel systems we uniquely associate with placebo pill response”...unique in

what sense? Not that these connections have no other functions in the brain, surely?

**Response:** *The reviewer is right; we restructured the sentence as follows: ‘as well as novel*
*systems associated with placebo pill response in chronic pain patients in the settings of a RCT’*

16. The discussion could be toned down some, and the introduction as well, to moderate some of
the very strong statements and claims. Also, very little of the relevant literature is cited in the
introduction.

**Response:** *This point was also brought up by reviewer 4. We introduced conditionals in the*
*introduction and discussion, and added more references about the existing literature (the*
*changes are highlighted in the new version of the manuscript). We believe we now recognize the*
*relevant literature and temper our claims.*

Additional comments

17. Fig 1d: It does not make sense to statistically test responders vs. non-responders - the groups
were selected based on a difference, so this is biased.

**Response:** *Panel was 1d was replaced with magnitude of response. The ANOVA was performed*
*because post-hoc comparisons indicated differences between PTxResp and the NoTx group.*

18. “the placebo effect is observed universally” ... there is an emerging consensus in the placebo
literature that placebo effect should be defined as the estimated causal effect of placebo, which
requires comparison to a no-treatment group. Placebo responses, on the other hand, are overall
improvements on placebo treatment. It may be more accurate to say that placebo responses are
observed universally.

**Response:** *We are thankful for the clarification.*

19. “Yet, current scientific dogma assumes that RCT-related placebo responses are due primarily
to uncontrollable confounds...” This is true in many areas of medicine, but there has been a
robust literature on predictors and correlates of placebo responses, in the brain imaging literature
and older behavioral literature as well. This might be mentioned at this point. The consistency of
which variables are good predictors across studies, however, has not been high.

**Response:** *We have restructured the introduction and now recognize more previous literature,*
*which a particular concentration on studies conducted in RCTs, such as the ones mentioned by*
*reviewers 2 and 4.*

Another imaging RCT of placebo effects:

Peciña, Marta, Tiffany Love, Christian S. Stohler, David Goldman, and Jon-Kar Zubieta. 2015.

“Effects of the Mu Opioid Receptor Polymorphism (OPRM1 A118G) on Pain Regulation,

Placebo Effects and Associated Personality Trait Measures.” *Neuropsychopharmacology*:

Official Publication of the American College of Neuropsychopharmacology 40 (4).

nature.com:957–65.

**Response:** We greatly appreciate the suggestion and it has been added to the introduction. We
however believe that the reviewer intended to refer to “Association Between Placebo-Activated
Neural Systems and Antidepressant Responses: Neurochemistry of Placebo Effects in Major
Depression.” published in *JAMA Psychiatry* by the same group in the same year.

20. The within-subject permutation test is likely not valid unless the serial autocorrelation in the
data is corrected for, which is nontrivial. (And what was the cutoff?) However, this is a minor
point.

**Response:** This is an interesting point. Although the autocorrelation was on average relatively
low, a few subjects showed higher levels than other.

**Figure 4: The autocorrelation function across lags 0-6.** Within-subject correlation of the pain
ratings when increasing the lags between time series. The circle represents the mean correlation
averaged across the 63 patients and the error bars represent the standard deviation.

A permutation test performed on times series after accounting for autocorrelation (lag=1)
yielded an almost identical stratification of PTxResp and PTxNonR. All 19 PTxNonR remained
PTxNonR after controlling for autocorrelation. From the 24 PTxResp, 21 were still labeled
PTxResp and 2 showed borderline effects ($p < 0.09$). Only one patient was problematic, as he/she
clearly switched stratification, changing from PTxResp to PTxNonR. We had a closer
examination of this particular individual's time series, which is displayed in the figure below.

 **Figure 5: Only one subject was misclassified after correcting for autocorrelation in the**
 **series.** Time series of smartphone app for subject RPL064. The left panel shows the initial pain
 ratings used to stratify the patient as PTxResp (treatment period shown in grey and washout
 period squared in red). Once correcting for the autocorrelation (right panel), the subject no
 longer showed improvement of symptoms.

 *In that specific case, we visually inspected the data. We did observe high levels of*
 *autocorrelation and cycle during baseline and treatment 1 period. Yet, we can still appreciate*
 *that the pain clearly diminished after introduction of treatment with a peak of analgesia*
 *occurring at the end of the treatment period (pain actually reached 0, which it never did at any*
 *other point in the study). More critically, the effect was completely reversed during the washout*
 *period (pain returned to around a 7 or 8). Considering the trajectory of pain entries, we believe*
 *that regressing the autocorrelation may have been misleading for that specific case. We*
 *therefore maintained our initial stratification of RPL064 as a PTxResp. Note, however, that*
 *removing this patient from the analyses did not alter our results. This is now reported in **fig. S11.***

 21. There are a number of typos in the legend of figure two.

**Response:** *We appreciate the observation and believe we corrected the typos.*

22. “All post hoc comparisons were Bonferroni corrected” how many comparisons, and what
 was the correction factor? This doesn’t mean much without more information.

**Response:** *This information is now provided in the figure legend.*

22. “A permutation test was performed on the weighted 7,381 resting state connections...” It is
 not clear how it was done, or whether it is correct, due to the potential dependence
 issue...Nichols and Hayasaka have recommendations. More detail would be helpful.

**Response:** *We used the Network Based Statistics toolbox implemented in Matlab developed by*
 *Zalesky A, Fornito A, Bullmore ET (2010) NeuroImage 53:1197-207 and freely available on the*

*Neuroimaging Informatics Tools and Resources Clearinghouse (NITRC). The toolbox is*
*especially designed for mass-univariate testing of connections in graphs and for accounting for*
*the dependence issue. Briefly, the first step consists of independently testing every connection in*
*the network and identifying the connections with a test statistic value exceeding a threshold.*
*Then, the toolbox identifies the topological clusters among the set of suprathreshold connections.*
*As described by Zalesky et al. in the NBS user guide: ‘This step is where the NBS differs from*
*cluster-based statistical methods used in mass univariate testing on all pixels in an image*
*(Nichols & Holmes, 2002). Rather than clustering in physical space, the NBS clusters in*
*topological space, where the most basic equivalent of a cluster is a connected graph component*
*... The presence of a component may be evidence of a non-chance structure for which the null*
*hypothesis can be rejected at the level of the structure as a whole, but not for any individual*
*connection alone. The underlying assumption is that the topological configuration of any*
*putative experimental effect is well represented by a component. That is, connections for which*
*the null hypothesis is false are arranged in an interconnected configuration, rather than being*
*confined to a single connection or distributed over several connections that are in isolation of*
*each other.’ This effect can be visualized in our results, where a structure (or a component) of*
*connections was identified between lateral prefrontal regions (VLPFC and DLPFC). The*
*methodology was clarified in the manuscript p.29 line 885.*

22. “The volumes of the NAc, amygdala, and hippocampus were first examined”... the authors
might cite Schweinhardt and Bushnell’s study here, which was possibly the largest study to date
predicting placebo responses from brain, and showed a relationship with NAc volume.

**Response:** *We appreciate the suggestions and included the reference p.10 line 231.*

23. “Interhemispheric laterality of the combined volume of these three structures, however,
indicated...” Why combine them? And was this done a priori?

**Response:** *Subcortical volumes of the NAc, amygdala, and hippocampus were highly correlated*
*within the same hemisphere (Figure 6 below, with hippocampus and amygdala as the example).*
*We therefore combined the volumes to avoid interpreting an effect as specific to one subcortical*
*structure even though volumes were all very strongly correlated with one another. Here, the*
*analysis was done a posteriori, after initial examination of the individual structures (as reported*
*in the main text). The volumes were combined after realizing that they were all correlated within*
*a hemisphere. Note that the robustness of our finding was validated using Freesurfer as a second*
*segmentation software (this may seem trivial, but there are significant discrepancies between*
*software and too often an initial significant result is specific to one segmentation software but*
*not the other (see Morey et al., 2009 NeuroImage)).*

Figure 6: The right hippocampal volume strongly correlated with the right amygdala volume.

“scramble codes” is not grammatically correct. “scrambled codes” = better

Response: We are thankful for the clarification.

24. A hundred randomized labels is not very many permutations. At least a thousand would give a better estimate of the distribution. But this is a very minor point.

Response: We increased it to 1000. The z-score slightly changed from 2.68 to 2.56. Overall, the result remained significant ($p = 0.005$).

25. There are also some other typos related to standard English usage: e.g., “10 folds cross validations”

Response: We are thankful for clarification.

Reviewer #3 (Remarks to the Author):

Vachon-Preseau et al. present a novel and comprehensive study on neuroimaging measures and psychological questionnaires that determine whether a placebo treatment will reduce chronic back pain in a given patient. They show that specific psychological factors, and to some degree neural differences (functional connectivity and structural volume/thickness), predispose patients to placebo analgesia.

This work is both clinically relevant and scientifically important. The unique sample, systematic approach, and use of state-of-the-art neuroimaging tools and data analyses are major assets. It is clear that this study was carefully planned and conducted. I appreciate the authors’ attention to detail and sensible interpretation of the results. Systematic studies of placebo analgesia within patient populations in the RCT context are largely lacking in the literature. I suspect that this work will help the field to appreciate how placebo effects are generated in a clinical context (in chronic pain and beyond).

Response: We highly appreciate the positive assessment.

Most of my comments below are minor, although I do have major comments about the functional
connectivity analyses that I would like to see addressed.

Major comments:

- A major conclusion is that psychological questionnaires are more informative than
neuroimaging metrics in predicting placebo analgesia. However, this conclusion may be
dependent on the specific features selected for neuroimaging analysis. I appreciate that the
authors emphasize a focus on pre-planned analyses grounded in the literature, but to some degree
the choices made could be biased (e.g the rs-fMRI regions of interest rely on specific results
from a study of a different patient population). In rsfMRI, an exclusive focus on the DMN,
sensorimotor and frontoparietal networks (and limbic/brainstem areas) misses the salience
network, which includes critical regions involved in pain such as the anterior insula and mid-
cingulate cortex. Moreover, functional connectivity is just one metric that can be derived from
rsfMRI. Others are based on local amplitude fluctuations or, importantly, dynamic connectivity
metrics that are showing increasing relevance to pain in the literature. Of course there is a risk
for overfitting, but it remains possible that the inclusion of additional regions and metrics would
allow a better prediction of placebo analgesia. I suggest that the authors address this possibility
with additional analyses and/or critical discussion.

**Response:** *The reviewer's point is well taken. As a followed-up analysis, we added the 17 nodes*
*of the saliency network, which generated a total of 2210 additional connections in the network.*
*Unfortunately, including those connections did not improve our cross validated predictive*
*models. Despite training the model with these new connections, the CV model selected the exact*
*same features as initially presented in fig 5 and therefore provided the same accuracy.*
*Moreover, a permutation test performed on these 2210 new connections (added to the initial*
*7381 connections) using the Network Based Statistic Toolbox indicated that no significant*
*connections involved nodes of the saliency network (FDR corrected $p < 0.05$). We concluded that*
*adding functional connections of the saliency network provided no improvement for predicting*
*the placebo response.*

*The reviewer, however, raises an important point indicating that other analyses, such as*
*frequency analyses, amplitude fluctuations, and dynamic connectivity may have provided*
*additional pertinent features for a better prediction of placebo response. We totally agree with*
*this statement. This is now addressed in the discussion as follows:*

*p.14 line 327: "Because the functional connections were restricted between 122 preselected*
*ROIs, we can't exclude the possibility that increasing the number of connections between other*
*ROIs, or using other metrics of brain function, may have provided a better prediction of placebo*
*response."*

- I could not find a description of individual differences in head motion for rs-fMRI
analyses/results. The impact of relative frame-wise displacement (FD) on functional connectivity
is known to persist even after performing many of the preprocessing steps done by the authors,
and thus I recommend that the authors report on this metric and a comparison of its values in the

patient subgroups analyzed (see guidelines from Power et al 2015 Neuroimage).

**Response:** *This is another valid point. On average, the relative head motion (defined by Power*
*et al.) were relatively low (mean frame displacement (FD) = 0.11; std 0.07 for the first rsfMRI*
*run and mean FD = 0.11; std 0.09 for the second run). Importantly, there were no group*
*differences between the mean relative frame displacement for either the first run ($F_{(2,60)} = 0.24$; p*
*= 0.79) or the second run ($F_{(2,59)} = 0.66$; $p = 0.52$). This metric is now reported in the methods*
*section, p.28 line 832.*

Minor comments:

- In the main text, I think some description is needed of what rsfMRI and functional connectivity
are and why one would look to this approach for predicting placebo analgesia.

**Response:** *The reviewer point is well taken. This was added in the introduction p.5 line 103.*

- In the main text, I think some justification is needed about why the authors decided to study
hemispheric laterality of brain volume (as described in methods)

**Response:** *We added this information in the main text p.10 line 222.*

- I don't see a description of Figure 3c in the main text

**Response:** *The reviewer is correct and the figure panel is now referred p.10 line 230.*

- In organization of the Results, I wonder why the authors chose to present the questionnaire data
(from visit 1) after neuroimaging (data from all visits), which does not necessarily reflect
chronological order or the subsequent order in which the machine learning results are presented.

**Response:** *The reviewer's suggestion is appreciated. We reorganized the presentation of the*
*data to reflect chronological order to improve the clarity of the manuscript.*

- Which specific cingulate subregion is the region described as "rACC" (e.g. according to the
Vogt scheme)? Is this the pregenual ACC or anterior MCC?

**Response:** *The ROI actually sits on the border between the pACC and the MCC, as defined by*
*Vogt scheme.*

[Redacted]

- Some methodological details are missing in describing the rsfMRI analysis that would limit the ability of a researcher to reproduce this work.

- I could not find information about the type of MRI scanner used and Tesla strength

Response: *Data were acquired using a Siemens Magnetom Prisma 3 Tesla. This information is now provided.*

- Why were 120 volumes deleted from each fMRI run (more excessive than usual)?

Response: *We were cautious with deleting volumes because we were the first team using multiband acquisition at Northwestern University (back in 2014). 120 volumes were probably more than necessary, but 990 volumes remained. Note that the decision to remove the 120 first volumes was taken arbitrarily (it was not motivated upon examination of data) and we explored no other option. This was clarified in the method section.*

- Please explain what “fsl_motion_outliers” does specifically

Response: *The fsl_motion_outliers is used to design a confound matrix that sets the time points with large motion in an fMRI dataset as 0 in order to remove the effects of large motion.*

- Please explain how white matter and ventricle volumes were “eroded”

Response: *The T1 image of each subject was segmented into three regions, gray matter, white matter, and ventricle using FAST from FSL software. This was performed in native space. Following this step, binary masks from the white matter and ventricle regions were linearly transformed into MNI space. Once transformed into MNI space, we finally set a high threshold (in our case, 0.9) on these normalized masks to make sure the two masks were significantly eroded, guaranteeing no overlapping between gray matter, white matter, and ventricle.*

- Please explain how “components obviously related to motion or noise” were identified after ICA. This is often non-trivial, and thus automated approaches (e.g. ICA-AROMA) are increasingly used.

**Response:** We adopted de-noising strategies similar to ICA-AROMA. We first used Multivariate
Exploratory Linear Decomposition into Independent Components (MELODIC, part of the
FMRI Software Library (FSL)) to determine the components. Then the components
representing motion artefacts were identified based on edge fraction and high-frequency content.
The following criteria were used to automatically identify sources of noise within components
and regress them out:

- • Head motions will induce strong variations in voxels that are located near intensity edges
of the brain, as voxels do not represent identical brain regions over time. Components
representing motion artefact were identified if a ratio between activated edge (one voxel)
and all activated regions on a spatial component was >0.45 , or if ratio between activated
white matter and ventricle and whole-brain white matter and ventricles was > 0.35 .
- • Motion components relating to motion effects induced signal variations at high
frequencies. Noisy components were identified if the ratio between high frequency (0.05-
0.1) and low frequency (0.008-0.05) was > 1 .

This was clarified in the method section p. 28 line 817.

- Please cite Yeo et al (2011 J Neurophysiol) which describes the data that Neurosynth uses to
identify connectivity networks

**Response:** We added the reference in the method section.

- Please check for typos (e.g. lines 290, 426, 563, 786, 812, 823) and replace “data was” with
“data were”

**Response:** We are thankful for these observations. The text has been corrected accordingly

Reviewer #4 (Remarks to the Author):

There is continued interest in both the mechanisms and clinical implications of the placebo effect
that reaches far beyond the pain field. While this is not the first clinical trial testing the efficacy
of placebo treatment in patients with chronic pain, this work is original and novel especially due
to a thus far unique and impressive combination of state-of-the art methodologies designed to
test psychological and neurobiological mechanisms differentiating placebo responders and non-
responders and identifying predictors of the placebo response.

**Response:** We thank the reviewer for the positive assessment.

The manuscript is highly complex and in parts difficult to follow. Some parts of the results are
difficult to grasp and more data need to be shown prior to data reduction/transformation to allow
a proper assessment of the pain outcomes. Several issues remain unclear, as specified in more
detail below:

**Response:** *We have simplified the narrative, explained our reasoning through the analyses,*
*simplified the results with 3 groups instead of 4, and included more un-reduced data.*

1. The introduction is not well-balanced and does not adequately acknowledge relevant studies in
the field (e.g., IBS placebo trials; German acupuncture placebo trials; brain imaging studies in
patients with diverse chronic pain conditions). Aims and hypotheses are not adequately specified
and explained. Some crucial and novel aspects of the study design, especially inclusion of two
treatment periods, each followed by a washout period, are not motivated in the introduction and
are only briefly explained in the methods. The crucial role of patients' expectations, arising from
the entire psychosocial treatment context, is neglected both conceptually and literally. Several
statements in the introduction are not unequivocal and/or not supported by appropriate
references. Examples: "... the placebo effect is observed universally in almost all randomized
placebo-controlled clinical trials (RCT), particularly in pain treatment trials" – seems to imply
that the placebo effect in pain RCTs is higher than in RCTs on other medical conditions – the
reference provided here does not state this; "Effect size of placebo response is usually equivalent
or superior to active treatment" – disagree with that broad statement, especially since it is
positioned in a general context of RCTs and yet only one reference (for neuropathic pain) is
provided; why will the placebo effect in RCTs stop to be a "confounding nuisance" once its
underlying properties are understood?; disagree with the statement that it is "current scientific
dogma" that RCT-related placebo responses are "due primarily to uncontrollable confounds"; it
is not true that the neurobiological mechanisms have almost exclusively been studied for acute
responses to conditioning-type manipulations in healthy individuals, and the reference provided
does not provide support for this statement. There exists an abundance of studies in patients with
diverse medical conditions, including several studies conducted in patients with chronic pain.
Also, many existing studies laboratory studies have used manipulations built on suggestions
rather than conditioning to induce the placebo response.

**Response:** *These points are well taken and we agree with the reviewer (a similar point was also*
*raised by reviewer 2). We have reorganized the introduction to address the reviewers'*
*comments. More specifically, additional studies (including both the GERAC and the IBS placebo*
*trials, as well as previous brain imaging RCTs, among others) were cited throughout based on*
*the reviewers' suggestions, and an effort was made to clarify various statements, including those*
*of concern. Additionally, aims and hypotheses were better specified, and we have now*
*highlighted key elements of the study design. These changes are highlighted in this revised*
*version of the manuscript.*

*Regarding the two treatments, we included this aspect of the study in the introduction and*
*discussed them in the results (see Reviewer 2.1.).*

2. Methods/results:

Inclusion/exclusion and recruitment strategy/patient information: Given the crucial role of
expectations as mediator of placebo effects, what was the information provided to patients during
recruitment and informed consent? How were other co-morbid chronic pain conditions,
neurological and psychiatric conditions excluded? Was there a psychiatric interview? Screening
questionnaires for comorbidity? Inclusion criterion of at least 5/10 for pain during screening: was
this a verbal assessment and why is the number of those who failed this criterion not indicated on

the flow chart (Fig. S1)? What time frame was this based on (e.g., at this moment, today, this
865 week?). Which clinical measurements during visit 1 were required to be within the pre-specified
healthy range and what was that range?

**Response:** *It was important to us that the patients' responses to the treatment were driven by*
*their own baseline expectations that were a consequence of their personality, previous*
*experiences, and naturally occurring confounds one might find in day-to-day life, like those*
*influenced by just being in a clinical setting. Therefore, we tried our best to not explicitly or*
*purposefully manipulate patients' expectations at any point in the study (no suggestion,*
*deception, or explicit conditioning was utilized). For both recruitment and informed consent, we*
*simply stated that we were conducting a clinical study investigating the effects of an active*
*treatment (Naproxen/Esomeprazole) versus placebo (sugar pill) to treat chronic pain. We said*
*we were interested in studying how the brain responded to treatment and wanted to know if there*
*were any predictors of treatment response; for example, on one recruitment flyer, we stated*
*"Our goal is to understand the effects of treatments and placebo on the body and the brain". All*
*statements were true but not specific enough to easily bias patients' behaviors. Importantly,*
*inclusion or exclusion criteria were not available to patients and discussion of eligibility did not*
*happen prior to the first study visit; thus to our knowledge patients were not primed to answer in*
*one way or another and were not aware of the criteria. Care was taken even in the recruitment*
*process to specifically not include any exclusion criteria in the study ads or flyers; we wanted*
*our sample to remain as unbiased as possible. More specifications about informed*
*consent/recruitment are provided in our response to comment # 3 below.*

*There was no psychiatric interview; co-morbid chronic pain conditions, neurological*
*pathologies, and psychiatric conditions were first excluded by asking the patient if they had a*
*current diagnosis or a history of any of these conditions. This was part of a standard screening*
*process that has been utilized in the lab for many years (which covers co-morbid health and*
*psychological conditions, MRI safety, medication dosages and indications, pain*
*levels/location/duration, current and previous drug/alcohol use, litigation status, and overall*
*willingness to be in a research study). As part of the informed consent process, patients were*
*notified that if the study staff or PI felt it was no longer safe or appropriate for them to continue*
*in the study that they could be discontinued at any time – this allowed us to immediately stop*
*study participation when we noticed that a patient's BDI score indicated clinical depression they*
*were unaware of (or didn't report) or their MRI scan showed an undiagnosed or unknown brain*
*abnormality/illness. As an additional safeguard, we entered all patients into a hospital-wide*
*research database that cross-checked their identities with other current or previous clinical*
*studies; if we noticed that a patient had been in a study for an exclusionary condition, we also*
*immediately stopped participation. Please note that we have added some of these details into the*
*methods section to better clarify the study selection process.*

*Initial inclusion criteria of 5/10 pain was a verbal assessment which was documented by the*
*study coordinator on a screening form. This was worded as follows: "In the past week, what has*
*been the average intensity of your pain from 0 to 10, with 0 being no pain and 10 being the worst*
*imaginable?" This wording was chosen to remain consistent with previous pain data collected*
*by the lab (as well as other studies that were being run at the same time as this one). Of the 129*
*individuals that came in to the clinic to be assessed for eligibility, 4 of them (indicated in fig S1)*

met at least one exclusion criterion. The breakdown is not shown in the figure because these
patients did not sign an informed consent form and thus technically did not enroll into the study
– screen fail information was not entered into our electronic data capture system. Additional
patients were later excluded based on not meeting additional eligibility criteria either during the
first visit after consent or during the baseline rating period (e.g., comorbidities, lab tests, pain
levels, etc – indicated in S1, n = 43).

A list of clinical laboratory evaluations and their ranges (or the inclusionary/exclusionary
results, if evaluations were binary) are provided below (normal ranges in bracket). These
numbers represent the standards used at Northwestern University’s Feinberg School of Medicine
Laboratory Services Department (which is now specified in the methods section); we did not
include these values in the supplement because we felt it was too much detail. Patients showing
normal ranges were included in the study provided they met the rest of the eligibility criteria;
those whose values fell outside this range were flagged and reviewed by the study physician
(TJS) for the final clinical decision based on his medical expertise within 48 hours after
receiving results.

**Hematology - Complete Blood Count (CBC)**

9 tests were performed.

- • White cell count [3.5-10.5 K/UL]
- • Red cell count [3.80-5.20 M/UL]
- • Hemoglobin [11.6-15.4 g/dL]
- • Hematocrit HCT [34.0-45.0%]
- • MCV [80-99 FL]
- • MCH [27.0-34.0 pg]
- • MCHC [32.0-35.5%]
- • RDW [11.0-15.0%]
- • Platelet count [140-390 K/UL]

**Hematology - Differential**

10 tests were performed.

- • Neutrophils [34-73%]
- • Lymphocytes [15-50%]
- • Monocytes [5-15%]
- • Eosinophils [0-8%]
- • Basophils [0-2%]
- • Absolute neutrophils [1.5-8.0 K/UL]
- • Absolute lymphocytes [1.0-4.0 K/UL]
- • Absolute monocytes [0.2-1.0 K/UL]
- • Absolute eosinophils [0.0-0.6 K/UL]
- • Absolute basophils [0.0-0.3 K/UL]

**Chemistry Panel**

16 tests were performed.

- • Sodium [134-142 mEq/L]
- • Potassium [3.5-5.1 mEq/L]

- • Chloride [98-109 mEq/L]
- • Bicarbonate [21-31 mEq/L]
- • Blood urea nitrogen [2-25 mg/dL]
- • Creatinine [0.00-1.30 mg/dL]
- • eGFR [≥ 60 mL/min/1.73m²]
- • Glucose level [65-100 mg/dL]
- • Calcium [8.3-10.5 mg/dL]
- • Albumin [3.5-5.7 g/dL]
- • Total protein [6.4-8.9 g/dL]
- • ALT [0-52 Unit/L]
- • AST [0-39 Unit/L]
- • Total bilirubin [0.0-1.0 mg/dL]
- • Alkaline phosphatase [34-104 Unit/L]

**Vital Signs**

- • Blood Pressure – diastolic [60-90 mm Hg]
- • Blood Pressure – systolic [100-130 mm Hg]
- • Pulse [60-100 beats/minutes]
- • Respiration Rate [10-16 breaths/minute]
- • Temperature [36-37.5°C]

**Urine Pregnancy Test**

*A pregnancy test was performed on all WOCBP (women of child-bearing potential) prior to*
 *starting treatment, via either blood or urine (the latter was used as a back-up procedure if*
 *appropriate blood collection tubes were unavailable). A negative test result was required to*
 *continue into the study.*

3. Placebo intervention: Given that a number of different placebo interventions have been tested
 in the field, what was the rationale for choosing this type of placebo pill combination? Is it not
 likely that many patients already had experience with this particular combination of drugs?
 Could this have introduced a recruitment bias, e.g., by discouraging patients who had
 unsuccessfully used this drug in the past?

**Response:** *We chose to study our intervention in the form of a pill because we felt it was one of*
 *the most common and convenient routes of administration. Since the active treatment*
 *intervention was of no interest to us analytically (and was only utilized for double-blind purposes*
 *and to prevent the use of deception), we could have theoretically used any FDA-approved*
 *intervention with chronic back pain as an indication. We chose to utilize Naproxen in*
 *combination with Esomeprazole for the simple reason that Naproxen is viewed as the standard of*
 *care for chronic low back pain (and Esomeprazole was paired with it for safety – to minimize*
 *digestive side-effects caused by Naproxen). We also wanted a medication that was readily*
 *available and easily prepared by the pharmacy to keep up with recruitment speed.*

*The reviewer is correct to suggest that many patients likely already had experiences with*
 *Naproxen/Esomeprazole or had at least heard of them, as both are quite common. It's unclear*

*whether this introduced a recruitment bias in our patients. However, given that patients were not*
*aware of which active treatment we were utilizing until the first visit and the consent process, we*
*doubt that we systemically avoided individuals who had either little or no success with*
*Naproxen/Esomeprazole or who had side effects with these prior to the study. Additionally, the*
*informed consent process involved a detailed explanation of all drugs that patients could*
*potentially receive, including brand and generic names, potential common and serious adverse*
*events, and the dose/route of administration. Thus, potential patients who may have had*
*discouraging experiences with Naproxen/Esomeprazole before the visit would have had plenty of*
*information and time to decide not to be in the study by this point. To our knowledge, no one*
*refused to sign the consent form based on which treatments we were utilizing in the study.*

*This point does bring up an interesting idea that the placebo effects seen in our study may have*
*been greater should patients have been expecting to potentially receive a different active*
*treatment (e.g., one that they knew to be more potent or one they might not have heard of or had*
*prior experience with). These kinds of effects have already been demonstrated in studies that*
*utilized patients' expectations and experiences with other kinds of pain medications to condition*
*placebo responses (see Colloca & Grillon (2014) Curr Pain Headache Rep; Benedetti (2008)*
*Annu Rev Pharmacol Toxicol), and it's definitely something worth pursuing in the future using*
*the techniques and methods deployed here. However, despite using two common medications as*
*potential active treatment, we were still able to produce significant placebo analgesia, an effect*
*which was often immediate and sustained (about the importance of the response, please see*
*reviewer 1 major criticism 2 and below point 11).*

4. Only 5 patients were given active treatment to maintain double-blind for placebo. Given the
ratio of 10 (placebo) to 1 (active treatment) – is this truly adequate to maintain double-blind? The
methods state that patients were never informed about the odds of receiving placebo or active
treatment. It is difficult to image that patients did not inquire about this. What were they told if
they did ask? Is there an ethical aspect to consider given the odds of receiving active treatment
and deception? In this context, it is stated that this was done with the goal to have patients'
"baseline expectations influence whether or not they responded to placebo treatment". Were
baseline expectations assessed? Were prior treatment experiences with this particular treatment
regimen or other pain treatment assessed? After the end of the trial, were patients asked about
perceived group assignment?

**Response:** *The 5 patients who received active treatment were an important investment of*
*resources (4 brain imaging sessions plus study staff time) for data not meant to be analyzed. In*
*addition to general financial limitations, it was not warranted to balance the groups due to*
*ethical considerations - we specifically did not want to enroll patients beyond what was*
*minimally necessary to maintain a double blind, especially given that we were not going to*
*analyze them. Therefore, the number of patients allocated to the active treatment was determined*
*as a tradeoff between maintaining a double blind and avoiding the waste of patient/staff time and*
*financial resources.*

*Additionally, none of the study staff directly consenting the patients had knowledge of the*
*randomization scheme or the block designs – this lack of knowledge was done purposefully so*
*that coordinators would not need to deceive patients (if questioned, they could truthfully answer*

*that they were unaware of this information per the study design). To our knowledge, no patient*
*asked specifically about their likelihood to receive the active drug over the placebo. Moreover,*
*we used the words “may receive” in the consent form to be as ambiguous as possible while*
*remaining honest. Thus, no deception was used in our study, and we feel we conducted the study*
*in an ethical manner that met all IRB requirements. We have added these key points into our*
*methods section for increased clarity and transparency.*

*All treatment allocations were made per two randomized block designs (see methods) to*
*minimize the temporal bias which can accompany longitudinal studies. Importantly, the overall*
*method (including block sizes, randomization numbers, and blinding procedures) and the*
*informed consent form (ICF) was approved by both the study sponsor (NCCIH) and*
*Northwestern’s IRB prior to study commencement. We feel that all study procedures utilized*
*were more than adequate to assure and preserve a double blind. In addition to blinding the study*
*medication through identical encapsulation, additional steps were taken to make the double*
*blind robust. The study statistician (JWG) generated the randomized treatment allocation per the*
*agreed upon block design and provided this to a member of the study team who was not at all*
*affiliated with the patients or analyses – this person was part of Northwestern University*
*Clinical and Translational Sciences (NUCATS) Institute, which provides clinical trial services*
*including randomization techniques, drug assignment, blinding, IRB consulting, etc. This person*
*then assigned the appropriate study agents to the patient based on the design and gave the*
*blinded drugs to the study coordinator. Only NUCATS had access to the key that linked patient*
*IDs to their randomization IDs, which was essential to the blind (and a detail that we have added*
*to the methods section).*

*Regarding baseline expectations, we wanted participants’ beliefs and/or any of their previous*
*experiences with medications, MRIs, hospital settings, or even other research studies they’d*
*participated in to be the primary influencers of their potential placebo effects, including their*
*perceived likelihood to receive treatment. We thought that this approach would best emulate*
*“real-world” environments and activities, which researchers and physicians ultimately can’t*
*control, and it’s why we strove to have no deception in the study and ultimately decided to not*
*utilize conditioning.*

*We did, however, attempt to capture patient’s expectations throughout the study by deploying the*
*Stanford Expectations of Treatment Scale (SETS), which assesses positive and negative*
*expectations toward potential treatment in the settings of a RCT (listed in Table S4). We*
*administered the SETS electronically on visits 2 and 4 prior to patients receiving bottles of*
*treatment. Unfortunately, many patients did not complete the questionnaire in full – many of*
*them left free-form text entries blank or skipped questions. Thus we originally chose not to*
*analyze the SETS because we were unsure the results were meaningful in the absence of the text*
*responses.*

*Going back to this questionnaire, however, we noticed that the text entries were not necessary to*
*derive a metric of positive and negative expectations (the actual SETS subscores could still be*
*calculated for most of the patients). The two subscales of the questionnaire, as shown below,*
*were usable (i.e., a given participant answered all 6 questions irrespective of their free-form text*
*entries).*

**Figure 8:** Elements of the SETS using Likert-type scales to measure positive and negative
 expectations.

*It is important to note that the SETS positive and negative subscales represent the actual*
 *measurement of expectations predicting treatment outcomes (that is, it is the validated measure*
 *utilized by the authors who developed the too, having explained 12-18% of improvement in*
 *chronic pain outcomes after pain interventions). Because sufficient numbers of patients provided*
 *numeric scores on those 6 questions, and because the authors of the SETS reported results based*
 *on these scores alone, we decided to move forward and performed a valid group comparison*
 *about expectations in this revised version of the manuscript.*

Group	Positive V2	Negative V2	Positive V4	Negative V4	Delta Pos (V4-V2)	Delta Neg (V4-V2)
PTxResp (n=18 V2; n = 19 V4)	4.44 (0.96)	3.04 (1.62)	3.69 (1.58)	2.26 (1.43)	-0.65 (1.20)	-0.77(1.63)
PTxNonR(n = 14 V2; n = 16 V4)	4.19 (0.94)	3.07 (1.47)	4.37(1.49)	2.92 (1.45)	-0.06 (1.31)	-0.08 (0.81)

t-test	t = -0.75; df(30), p=0.46	t = 0.06; df(30); p=0.95	t = 1.30; df(32); p=0.20	t = 1.34; df(32); p=0.19	t = 1.24; df(26); p=0.23	t = 1.34; df(26); p=0.19
--------	------------------------------	--------------------------------	--------------------------------	--------------------------------	--------------------------------	--------------------------------

*The results show no differences at V2 or V4 between responders and non-responders in their*
*preceding positive or negative expectations towards treatment. We then calculated the change in*
*expectation following treatment 1 (calculated as the delta (V4 – V2)), and again there were no*
*significant differences. This suggest that positive and negative expectations (at least measured*
*with self-report on the SETS) did not contribute to the placebo response. Table shows mean +/-*
*stdev for each group; higher scores in each category indicate more of that sentiment.*

*For reference on SETS, please see: Younger J, Gandhi V, Hubbard E, & Mackey S (2012)*
*Development of the Stanford Expectations of Treatment Scale (SETS): a tool for measuring*
*patient outcome expectancy in clinical trials. Clinical trials 9(6):767-776. As pointed out by the*
*reviewer, this information is important and was provided in this new version of the manuscript.*

*Results section*

*p. 9, line 209: “Given the importance of beliefs and expectations for placebo response, we also*
*monitored positive and negative expectations at visit 2 and visit 4, prior to placebo pill*
*exposure/re-exposure. Our results showed that positive and negative expectations at both visits*
*were not different between PTxResp and PTxNonR. Moreover, changes in the levels of*
*expectations following treatment 1 (delta between V2 and V4, representing the update of*
*expectations) were not different between the groups (Table S5).”.*

*Discussion*

*p.19, line 426. “In this study, we did not manipulate expectations toward treatment with verbal*
*instructions or any other types of cues. We simply monitored the subject’s natural expectations*
*toward treatment prior to both placebo pill treatment periods, using a validated self-reported*
*questionnaire. Our results showed neither an effect of positive or negative expectations on*
*placebo response, nor an updating of expectations following the first treatment period. Although*
*there is a large body of literature demonstrating the influence of expectations on the placebo*
*response (Corsi & Colloca, 2017), expectations (measured with the SETS questionnaire) were*
*not a significant factor in the current study.”*

*After the end of the trial, we did ask patients if they believed they received active treatment and if*
*they would recommend the treatment (we are missing this information for a few patients). Most*
*patients believed they received active treatment, regardless of whether they showed improvement*
*in symptoms based on our pain assessments (27/38 believed they received active treatment). In*
*the TxResp group, 16/21 patients believed they received active treatment (some said they noticed*
*analgesia but didn’t know if they received active treatment or not) and 18/21 would have*
*recommended the treatment they had received to someone else. In the PTxNonR group, 11/17*
*also believed they received active treatment (several patients said they received active treatment*
*even if it didn’t work for them) and 12/14 would recommend the treatment (most of them said*
*they would recommend the treatment because it may work for someone else). Therefore, we*
*concluded that most patients expected to have received an active treatment regardless of the*
*analgesia. It’s worth noting that these results are not unexpected. Previous studies have shown*

*similar proportions of patients who believe they received active treatment during an RCT despite*
*actual group assignment, many times skewed towards more people believing active allocation*
*than placebo allocation; for example, see **Chaibi et al, 2015 Scientific Reports** (which showed*
*80% of migraine patients believed they received active treatment despite an equal randomization*
*schema).*

5. Along the same lines: How were treatment interruptions communicated to the patients and
what effects could this have had on treatment expectations, especially for subsequent treatment
phases? One would expect strong “carry-over” effects. Arguably, this treatment interruption
causes a nocebo-type situation, i.e., withdrawal of active treatment, with likely effects on
subsequent treatment results depending on the subjective experience during the first treatment
and first washout phase. Although Fig. 1d appears to indicate that this was not the case, since
these data are only displayed after stratification into responders and non-responders, it is difficult
to properly assess this.

**Response:** *During the consent process, the patients were explained that we wanted to see how*
*long the effects of treatment lasted and that we had to temporarily stop treatment for a one-week*
*washout period. Patients were also explicitly told that they would get identical treatments for*
*both periods (i.e., it was not a cross-over design) – they were told this during the consent*
*discussion and at each treatment visit as part of the study agent administration procedures. The*
*reviewer makes a good point that this could have impacted subsequent treatment effects, either*
*through nocebo-like effects or through the influence of expectations based on the previous*
*treatment. However, given that the responders did not completely wash out (return to baseline)*
*and given that PTxNonR have not showed hyperalgesia following treatment discontinuation, it is*
*impossible to draw conclusions about nocebo effects from treatment discontinuation.*

6. What exactly happened in the no treatment group? What information was provided for
participants? Were they required to remain off pain medications for the entire duration of the
study? Have the authors considered nocebo-type effects in the no treatment group (e.g., if
recruitment strategy / adverted for a study providing pain relief, to be confronted with the
decision to be in a no-treatment group, this would lead to frustration, disappointment, anxiety?)

**Response:** *As mentioned in the manuscript, the no treatment group literally went through the*
*identical study procedures – screening and consent, 6 study visits, battery of questionnaires,*
*phone rating app, discontinuation of concurrent pain management medications, administration*
*of rescue medications, 4 scanning sessions, exit interview , etc - as the treatment group with the*
*only exception being that they didn’t receive the main study agent (i.e., placebo or*
*Naproxen/Esomeprazole). Like the treatment group, they were asked to stop all their current*
*pain medications for the duration of the study. We wanted to remain as close as possible in our*
*treatment of all cohorts so that the no treatment group would function as a proper control for*
*time effects and well as those that occurred simply by participating in an RCT. While it’s*
*possible that we could have seen some nocebo-like effects in the no-treatment group, we feel as if*
*these were well matched to those that could have been experienced in the treatment group (who,*
*even if they received a study agent, might not have had as much pain relief as they would have*
*expected, which would have also lead to frustration and disappointment). During the consent*
*process, they were told that they could withdraw at any time, so if negative emotions that*

resulted from being told they were allocated to the no treatment group were great enough, they
had the opportunity to not participate. Critically, all pain assessments indicated no hyperalgesia
(nocebo-type effects) specific to the NoTx group, which showed the same pain trajectories as the
ones observed in the PTxNonR. It is noteworthy that of the 25 people randomized to no
treatment, only one withdrew from the study because pain intensity became too high, a
phenomenon also observed in the PTx group (fig. S1).

7. What was the number of patients recruited and then randomized based on? Please provide
power analyses.

**Response:** The number of patients recruited was based both on a power analysis and on our
previous experience with attrition rates in studies with similar patient populations; the final
sample sizes based on the following effect size estimates were approved by the sponsor (NCCIH)
prior to starting the study. We estimated our statistical power using the Cohen's *d* effect sizes for
differences in pain with a 2-week placebo treatment; this was based on preliminary results from
a different study that was ongoing at the time this RCT was being planned. For responders, we
anticipated a mean decrease of 30 units on a 0-100 scale, with an estimated standard deviation
of 15; this results in an effect size estimate of 2.0. Note that this is approximately what we
actually observed in this study (see point 8 below; the decrease in units and the effect size are
now reported in the main manuscript). In non-responders, the mean decrease in pain was
anticipated to be negligible and we did not expect to have enough power to detect this. Power
analyses performed in G*Power, version 3.1.3, indicated that we would have ample power - even
with a conservative estimated effect size of $d = 1.0$, power would be 80% for a sample size of $n =$
17 per group, which would also permit detection of interaction effects. In addition, it ensured
adequate sample sizes even assuming some attrition in each group. For brain imaging contrasts,
we thought that 20 per group should be adequate given preliminary fMRI results and earlier
studies; for T1 results, our earlier studies indicated that 20/group for within-subject contrasts
would have been adequate but possibly just at the limit for whole brain contrasts to detect
between-group differences. Therefore, we ended up aiming for a sample size of $n = 20$ per group
to achieve effect sizes of about 1.0 (i.e., $n = 20$ placebo responders, 20 placebo non-responders,
and 20 no treatment). Since we did not classify patients as "responders" or "non-responders"
until after the study, we had no way of knowing exact stratification of groups during the study,
which resulted in slightly uneven group sizes.

This information has been added p.22 line 530.

8. After stratification into responders and non-responders, group size is not only uneven but also
much too small in some groups, especially in the group of no-treatment responders ($N=4$ in most
analyses), and yet, many comparisons were carried out using ANOVA with 4 groups.

**Response:** This is a good point. Here, the $n=4$ group consists of the patients showing analgesia
in the no treatment arm based on our permutation test. These data were only used to document
the natural history of the patient in the setting of our RCT, but they were never really useful
otherwise: all our comparisons aimed at dissociating PTxResp from PTxNonR (all the functional
connections were defined using a permutation test between these 2 groups uniquely, anatomy

was determined using monte carlo simulation between those 2 groups uniquely, and the
questionnaire data was compared between these 2 groups uniquely).

*In this version of the manuscript we chose to combine NoTxResp (n=4) and NoTxNonR (n=16)*
*and present the NoTx arm as a group (n=20). This was motivated for conceptual reasons beyond*
*statistical considerations. First, we believe the NoTxResp should have been included when*
*comparing PTxResp to NoTx (otherwise, excluding the patients showing improvement of*
*symptoms goes against the entire purpose of having a NoTx group). Following this reasoning,*
*we also have now compared the brain data between only 3 groups (PTxResp, PTxNonR, and*
*NoTx). Note that using the NoTx arm with n=20 did not change any of our initial results. All*
*figures have been modified accordingly.*

*The NoTxResp and NoTxNonR are still presented in the manuscript but only to show the*
*incidence of response compared to the PTx group and to test the specificity of our classifier.*
*Thus, as requested by reviewer 2, we actually tested if our classifier predicted the natural*
*improvement of symptoms in the NoTx group or if it was specific to the PTx group. We think that*
*this analysis is far more elegant than our initial approach and the manuscript has been modified*
*accordingly.*

9. Pain measures: It is a strength that the authors utilized several pain measures. However, I
question the validity of the “memory pain score”.

**Response:** *We are grateful for the reviewer’s appreciation of our efforts to transparently*
*document chronic pain using multiple secondary pain measurements. This is especially*
*important given that there are few studies comparing the different pain measurements in RCTs*
*and that we observed that improvement of symptoms by our primary phone measurements were*
*not reflected by some questionnaires (where we observed important analgesia in the NoTx arm).*
*This was only possible given the use of a NoTx arm documenting natural history of pain. See*
*Reviewer 2.2. for the covariance matrix across pain measures at baseline. As displayed, some*
*pain assessments poorly correlated together at baseline, prior to placebo pills. This has also*
*been added in **fig. S3**.*

*Regarding memory of pain, we just published a paper on memory bias and how it relates to*
*phone app ratings (Berger et al NeuroImage 2018). This measure was collected because it*
*represents a verbal report of pain on a numerical scale, a common assessment that many*
*patients are expected to give when interacting with health care providers in their day-to-day*
*lives. This was, however, only used as a **secondary** pain assessment. The results here (and those*
*in Berger et al) show that memory of pain closely correlates with pain ratings that are recorded*
*in the moment via the phone; others have also shown a significant correspondence between pain*
*memory measures and instantaneous pain measures (see as one example: Jamison et al, 2006,*
*The Journal of Pain).*

10. In addition, presentation of all data as % change from baseline does not allow the reader to
properly inspect and understand the results. While the need for data reduction is obvious, percent
change arguably does not allow conclusions about clinical relevance of changes and “distorts”
actual changes.

Response: This is a good point, which was also raised by Reviewer 1.5 above. Note that there were no significant group differences between absolute scores across pain measurements (both primary and secondary). These results are now provided in Table 1.

11. At minimum, in a first step, the authors should provide more details on all pain measures across measurement points without group stratification into responders and non-responders and show how real changes (in mm or original scale scores) compare to % change used in subsequent analyses. It is crucial to clarify which of the pain measures actually responded to placebo intervention compared to no treatment. As is, Fig. 1c is the only such evidence that is provided within the manuscript and that is not sufficient.

Response: First, we would like to clarify that the greater incidence of response in the PTx arm (24/43) compared to the NoTx arm (4/20) was the actual evidence of placebo response ($\chi^2 = 7.09, p = 0.008$). This is critical because response was defined by **within-subject** analgesia when considering the variability of pain during a 2-week baseline period. This represents a major methodological improvement (more on EMAs below). This clarification is especially important as all our univariate brain analyses and our SVM classifiers were trained on this group stratification.

However, the reviewer's point is well taken and we recognize the importance of showing additional data in the original (absolute) scale. As a first step, the original scores at baseline are now provided in Table 1 for our primary and secondary pain assessments. Group differences on the pain measurements in their original scores are also provided below.

Starting with our primary pain measurement (phone app), we show the correspondence between the %change and difference in pain intensity for the initial comparison of the 43 PTx with the 20 NoTx.

Figure 9: Correspondence between %analgesia and the difference in the absolute scale scores between baseline and treatment periods. Compared to the NoTx group, PTx showed a 13.3% difference in average %analgesia from the phone app at treatment 1 (upper left, originally presented in Fig1c), which corresponded to an average of 0.71 units in absolute pain intensity. Similarly, compared to the NoTx group, PTx showed a 15.6% difference in average %analgesia (lower left), which corresponded to an average of 0.84 units in absolute pain intensity. We computed the effect size for the differences in absolute pain intensity scores, which corresponded to a Cohen’s $d' = 0.62$ (95% CI: 0.33 to 0.90) for treatment 1 and Cohen’s $d' = 0.73$ (95% CI: 0.45 to 1.01) for treatment 2. * $p < 0.05$; ** $p < 0.01$.

Note that the effect sizes seen here correspond to moderate-to-strong responses that are comparable to the ones observed in other chronic pain populations: placebo vs no treatment in OA patients $E.S. = 0.51$ (Zhang et al., 2008 Ann Rheum Dis); placebo vs no treatment in fibromyalgia patients $E.S. = 0.53$ Chen X et al. 2017 Clin Rheumatol). This information has been added in the main manuscript.

The following figure shows the difference between baseline and treatment in the original scale scores after stratification. This is the same figure presented in fig 1d but displaying the absolute scores instead of %analgesia. This figure was added in the supplementary material.

**Figure 10: The difference between baseline and treatment, in absolute scale, after**
 **stratification of PTx. a.** The y-axis shows the pain analgesia in absolute scale (from the phone
 app) across the RCT. Two-way repeated-measure ANOVA show a group effect: $F_{(2,57)} = 3.85$, p
 $= 0.027$. Post-hoc test: PTxResp vs PTxNonR $p < 0.02$. **b.** Compared to the NoTx group,
 PTxResp showed an average of 1.34 units in pain intensity difference in treatment 1 and 1.27
 units in pain intensity difference in treatment 2. * $p < 0.05$; *** $p < 0.01$. Post hoc comparisons are
 Bonferroni corrected for 3 comparisons.

 Lastly, because our study design included 2 treatment periods, we also determined the highest
 1349 %analgesia between the 2 treatment periods for each patient, which was referred in the
 1350 manuscript as the magnitude of analgesia. The magnitude of the response is reported below both
 in %analgesia and in original scale scores.

**Figure 7: The magnitude of response in %analgesia and absolute scale score.** The magnitude
 of analgesia refers to the patients' highest pain diminution that occurred in either of the two
 treatment periods. Compared to the NoTx group, PTxResp showed a 30% difference in highest
 1356 %analgesia (left panel). This correspond to a 1.70-unit pain intensity difference (right panel).
 *** $p < 0.001$. Post hoc comparisons are Bonferroni corrected for 3 comparisons. These 2 panels
 have been included in figure 1.

 A number of secondary pain measurements were collected to determine the extent to which they
 differed from the EMAs ratings used as a primary pain measurement. The NRS refers to a single

numerical rating scale completed in the lab at each visit. As expected, this measure reflected the
results of the phone app but with more variability (because the phone app ratings were averaged
across two weeks of baseline and compared to the averaged ratings across the last week of
treatment).

**Figure 8: The difference between NRS scores entered at baseline and after treatment in the**
**original scale score before stratification.** The NRS reflected our primary pain measurement
results. & $p < 0.09$ * $p < 0.05$.

We expected no group difference on measures of pain quality, as these pain measurements went
down in all groups (including the NoTx group). Accordingly, the results showed no greater pain
improvement in the PTx group on MPQa and MPQs scales for both treatment periods (all $ps >$
0.283). Pain detect showed no group difference for treatment 1 period ($p = 0.75$), but showed an
unexpected barely significant group difference at treatment 2 ($p = 0.044$), where the NoTx group
actually showed higher pain diminution compared to PTx. As in the original version of this
manuscript, we concluded that pain quality was not diminished by the administration of a
placebo pill. Instead, qualities of pain generally showed improvement across the study in all
groups, including in the NoTx arm.

Together, these findings support our initial conclusion: 1- placebo interventions decreased pain
intensity and 2- placebo intervention did not decrease measures of pain quality.

12. Based on pain data provided e.g. in Fig. S3, it appears that placebo treatment in this study
was possible not superior to no treatment (?). After stratifying into responders and non-
responders, group differences in pain ratings (e.g. as indicated in Fig. S3) merely reflect the
stratification.

**Response:** This is a good point, and we believe that the figure was misleading. In fig. S3, the
pain ratings were ordered from 0-11 (irrespective of the time lag between pain ratings across the
patients) and averaged together. The idea was to display the immediacy of analgesia (e.g., how
pain relief was time-locked with the introduction of treatment). However, there was an average
of 24 pain ratings during the treatment 1 period, and the analgesia measured on the phone app
was progressively increasing throughout the treatment period, with peak relief occurring at the
end of the treatment period. Fig 1d shows this decrease when organizing the pain ratings into
four bins matching the time lag (i.e. matching the date of entry of pain ratings across all
patients). Moreover, one can appreciate the effect size of the placebo response in figures 5-7

presented above. We chose to remove the initial **fig. S3** to avoid any confusion about effect size
 of the placebo response in PTxResp. The immediacy of the response is, however, still reported in
 **fig 1f** as the analgesia observed the first day of treatment.

 13. Table S1 should provide basic demographics for groups as shown in Fig. S1 rather than for
 responders and non-responders. The table legend states “There were no differences between
 groups in age, gender, duration of pain reported, or years of education” – is this the result of
 some sort of statistical analysis or based on visual inspection? Given the small N, it appears
 difficult to properly analyze these data statistically and visual inspection at least does not fully
 support that all groups were equal, especially for proportion of females and duration of pain.

 **Response:** It is unclear exactly which groups within Figure S1 the reviewer is requesting that we
 translate into Table S1. Based on the questions immediately following this comment and the fact
 that initial groups were determined by randomization, we assumed that the request was to look
 at patients who completed the study and simply summarize demographics based on those treated
 with placebo (n=43) and those in the no-treatment group (n =20), as opposed to further
 subdividing by response as we had originally done.

	Age (years)	Female (%)	Pain duration (months)	Education (years)
PTx	46.1 (12.1)	14 (33)	52.8 (80.1)	12.3 (3.5)
NoTx	46.2 (13.2)	10 (50)	57.6 (61.2)	13.8 (3.7)
p_vals	0.98	0.18	0.81	0.15

 **Table S1: Basic Demographics.** There were no significant differences between treatment
 groups in age (unpaired t-test; $t = -0.024$, $p = 0.98$), gender ($\chi^2(df:1)=1.76$, $p=0.185$), duration
 of pain reported (unpaired t-test; $t = -0.237$, $p = 0.81$), or years of education (unpaired t-test; t
 $= -1.47$, $p = 0.15$). Table shows mean \pm STD; all variables here were reported at visit 1.

 We apologize for not including the statistics in the first version.

 14. Since the main goal was to predict placebo response: what was the amount of variance that
 could be explained with psychological (questionnaire) versus biological (imaging) variables?

 **Response:** Univariate results: the r^2 (variance explained) are already provided in fig. 2-3-4.
 Multivariate results: regarding the actual prediction from the nested leave one out cross-
 validation procedure, the r^2 between predicted analgesia and observed analgesia are displayed
 in fig. 5. The specific comparisons regarding the 2 models has also been addressed with greater
 detail on p.15 line 342.

 Discussion:

15. As in the introduction, the results are not properly discussed in the context of existing
 placebo brain imaging studies and placebo trials.

 **Response:** We have somewhat modified the discussion and now include more comparisons with
 earlier literature.

16. How does the finding that pain intensity (assessed with daily ratings) responds to placebo
treatment and validated other pain measures do not respond compare to the literature? Should
existing “gold standards” for assessing pain in pain trials be changed?

**Response:** *These results are novel and do not readily compare with existing studies. However,*
*they do not contradict existing studies either. Pain intensity is currently, almost universally, used*
*as the primary outcome measure in clinical trials. Most therapies FDA approved for chronic*
*pain are based on this measure. Our results confirm this clinical tradition. Our results provide*
*important new insight regarding qualitative PROs. In a sense, they suggest that if a trial shows*
*improvement with the latter but not with pain intensity then most likely this is a non-specific*
*effect. On the other hand, if a study shows improvement both with PRO and with intensity*
*superior to placebo then there is a good chance that the effect is specific to the treatment itself.*
*Note, none of these outcomes contradict existing standards. What we are criticizing is the use of*
*a single pain measurement to define improvement of symptoms. In fact, the field is moving more*
*and more away from traditional methods that assess pain sparsely at pre-determined time points*
*and increasing towards ecological momentary assessments (EMAs) that prompt patients for pain*
*in real-time and in their natural environments; some literature even goes as far as saying that*
*these EMAs (electronic or diary based, both of which have been validated) are becoming the new*
*“gold standards” for pain trials and chronic pain management tracking (see Salaffi et al, 2015,*
***Best Practice & Research Clinical Rheumatology****). Additionally, the pain field is also seeing*
*value in going beyond cross-sectional unidimensional pain scales (e.g., the in-clinic NRS). Our*
*study provides an example of how researchers and clinicians can use new technologies and*
*methodologies to capture pain and treatment dynamics, as well as study how these new measures*
*relate with other validated secondary pain assessments. We believe studies should include more*
*secondary pain measurements to better document and interpret the effectiveness of any given*
*treatment. Although we observed different effects of placebo pills on different pain*
*measurements, we believe that this represented a strength of the study rather than a weakness.*
*We hope these results will have an impact on future studies for collecting additional pain*
*measurements to better understand chronic pain and the treatment outcomes for this disease.*

17. Conclusions about correspondences in the mechanisms across different types of pain and
settings are speculative and not referenced properly.

**Response:** *This simply refers to the correspondence between the current study and our earlier*
*placebo study and healthy controls. We modified this statement and properly referenced our*
*previous study. Regarding correspondence with healthy controls, the whole paragraph describes*
*the brain regions involved in experimental placebo in healthy individuals.*

The possibility that prediction of placebo effects is better in patients with chronic pain in the
context of a clinical trial is pure speculation and in fact counter-intuitive especially due to
different individual treatment histories.

**Response:** *We removed this statement from the conclusion.*

The authors did not assess expectations – the mediator of placebo effects – which is not
acknowledged in the discussion section.

**Response:** *We did assess expectations via the SETS questionnaire prior to each treatment period*
*(please see our response to point # 4 above for why this information wasn't included in the first*
*version of the manuscript). The results show no group difference in positive and negative*
*expectations. Although we recognize that there is abundant literature on the topic (see the*
*discussion section), our data shows that expectations were not mediating placebo effects (at least*
*as measured with the SETS). We rather observed that other factors, such as VLPFC and DLPFC*
*connectivity and other psychological factors predicted placebo response.*

The conclusion that placebo effects have their own psychological and neurobiological
determinants is almost trivial at this stage given the abundance of research on the topic.

**Response:** *We modified our conclusions p.21 line 484: “Our results contribute to the placebo*
*literature by demonstrating the existence of psychological and neurobiological principles*
*determining the placebo response in RCTs.”*

Reviewers' comments:

Reviewer #1 (Remarks to the Author):

The authors have provided an immensely comprehensive rebuttal to the previous reviews, which is commendable. I still see the final manuscript as too dense (and quite long) for a general report, but I won't labour that point further. The research was performed to an exceptional standard and the findings will have a substantial impact on the field.

Reviewer #2 (Remarks to the Author):

This is a unique study and a massive effort. To follow 60 patients across 6 lab visits and 4 imaging sessions each (and including EMA measures daily) is a laudable effort, the findings from which will be very valuable for the field. The authors' recent demonstrated commitment to sharing the data from similar past efforts also increases the value and longevity of the work. Other strengths include the blinding of the analysis, cross-validation of predictive models, and novelty of the results predicting placebo responses, and the great importance of predicting placebo responses in medicine.

I have reviewed the responses to the prior reviews and re-read the manuscript. I am convinced that there is much value to the study and it will be a very important, highly cited contribution to the field. Placebo vs. no-treatment effect sizes are significant and clinically meaningful in size (comparable in effect sizes to many active treatments). The dichotomization of patients into responders and non-responders is defensible because it is a standard in the field (though there are alternatives, i.e., analyzing continuous measures, that many would prefer), and the authors supplement this with continuous regressions, which is very helpful. Overall, this is a very well-done and important study, and I am highly enthusiastic.

The manuscript is complex, but this stems in part from the need to present many results, which is essential for the science. There are many, many examples of similar or greater complexity (and the use of many panels to display results) in neuroscientific papers. The authors have worked to clarify the manuscript and findings, and I believe that this has been helpful and the manuscript is readable by a broad audience.

The low correlations among pain measures is an interesting addition, and points to the complexity of assessing pain in different ways and at different time scales.

I have only a few additional suggestions.

To capitalize on the power of the RCT, placebo must be compared to no-treatment. The basic results and statistics do this, but Fig. 1 is full of statistical comparisons (and plots) of responders vs. non responders, and responders vs. no treatment. These comparisons are statistically invalid. One simply cannot condition on a variable (placebo response) and then compare the conditional responses (responders) to no-treatment and perform a meaningful statistical test. This does not invalidate the paper as the relevant comparison, placebo versus no-treatment, is significant. But it is distracting and misleading for most readers. It is valid to separate responders and nonresponders and then look for predictors (brain, psychological) of this difference, so the main results are valid. I recommend removing stars/stats for invalid statistical tests from Fig. 1 and elsewhere, and including bars in all panels for overall placebo, no treatment, and then separate bars for responders and non-responders.

The legend in Fig 2 is incomplete, because it does not describe what all the abbreviations mentioned in the figure mean. In addition, I could not find information on what the subscales of MAIA mean (a, e., etc.), which are some of the few significant associations. Please add more

description. For example, do high scores on “nd” mean that participants were not distracted by internal bodily sensations, or that they are more distracted? Or something else? Finally, some measures of interest (suggestibility, LOT-R, others) do not appear in Fig 2, and don’t seem to be reported. This is a minor comment requesting clarification.

Having said that, the findings on placebo responses being related to emotional awareness, mindfulness, and openness are very interesting!

To “disentangle placebo pill-related analgesia from non-specific effects”, an important goal, predictors of improvement must be more predictive in the placebo than no-treatment group. The revision and response are confusing on this point, but this is partially addressed in the section, “The Predictive models did not generalize to spontaneous improvement of symptoms observed in the NoTx arm”. However, other results prior to Fig 5 seem to suggest it’s not necessary to test this, and the results are presented with 3 groups — responders, non-responders, and no-treatment. What is missing is information about no-treatment “responders”, which reflects the natural history control in this case. If a brain finding is related to placebo responders vs. non responders but *unrelated* to no-treatment responders vs. non-responders (and there is a sig. difference in these associations), there is evidence that the placebo treatment mattered. If the brain finding is related to placebo responders vs. non responders and also no-treatment responders vs. non-responders, it may be a measure related to placebo OR natural history/spontaneous improvement irrespective of placebo treatments, and the study cannot distinguish whether it is placebo-caused or not. Identifying which brain findings are in each of these categories should, in my opinion, be an important part of the results of the paper. My suggestion is that this could be a secondary characterization of the findings presented in other parts of the paper. Also, could the others perform significance tests to test whether brain measures were significantly MORE predictive in placebo than no-treatment groups? Such tests are readily available, and they would help assess what we can and cannot claim from these results.

“we designed a comprehensive RCT with two identical treatment periods”...with what treatments? Both placebo?

“responses to placebo pills is predetermined “...are.

“63 patients were dichotomized into Resp and NonR based on a permutation test “ ... what is the cutoff for classification?

“the magnitude of response in the PTxResp was 1.7 units higher than in the NoTx arm (E.S.: 1.71 [95% CI: 1.42 to 2.0]), which corresponded to a 33% analgesia” I am not in favor of reporting CIs here because the “best of two periods” capitalizes on chance. This analysis is not necessary to demonstrate statistical or clinical significance in any case.

“Co-activation maps derived from these seeds (green) (1000 healthy subjects; <http://neurosynth.org>) “ This is confusing as 1000-subject FC and neurosynth appear to be different datasets?

““consensus” was generated by averaging the weights across the n=43 loops to create a final single set of weights “ an alternative to doing this, which is probably preferred, is to run the model on the full dataset for purposes of obtaining the final weights. In this case, (1) bootstrapping and (2) permutation tests can be used to (2) identify significant predictive features and (2) test the full model for significance and bias, respectively. Cross-validation is a separate procedure used for assessing model performance. But the averaging method is not invalid.

Reviewer #3 (Remarks to the Author):

I am satisfied with the authors' responses to my concerns.

Reviewer #4 (Remarks to the Author):

I am happy with the detailed responses and changes to the manuscript.

We are pleased to see that reviewer 1,3, & 4 are satisfied with our revisions. We believe that Reviewer 2 made some good comments in this additional round of revisions that improves the overall manuscript. The reviewer pointed to invalid comparisons in figure 2 and asked for clarifications regarding the specificity of the model for placebo analgesia. We changed figure 2 accordingly and provided an additional figure in the supplementary material showing that brain properties were not associated with improvement of symptoms in the NoTx arm. The reviewer's queries are addressed point by point below. The modifications are highlighted in blue and the initial changes from the previous round of revision are still highlighted in yellow. We hope our response will be satisfactory.

Reviewer 2:

1-To capitalize on the power of the RCT, placebo must be compared to no-treatment. The basic results and statistics do this, but Fig. 1 is full of statistical comparisons (and plots) of responders vs. non responders, and responders vs. no treatment. These comparisons are statistically invalid. One simply cannot condition on a variable (placebo response) and then compare the conditional responses (responders) to no-treatment and perform a meaningful statistical test. This does not invalidate the paper as the relevant comparison, placebo versus no-treatment, is significant. But it is distracting and misleading for most readers. It is valid to separate responders and nonresponders and then look for predictors (brain, psychological) of this difference, so the main results are valid. I recommend removing stars/stats for invalid statistical tests from Fig. 1 and elsewhere, and including bars in all panels for overall placebo, no treatment, and then separate bars for responders and non-responders.

Response: This is a valid point noted by reviewer 2. We followed the recommendation and now provide both comparisons between PTx Vs NoTx (effect of placebo pills) as well as between PTxResp Vs PTxNonR (effect of placebo response) on primary and secondary pain outcomes. The figure 1 has been modified as follow. The main text and the figure legend have been slightly modified according to these analyses (the changes are highlighted in blue in the manuscript).

2-The legend in Fig 2 is incomplete, because it does not describe what all the abbreviations mentioned in the figure mean. In addition, I could not find information on what the subscales of MAIA mean (a, e., etc.), which are some of the few significant associations. Please add more description. For example, do high scores on “nd” mean that participants were not distracted by internal bodily sensations, or that they are more distracted? Or something else? Finally, some measures of interest (suggestibility, LOT-R, others) do not appear in Fig 2, and don’t seem to be reported. This is a minor comment requesting clarification.

Having said that, the findings on placebo responses being related to emotional awareness, mindfulness, and openness are very interesting!

Response: We are thankful for the reviewer’s enthusiasm. Note that LOT-R was presented (8th variable in the matrix starting from the last). If other measures aren’t shown (such as the Multidimensional Iowa Suggestibility Scale – MISS), it’s because we only collected these measures at the very last visit (not at the first visit, preceding treatment), so there was no way to test whether they were predictive of placebo response or magnitude (even if they may have been related). Supplementary Table S4 lists the visits that they were administered.

MAIA has 8 subscales total. Noticing captures a patient's ability to realize when and where in the body they are comfortable or uncomfortable; Attention Regulation patient's ability to focus and control attention away from or towards their body. Emotional Awareness measures how well patients are aware of their emotions and how their body states can influence their emotions and vice versa. Self-Regulation measures the ability to use breath and mental awareness to make someone feel calmer or less tense. Body-Listening is similar to Noticing but involves a patient's tendency to use their body as a source of information about how well they are doing or what they should do. Trusting captures the extent to which patients rely on what their bodies tell them and feel like they can trust the sensations. Not-Distracting captures whether bodily sensations (including pain and discomfort) are able to distract patients or whether they can distract themselves from these things. Finally, Not-Worrying measures how much patients worry or become upset by discomfort or pain.

6 of the 8 are scored in an additive manner, meaning the answers are averaged within a subscale; these are: Noticing, Attention Regulation, Emotional Awareness, Self-Regulation, Body Listening, and Trusting. The remaining 2 subscales – Not-Distracting and Not-Worrying – use a combination of additive and reverse scoring (with averaging after). Reverse scoring for these two scales essentially puts all subscales along the same direction for easier and consistent interpretation. In this case, the higher the score of a subscale, the more of that quality or tendency a patient has. For example, if a patient scored high on both noticing and not-worrying, it would mean that they have increased awareness of body sensations (uncomfortable, comfortable, or neutral) AND they have the tendency to not worry or experience emotional distress when they have pain. In the case of the correlations shown in Figure 2, patients with increased emotional awareness (awareness of the connection between bodily sensations and emotions) experience greater placebo analgesia, whereas patients who tend not to ignore or distract themselves from discomfort (i.e., they focus more on it), experience less placebo analgesia.

All abbreviations of the subscales have been described in fig 2 legend and the main results are better explained.

3-To “disentangle placebo pill-related analgesia from non-specific effects”, an important goal, predictors of improvement must be more predictive in the placebo than no-treatment group. The revision and response are confusing on this point, but this is partially addressed in the section, “The Predictive models did not generalize to spontaneous improvement of symptoms observed in the NoTx arm”. However, other results prior to Fig 5 seem to suggest it's not necessary to test this, and the results are presented with 3 groups — responders, non-responders, and no-treatment. What is missing is information about no-treatment “responders”, which reflects the natural history control in this case. If a brain finding is related to placebo responders vs. non responders but *unrelated* to no-treatment responders vs. non-responders (and there is a sig. difference in these associations), there is evidence that the placebo treatment mattered. If the brain finding is related to placebo responders vs. non responders and also no-treatment responders vs. non-responders, it may be a measure related to placebo OR natural history/spontaneous improvement irrespective of placebo treatments, and the study cannot distinguish whether it is placebo-caused or not. Identifying which brain findings are in each of these categories should, in my opinion, be an important part of the results of the paper. My suggestion is that this could be a secondary characterization of the findings presented in other parts of the paper. Also, could the others perform significance tests to test whether brain measures were significantly MORE predictive in placebo than no-treatment groups? Such tests are readily available, and they would help assess what we can and cannot claim from these results.

Response: We generated a new figure in the supplementary material to address these specific questions. Please note that the sample sizes were very small and we therefore presented boxplots and used non-parametric tests (Mann-Whitney U test). The figure below presents the group comparisons in the NoTx arm on each brain parameters dissociating PTxResp from PTxNonR.

First, we compared the NoTxResp Vs NoTxNonResp on all brain parameters and none of them were close to be significant. Next, we applied the classifier and the regression model using only the psychological factors to the patients to predict analgesia in the NoTx group. The classifier was 65% accurate and the regression model predicted $r^2 = 0.06$ (not shown) of the actual analgesia. Both were considered not significant. Finally, we determined if the final model (presented in figure 5 k) was more predictive in the PTx group compared to the NoTx. The r^2 between predicted and actual analgesia were respectively 0.36 in the PTx group and 0.1 in the NoTx group (fig. 6d). We converted these into correlation coefficients ($r = 0.6$ and 0.1) and applied a fisher z-transformation to compare them using a z test statistic ($z = 2.00$, $p = 0.046$; two-tail). We conclude that 1) our predictive model from the joint contribution of psychological factors and brain properties was not informative to predict improvement of symptoms in the Notx group and that 2) the final model was significantly more predictive in the PTx group. This is described as follows in the results p.16 line 361:

‘The Predictive models did not generalize to spontaneous improvement of symptoms observed in the NoTx arm

We finally tested if our models were specific for placebo pills analgesia or if they were predicting unspecific improvement of symptoms. We first used univariate statistics to compare NoTxResp with NoTxNonResp on the brain properties dissociating PTxResp from PTxNonR (fig. 3-4). There were no differences on any of these parameters in the NoTx group suggesting that our results were specific for placebo response (fig. S11). Second, we tested if our multivariate classifier based on psychological factors (fig. 5a) was capable of stratifying the patients randomized in the NoTx arm. The classifier accuracy was considered non-significant (0.65) and the difference between predicted response rate in the NoTxResp was not higher than in the NoTxNonR ($\chi^2 = 1.82$ $p = 0.18$; fig. 5i). Finally, we applied the regression model predicting the magnitude of response (fig. 5k) to patients in the NoTx group, and here again, the model was inaccurate in the NoTx (fig. 5i). In this case, the coefficient of correlation between the predicted and the actual analgesia was stronger in the PTx group (fig. 5i) than in the NoTx group (fig.

5i; z test = 2.00, $p = 0.046$), indicating that the model was more predictive in individuals ingesting placebo pills.’

“we designed a comprehensive RCT with two identical treatment periods” ...with what treatments? Both placebo?

Response: We clarified that these were identical placebo or active treatment periods depending upon randomization. In other words, individuals receiving active treatment were never switched to placebo treatment and individuals receiving placebo were never switched to active treatment (this was not a cross-over design).

“responses to placebo pills is predetermined “ ...are.

Response: This has been corrected.

“63 patients were dichotomized into Resp and NonR based on a permutation test “ ... what is the cutoff for classification?

Response: Cutoff was set to $p < 0.05$. This has been clarified in the manuscript.

“the magnitude of response in the PTxResp was 1.7 units higher than in the NoTx arm (E.S.: 1.71 [95% CI: 1.42 to 2.0]), which corresponded to a 33% analgesia” I am not in favor of reporting CIs here because the “best of two periods” capitalizes on chance. This analysis is not necessary to demonstrate statistical or clinical significance in any case.

Response: This is a valid point and we opt to remove the CIs.

“Co-activation maps derived from these seeds (green) (1000 healthy subjects; <http://neurosynth.org>) “ This is confusing as 1000-subject FC and neurosynth appear to be different datasets?

Response: This was clarified.

““consensus” was generated by averaging the weights across the $n=43$ loops to create a final single set of weights “ an alternative to doing this, which is probably preferred, is to run the model on the full dataset for purposes of obtaining the final weights. In this case, (1) bootstrapping and (2) permutation tests can be used to (2) identify significant predictive features and (2) test the full model for significance and bias, respectively. Cross-validation is a separate procedure used for assessing model performance. But the averaging method is not invalid.

Response: We initially opted to average the weights because it reflects what was used for prediction in the cross-validation procedure. Here, the model is very sparse (few connections) and the idea was not to identify the significance of each of these connections. The importance of the contributors was instead determined by average weight and the number iteration where the connection was selected in the LOOCV procedure. Note that this strategy has been used previously by several groups (such as Siegel JS et al., 2016 **Disruptions of network connectivity predict impairment in multiple behavioral domains after stroke**. Proc Natl Acad Sci U S A.).

REVIEWERS' COMMENTS:

Reviewer #2 (Remarks to the Author):

The authors have done a good job overall in responding to the issues raised. I have no further major issues.

However, some of the problematic comparisons (comparisons of responders to non-responders on the same variable used to define groups) that were adjusted in some analyses are still present in others. I do not think this requires re-review, but the authors should correct these problems in other parts of the manuscript. For example, plots in Fig S5 suggest that the raw reported scores do not show as strong a placebo effect as the data after rescaling, and the critical placebo (Tx) vs. no-treatment comparison is not made. Instead, the statistics report invalid post-hoc comparisons, as raised in the last review. Figure 1 still reports some invalid comparisons as well. So I recommend that these are fixed throughout the manuscript before final publication.

Reviewer #2 (Remarks to the Author):

The authors have done a good job overall in responding to the issues raised. I have no further major issues.

However, some of the problematic comparisons (comparisons of responders to non-responders on the same variable used to define groups) that were adjusted in some analyses are still present in others. I do not think this requires re-review, but the authors should correct these problems in other parts of the manuscript. For example, plots in Fig S5 suggest that the raw reported scores do not show as strong a placebo effect as the data after rescaling, and the critical placebo (Tx) vs. no-treatment comparison is not made. Instead, the statistics report invalid post-hoc comparisons, as raised in the last review. Figure 1 still reports some invalid comparisons as well. So I recommend that these are fixed throughout the manuscript before final publication.

Response: We modified Supplementary Figure 5 accordingly. Note that the raw reported score do show a strong placebo effect (Supplementary Figure 5). Thus, the critical placebo (Tx) vs. no-treatment comparison was significant for both treatment periods. Regarding figure 1, we have removed all potential problematic comparisons in the figure.